# The spatiotemporal distribution of human pathogens in ancient Eurasia

Martin Sikora[1]✉, Elisabetta Canteri[2], Antonio Fernandez-Guerra[1], Nikolay Oskolkov[3], Rasmus Ågren[4], Lena Hansson[5], Evan K. Irving-Pease[2], Barbara Mühlemann[6,7], Sofie Holtsmark Nielsen[8], Gabriele Scorrano[1,9], Morten E. Allentoft[1,10], Frederik Valeur Seersholm[1], Hannes Schroeder[2], Charleen Gaunitz[1], Jesper Stenderup[1], Lasse Vinner[1], Terry C. Jones[6,7,11], Björn Nystedt[12], Karl-Göran Sjögren[13], Julian Parkhill[14], Lars Fugger[15,16,17], Fernando Racimo[2], Kristian Kristiansen[1,13], Astrid K. N. Iversen[1,17]✉ & Eske Willerslev[1,18,19]✉

Infectious diseases have had devastating effects on human populations throughout history, but important questions about their origins and past dynamics remain[1]. To create an archaeogenetic-based spatiotemporal map of human pathogens, we screened shotgun-sequencing data from 1,313 ancient humans covering 37,000 years of Eurasian history. We demonstrate the widespread presence of ancient bacterial, viral and parasite DNA, identifying 5,486 individual hits against 492 species from 136 genera. Among those hits, 3,384 involve known human pathogens[2], many of which had not previously been identified in ancient human remains. Grouping the ancient microbial species according to their likely reservoir and type of transmission, we find that most groups are identified throughout the entire sampling period. Zoonotic pathogens are only detected from around 6,500 years ago, peaking roughly 5,000 years ago, coinciding with the widespread domestication of livestock[3]. Our findings provide direct evidence that this lifestyle change resulted in an increased infectious disease burden. They also indicate that the spread of these pathogens increased substantially during subsequent millennia, coinciding with the pastoralist migrations from the Eurasian Steppe[4,5].

Pathogens have been a constant threat to human health throughout our evolutionary history. Until around 1850, at least a quarter of all children died before the age of one, and around another quarter before turning 15. Infectious diseases are estimated to have been responsible for more than half of these deaths[6]. Larger disease outbreaks have profoundly affected human societies, sometimes devastatingly affecting entire civilizations[7]. Infectious diseases have left lasting impressions on human genomes, as selective pressures from pathogens have continuously shaped human genetic variation[8,9]. Where and when different human pathogens first emerged, how and why they spread, and how they affected human populations are important but largely unresolved questions.

During the Holocene (beginning roughly 12,000 years ago), the agricultural transition created larger and more sedentary communities, facilitating pathogen transmission and persistence within populations[10]. Simultaneously, the rise of animal husbandry and pastoralism are thought to have increased the risk of zoonoses[3]. Technological advances, such as horses and carts, increased both mobility and the risk of disease transmission between populations[11]. It has been proposed that these changes led to the so-called 'first epidemiological transition' characterized by increased infectious disease mortality[3]. However, direct evidence remains scarce, and the idea is debated[12]. Palaeopathological examinations of ancient skeletons offer insights into past infectious disease burden[13], but are limited to the few diseases identifiable from the available tissue. Recent advances in ancient DNA (aDNA) techniques allow for the retrieval of direct genomic evidence of past microbial infections, enabling the reconstruction of complete

[1]Centre for Ancient Environmental Genomics and The Lundbeck Foundation GeoGenetics Centre, Globe Institute, University of Copenhagen, Copenhagen, Denmark. [2]Section for Molecular Ecology and Evolution, Globe Institute, University of Copenhagen, Copenhagen, Denmark. [3]Department of Biology, National Bioinformatics Infrastructure Sweden, Science for Life Laboratory, Lund University, Lund, Sweden. [4]Department of Biology and Biological Engineering, National Bioinformatics Infrastructure Sweden, Science for Life Laboratory, Chalmers University of Technology, Gothenburg, Sweden. [5]Definitive Healthcare, Gothenburg, Sweden. [6]Institute of Virology, Charité – Universitätsmedizin Berlin, corporate member of Freie Universität Berlin, Humboldt-Universität zu Berlin and Berlin Institute of Health, Berlin, Germany. [7]German Centre for Infection Research (DZIF), Partner Site Charité, Berlin, Germany. [8]Department of Bacteria, Parasites and Fungi, Statens Serum Institut, Copenhagen, Denmark. [9]Center for Molecular Anthropology for the Study of Ancient DNA, Department of Biology, University of Rome 'Tor Vergata', Rome, Italy. [10]Trace and Environmental DNA (TrEnD) Laboratory, School of Molecular and Life Sciences, Curtin University, Perth, Western Australia, Australia. [11]Centre for Pathogen Evolution, Department of Zoology, University of Cambridge, Cambridge, UK. [12]Department of Cell and Molecular Biology, National Bioinformatics Infrastructure Sweden, Science for Life Laboratory, Uppsala University, Uppsala, Sweden. [13]Department of Historical Studies, University of Gothenburg, Gothenburg, Sweden. [14]Department of Veterinary Medicine, University of Cambridge, Cambridge, UK. [15]Oxford Centre for Neuroinflammation, Nuffield Department of Clinical Neurosciences, John Radcliffe Hospital, University of Oxford, Oxford, UK. [16]MRC Human Immunology Unit, John Radcliffe Hospital, University of Oxford, Oxford, UK. [17]Nuffield Department of Clinical Neurosciences, Weatherall Institute of Molecular Medicine, University of Oxford, Oxford, UK. [18]Department of Genetics, University of Cambridge, Cambridge, UK. [19]MARUM Center for Marine Environmental Sciences and Faculty of Geosciences, University of Bremen, Bremen, Germany. ✉e-mail: martin.sikora@sund.ku.dk; astrid.iversen@ndcn.ox.ac.uk; ewillerslev@sund.ku.dk

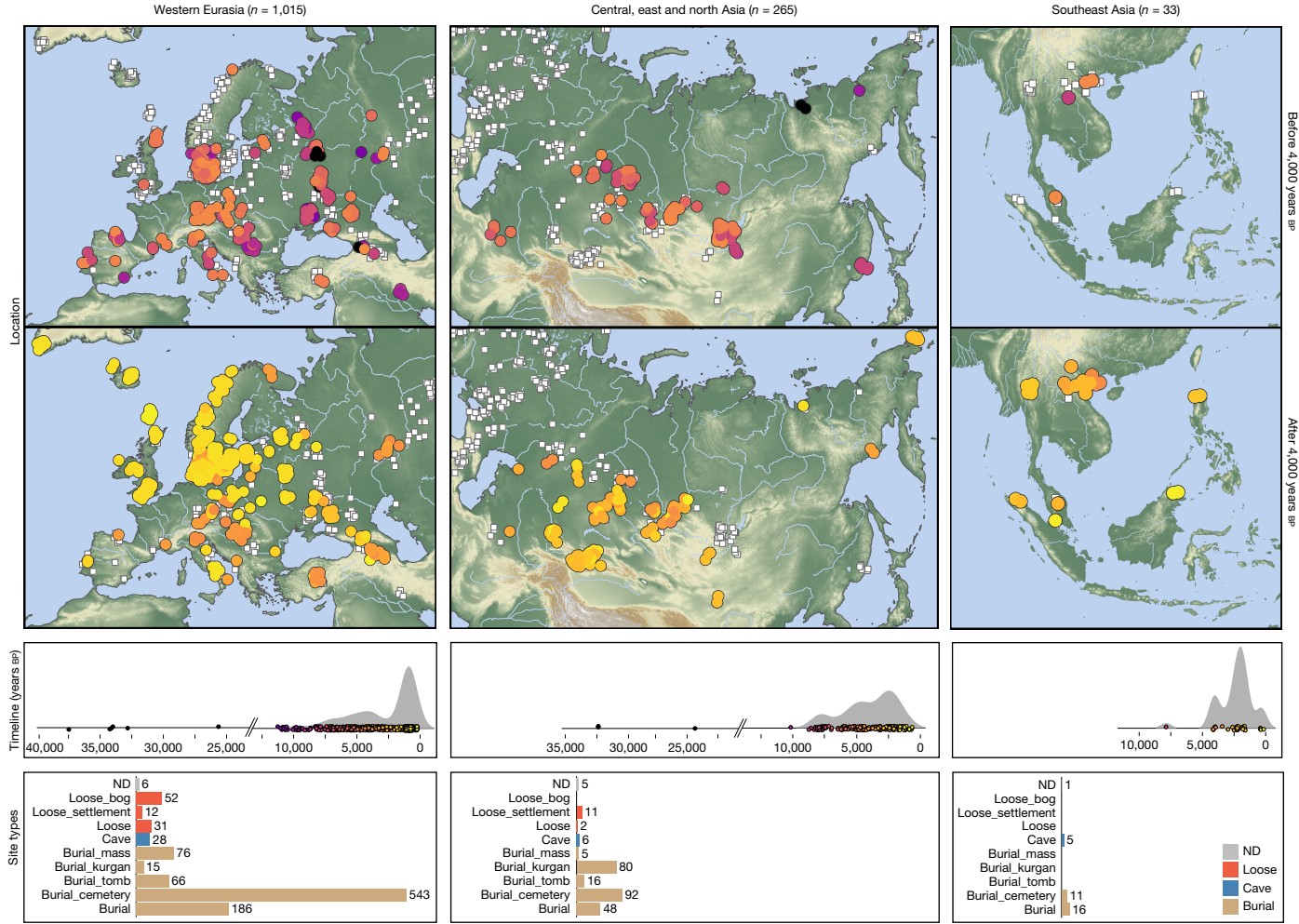

**Fig. 1 | Dataset overview.** Spatiotemporal distribution and site contexts of the study samples. White squares in the geographic maps indicate locations of the full set of $n$ = 1,313 study samples, whereas coloured circles highlight location and age of samples from the time period and region indicated in the respective panel. Bar plots show the number of samples obtained from different site type contexts in each region. ND, not determined.

ancient pathogen genomes. These studies have typically focused on specific pathogens and have provided surprising insights into the evolutionary history of the causative agents of some of the most historically important infectious diseases affecting humans, including plague (*Yersinia pestis*)[14,15], tuberculosis (*Mycobacterium tuberculosis*)[16,17], smallpox (variola virus)[18,19], hepatitis B (hepatitis B virus, HBV)[20–22] and others[23–28]. However, there is an unmet need to investigate the combined landscape of ancient bacteria, viruses and parasites that affected our ancestors across various regions and time periods. Here we use a new high-throughput computational workflow to screen for ancient microbial DNA and use our data to investigate long-standing questions in paleoepidemiology: when and where did important human pathogens arise? And what factors influenced their spatiotemporal distribution?

## Ancient microbial DNA from 1,313 Eurasians

To understand the distribution of ancient pathogens, we developed an accurate and scalable workflow to identify ancient microbial DNA in shotgun-sequenced aDNA data (Extended Data Figs. 1–4 and Supplementary Information 1). The data (roughly 405 billion sequencing reads) are derived from 1,313 ancient individuals from western Eurasia ($n$ = 1,015; 77%), central and north Asia ($n$ = 265; 20%) and southeast Asia ($n$ = 33; 3%), spanning a roughly 37,000-year period, from the Upper Palaeolithic to historical times (Fig. 1, Supplementary Table 1 and Supplementary Information 2). As burial practices varied across cultures

and time, these samples represent a subset of groups within past societies. Nevertheless, the identified pathogens probably affected the broader population, as diseases spread easily in communities with poor sanitation and hygiene[29]. Initial metagenomic classification showed a large fraction of reads classified as soil-dwelling taxa including genera such as *Streptomyces* or *Pseudomonas*, reflecting a predominantly environmental source of microbial DNA. Further characterization using a topic model, however, suggested that microbial DNA in ancient tooth samples often derives from genera commonly associated with the human oral microbiome such as *Actinomyces* or *Streptococcus* (Extended Data Fig. 1d–g).

We selected a set of 136 bacterial and protozoan genera (11,553 species total) containing human pathogenic species[2] as well as 1,356 viral genera (259,979 species total) for further authentication and detection of ancient taxa. We found that ancient microbial DNA was widely detected, with 5,486 authenticated individual hits identified across 1,005 samples (*Z*-score for aDNA damage rate from metaDMG of greater than or equal to 1.5; Fig. 2a, Supplementary Table 2 and Extended Data Fig. 4). Of those, 3,384 hits were found among 214 known human pathogen species[2], with the remaining 2,104 hits involving 278 other species. The highest numbers were observed in bacterial genera associated with the human oral microbiome, such as *Actinomyces* (380; 28.5% of samples) and *Streptococcus* (242; 18.1% of samples), or those commonly found in soil environments, such as *Clostridium* (252; 18.9% of samples) and *Pseudomonas* (111; 8.3% of samples).

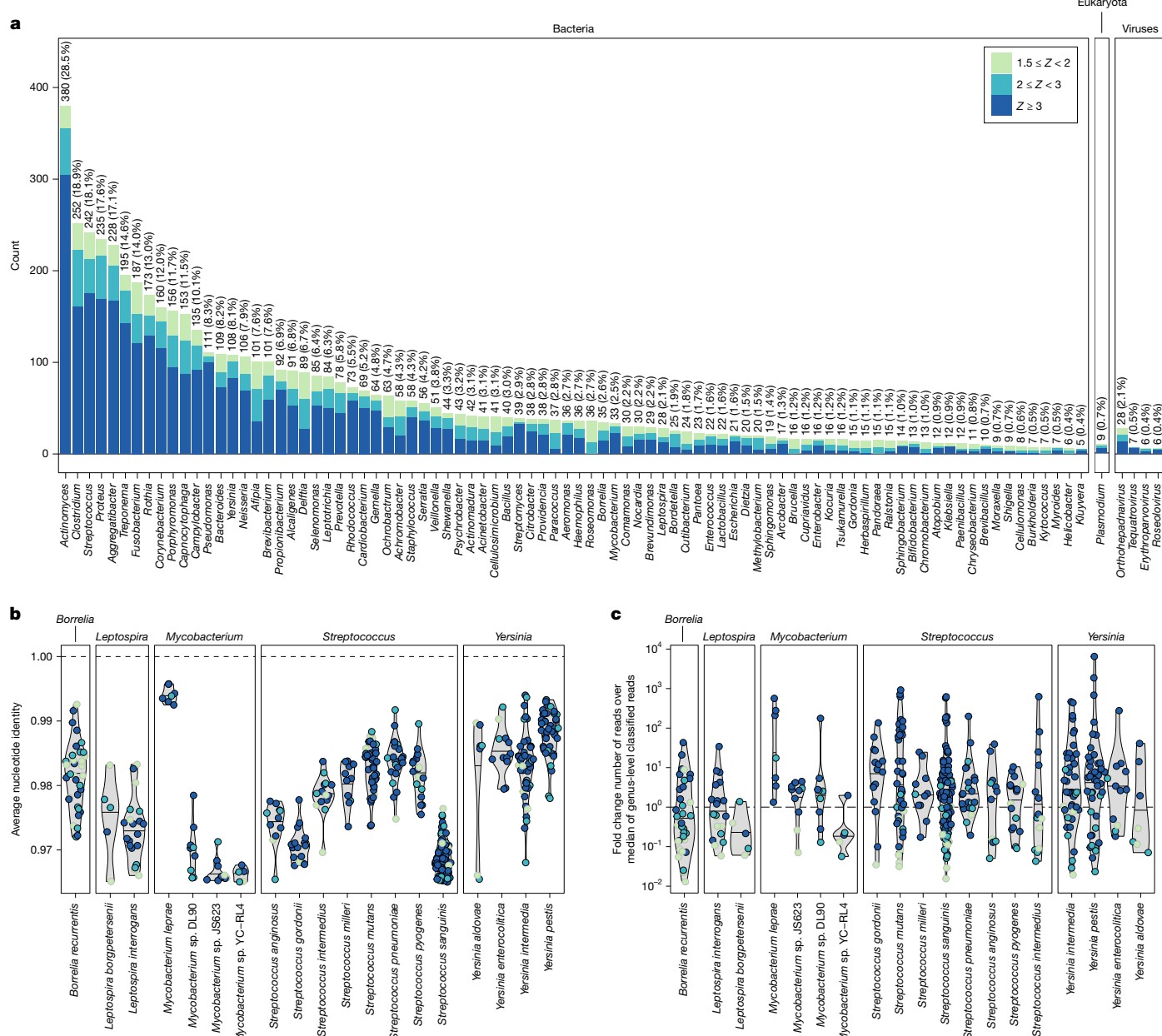

**Fig. 2 | Overview and characteristics of detected ancient microbial DNA.** **a**, Bar plot showing the total number of putative ancient microbial hits (overall detection rate in brackets) for bacterial, eukaryotic and viral ($n \geq 4$) genera. Bar colour and shading distinguishes counts in the different aDNA damage categories. **b,c**, Distributions of ANI (**b**) and $\log_{10}$ fold change of mapped reads over median of reads classified at taxonomic rank of genus per sample (**c**) for individual species hits in selected example genera. Symbol colour indicates aDNA damage category.

We observed marked differences in the distributions of the genetic similarity of the ancient microbial sequences to their reference assemblies, both among genera and between species within a genus (Fig. 2b and Extended Data Fig. 5). High average nucleotide identity (ANI) indicates that ancient microbial sequences are closely related to a reference assembly in the modern database, and was observed in hits across all species from some genera (for example, *Yersinia*, Fig. 2b). In other genera, only a few hits had a closely related database reference assembly match. An example is the genus *Mycobacterium*, in which only hits of the leprosy-causing bacterium *Mycobacterium leprae* were highly similar to their reference assembly (ANI > 99%, Fig. 2b). Low ANI indicates that the ancient microbial DNA is only distantly related to the reference assembly, for example, because of aDNA damage, poor representation of the diversity of the genus in the database or false-positive classification of ancient microbial reads deriving from a related genus (Extended Data

Fig. 3). Alternatively, ANI can also be reduced when reads mapped to a particular reference assembly originate from many closely related strains or species in a sample. To test for such mixtures, we quantified the rate of observing different alleles at two randomly sampled reads at nucleotide positions across the genomes of hits with read depths greater than or equal to one. We found a high multi-allele rate in many species associated with the human oral microbiome, such as *Streptococcus sanguinis* or *Treponema denticola*. Hits for these species also showed lower ANI, consistent with the expectation for mixtures of ancient microbial DNA (Extended Data Fig. 6a).

The rate of read mapping varied by orders of magnitude between species, from hits in species with high read recruitment, such as *M. leprae* (greater than 100-fold enrichment over the median number of classified reads across target genera) to hits at the lower limits of detection, for example, for the louse-borne pathogen *Borrelia recurrentis*

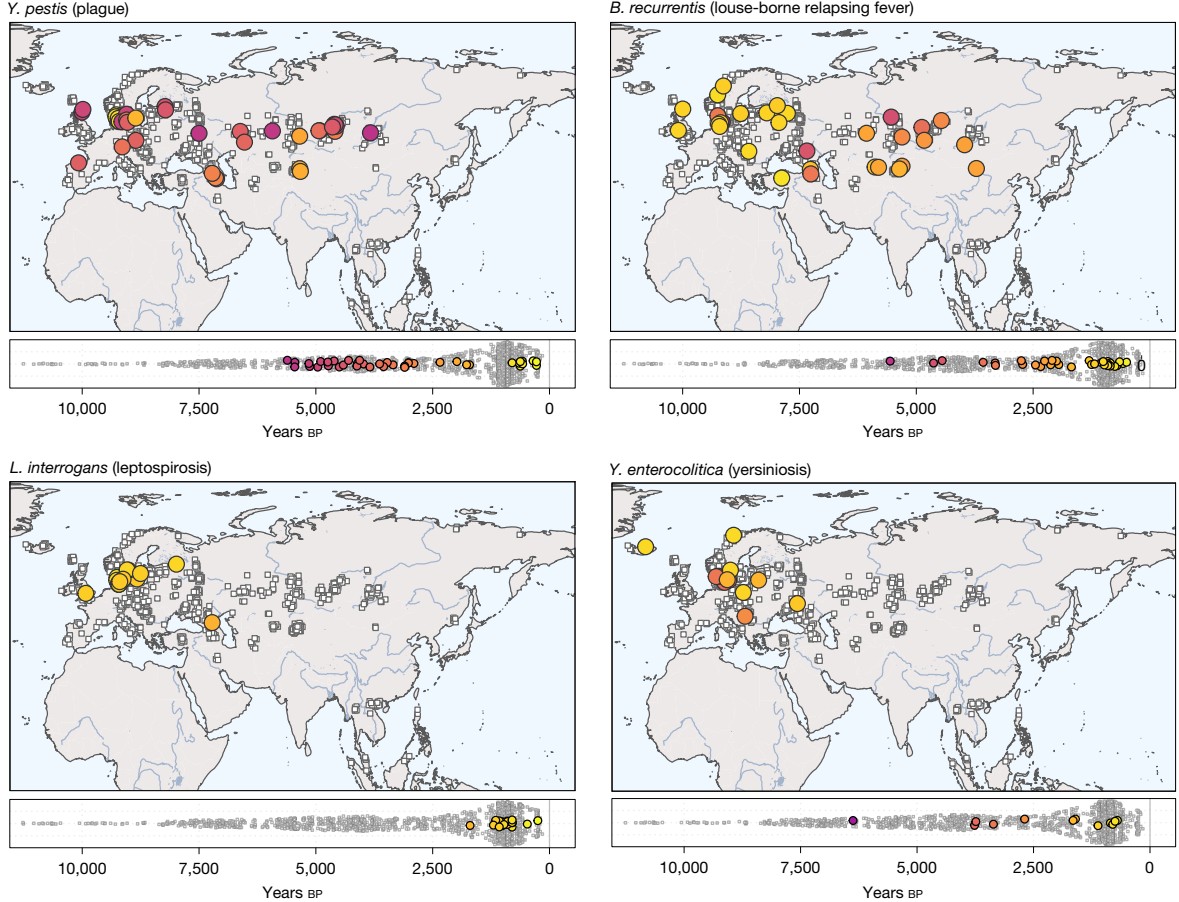

**Fig. 3 | Spatiotemporal distribution of selected ancient pathogens.**
Each panel shows geographic distribution (top) and timeline (bottom) for
identified cases of the respective pathogen (indicated by coloured circles).
Geographic locations and age distributions of all $n$ = 1,313 study samples are
shown in each panel using white squares.

(lowest read recruitment roughly 100-fold less than the median num-
ber of classified reads across target genera; Fig. 2c and Extended Data
Fig. 5b). Ancient microbial DNA from species commonly found in soil,
such as *Clostridium botulinum*, was detected at similar rates in tooth
and bone samples. Conversely, species associated with the human oral
microbiome (for example, *Fusobacterium nucleatum*, *Streptococcus
mutans* and *Porphyromonas gingivalis*) or pathogenic infections (for
example, *Y. pestis* and HBV) were significantly more frequently identi-
fied in tooth samples (Extended Data Fig. 6b). To further verify hits with
low read numbers, we performed a BLASTn search for all reads of each
hit with $n \leq 100$ final reads ($n$ = 712 hits total; Supplementary Table 3).
Most hits showed a high proportion (greater than or equal to 80%) of
reads assigned to the same species using BLASTn, and the species with
the most top-ranked BLASTn hits generally matched the inferred hit
species (Extended Data Fig. 7a,b).

Our results show that ancient microbial DNA isolated from human
remains originates from complex mixtures of distinct endogenous and
exogenous sources. The high detection rate, high read recruitment,
lower ANI and evidence of mixtures in genera such as *Clostridium* or
*Pseudomonas* (Fig. 2 and Extended Data Figs. 5 and 6) suggest that a
substantial fraction of this ancient microbial metagenome derives from
environmental sources, possibly associated with the 'necrobiome'
involved in post-mortem putrefaction processes[30,31] (Supplementary
Information 3). By contrast, species from other frequently observed
genera, including *Actinomyces* or *Streptococcus*, were predominantly
identified from teeth and probably originated from the endogenous
oral microbiome[30]. Species representing likely cases of pathogenic
infections (for example, *Y. pestis* and *M. leprae*) were often characterized

by higher ANI and/or low multi-allele rate, consistent with pathogen
load predominantly originating from a single dominant strain.

## The ancient Eurasian pathogen landscape

Our dataset provides a unique opportunity to investigate the origins
and spatiotemporal distribution of human pathogens in Eurasia,
expanding the known range of some ancient pathogenic species and
identifying others not previously reported using paleogenomic data
(Supplementary Tables 3 and 5).

Considering bacterial pathogens, we found widespread distribution
of the plague-causing bacterium *Y. pestis*, consistent with previous
studies[15]. We identified 42 putative cases of *Y. pestis* (35 newly reported,
Extended Data Fig. 6e), corresponding to a detection rate of roughly
3% in our samples. These newly identified cases expand the spatial and
temporal extent of ancient plague over previous results (Fig. 3). The
earliest three cases were dated between around 5,700–5,300 calibrated
years before present (cal. BP), across a broad geographic area rang-
ing from western Russia (NEO168, 5,583–5,322 cal. BP), to central Asia
(BOT2016, 5,582–5,318 cal. BP) and to Lake Baikal in Siberia[32] (DA342,
5,745–5,474 cal. BP). This broad range of detection among individu-
als predating 5,000 cal. BP challenges previous interpretations that
early plague strains represent only isolated zoonotic spillovers[33]. We
replicated previously identified cases of plague in Late Neolithic and
Bronze Age (LNBA) contexts across the Eurasian Steppe[15] and iden-
tified many instances in which several individuals from the same
burial context were infected (Afanasievo Gora, Russia; Kytmanovo,
Russia; Kapan, Armenia; Arban 1, Russia) (Supplementary Table 2).

These results indicate that the transmissibility and potential for local epidemic outbreaks for strains at those sites were probably higher than previously assumed[33]. Finally, 11 out of 42 cases were identified in late medieval and early modern period individuals (800–200 BP) from two cemeteries in Denmark (Aalborg, Randers), highlighting the high burden of plague during this time in Europe. All but one hit (NEO627, n = 84 reads total) showed expected coverage for the virulence plasmids pCD1 and pMT1, with hits before 2,500 years BP characterized by the previously reported absence of a 19 kilobase region on pMT1 containing the *ymt* gene[15] (Extended Data Fig. 8c and Supplementary Information 4).

Another bacterial pathogen frequently detected was the spirochaete bacterium *B. recurrentis*, the causative agent of louse-borne relapsing fever (LBRF), a disease with a mortality of 10–40% (Supplementary Information 5). Whereas previous paleogenomic evidence for LBRF is limited to a few cases from Scandinavia and Britain[23,34], we report 34 new putative cases (2.5% detection rate; Extended Data Fig. 6e), with wide geographic distribution across Europe, central Asia and Siberia (Fig. 3). We detected the earliest case in a Neolithic farmer individual from Scandinavia (NEO29, Lohals, 5,647–5,471 cal. BP), suggesting that human body lice were already vectors for infectious disease during the Neolithic period, supported by phylogenetic analyses of *B. recurrentis*[34]. The highest detection rates were found during the Iron and Viking Ages. LBRF outbreaks were historically associated with crowded living conditions, poor personal hygiene and wet and cold seasons, but are rare today in most regions (Supplementary Information 5). Our results indicate that *B. recurrentis* infections exerted a substantial disease burden on past populations.

We also report new cases of other bacterial pathogens previously detected in paleogenomic data. The leprosy-causing bacterium *M. leprae* was identified in seven individuals (0.5% detection rate) from Scandinavia and only appeared from the Late Iron Age onwards (earliest case RISE174, 1,523–1,339 cal. BP). Because *M. leprae* can infect both red squirrels and humans[35], and archaeological evidence demonstrates that fur trade from Scandinavia, including squirrel fur, increased substantially during the late Iron and Viking Ages[36], our results support the suggestion that squirrel fur trade could have facilitated transmission[37]. Our findings are also consistent with the widespread distribution of leprosy in medieval Europe[38]. We further detected three putative cases of *Treponema pallidum*—subspecies of which are the causative agents of treponematoses such as yaws, and endemic and venereal syphilis—in three individuals from recent time periods (earliest case 101809T, Denmark, 600–500 BP; Extended Data Fig. 7). Two cases were identified in individuals from Borneo in southeast Asia (around 500–300 years BP), expanding the range of paleogenomic evidence for treponemal disease into this region.

Among the species newly reported using paleogenomic data, we identified 12 putative cases of *Yersinia enterocolitica*, the causative agent of yersiniosis, commonly contracted through consuming contaminated raw or undercooked meat (Fig. 3). The animal reservoirs for *Y. enterocolitica* include boars, deer, horses, cattle and sheep. As *Y. enterocolitica* rarely enters the bloodstream, our results probably underestimate the disease burden. This species includes some of the only identified putative zoonotic infections in individuals from Mesolithic hunter-gatherer contexts (NEO941, Denmark, 6,446–6,302 cal. BP). We also detected other members of the order Enterobacterales, transmitted through the faecal–oral route, including members of the genera *Shigella*, *Salmonella* and *Escherichia* (Supplementary Table 2). We report the earliest evidence for ancient leptospirosis (genus *Leptospira*) dating back to the Neolithic, 5,650–5,477 cal. BP (NEO46, Sweden; *Leptospira borgpetersenii*). Whereas earlier cases predominantly involved *L. borgpetersenii* (n = 5, 0.4% detection rate), most hits were *Leptospira interrogans* (n = 20, 1.5% detection rate), almost exclusively in Scandinavian contexts from the Viking Age onwards (Fig. 3). *L. borgpetersenii* is today primarily found in cattle,

whereas *L. interrogans* is detected more broadly in both domestic and wild animals. Although the clinical manifestations are similar, with an untreated fatality rate of 1% today, transmission routes vary[39]. Although host-to-host transmission predominates for *L. borgpetersenii*, transmission by means of urine-contaminated environments dominates for *L. interrogans*. We also report two putative cases of *Corynebacterium diphtheriae*, the causative agent of diphtheria; the oldest of which dates back to the Mesolithic (Sidelkino, 11,336–11,181 cal. BP) (Supplementary Table 2).

Other diseases associated with animals and livestock, such as listeriosis (*Listeria monocytogenes*) and brucellosis (genus *Brucella*), could not be reliably identified. Another main human pathogen not identified in our dataset is *M. tuberculosis*, which causes tuberculosis. However, as the *M. tuberculosis* load in blood is typically low in immunocompetent patients without advanced disease[40] and latent tuberculosis develops in 60% of cases and can persist for decades, it is, on the basis of current knowledge, unlikely to be readily identified using aDNA data from tooth and bone remains sampled for ancient human DNA.

Identifying eukaryotic pathogens is challenging as sequence contamination from other organisms frequently occurs in their large and often fragmented reference genomes[41]. An illustrative example in our dataset is the protozoan parasite *Toxoplasma gondii*, which we readily identified in hits with high ANI and aDNA damage but low support from coverage evenness statistics owing to reads mapping to short contigs representing human contamination (Extended Data Fig. 4a,b). Despite these challenges, we identified nine putative malaria infections across three different human-infecting species (*Plasmodium vivax* n = 5; *Plasmodium malariae* n = 3; *Plasmodium falciparum* n = 1; Extended Data Fig. 7 and Supplementary Table 2). The most widely detected parasite species was *P. vivax*, with the earliest evidence in a Bronze Age individual from central Europe (RISE564, 4,750–3,750 BP based on an archaeological context). Other cases include a medieval individual from central Asia (DA204, Kazakhstan; 1,053–1,025 cal. BP) and two Viking Age individuals from eastern Europe (VK224, 950–750 BP and VK253, 950–850 BP; Russia). The *P. vivax* malaria vector *Anopheles atroparvus* is at present widespread in Europe and nearby regions, including the Pontic Steppe, and our cases indicate this was also true in the past[42]. The single case of *P. falciparum* malaria was found in a sample from Armenia (NEO111; 463-0 cal. BP), where malaria was eliminated in the 1960s[43].

Among DNA viral species, we found widespread infections with HBV (28 cases, 2.1% detection rate), consistent with previous studies[20-22] (Extended Data Fig. 6e). Our newly reported HBV cases include individuals from Mesolithic (Kolyma River, n = 1) and Neolithic (Lake Baikal, n = 3) contexts in Siberia dating back to 9,906–9,665 cal. BP, expanding the spatiotemporal range of ancient HBV into those regions (Extended Data Fig. 7). We also report a putative ancient case (n = 1) of torque teno virus dating back roughly 7,000 years (NEO498, Ukraine; 7,161–6,950 cal. BP). Torque teno virus infects around 80% of the human population today and, although it is not associated with any particular disease, it replicates rapidly in immunocompromised individuals[44]. Other ancient virus hits included viruses not known to infect humans, such as ancient phage DNA (for example, *Escherichia* phage T4, *Proteus* virus Isfahan; Supplementary Table 2) and one putative case of an ancient insect virus (Invertebrate iridescent virus 31 (IIV-31)) in a tooth sample of a Viking Age individual from Sweden (VK30, Varnhem; 950–650 BP)[45]. The virus source is probably exogenous, potentially originating from aDNA of food sources in the tooth remains.

Coinfections with many pathogens can worsen disease progression and outcomes[46] and they were probably an important morbidity factor in ancient human populations. Searching for individuals showing co-occurrence of distinct ancient microbial species, we identified 15 cases of putative coinfections in our dataset (Supplementary Table 2). A striking case was a Viking Age individual from Norway (VK388), in which we replicated previous results of infection with a probably smallpox-causing variola virus[19] and furthermore found evidence of

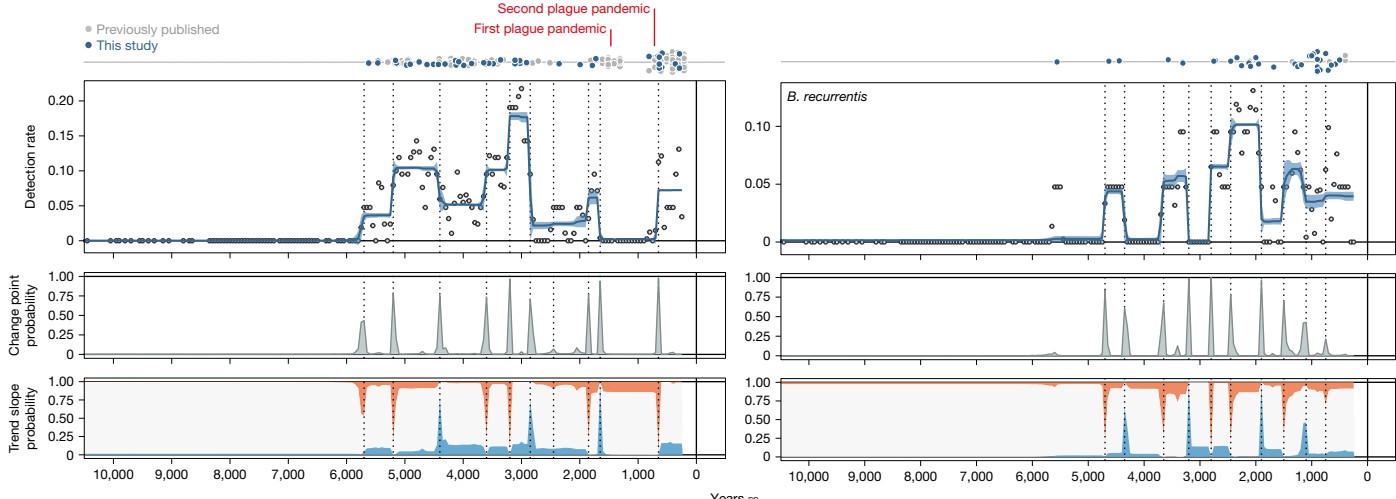

**Fig. 4 | Bayesian time series decomposition of main epidemic pathogens.**
Panels show estimated trendlines and 95% credible interval for detection rates
(top), probability distributions and locations (dotted lines) for change points
(middle) and probability of trend slope (bottom) being positive (red), negative
(blue) or zero (white), inferred using Bayesian change-point detection and time
series decomposition. Tops of panels show temporal distributions of newly
reported pathogen hits (blue circles) as well as previously published ancient
pathogens (grey circles) from the respective species.

infection with the leprosy-causing bacterium *M. leprae*. Another case
of possible coinfection with *M. leprae* was found in VK366, a Viking Age
individual from Denmark, who also showed evidence for leptospirosis
(*L. interrogans*). Among the 15 cases, 6 involved coinfections of HBV
with non-viral pathogens (*Y. pestis* n = 3; *B. recurrentis* n = 2; *P. malariae*
n = 1; Supplementary Table 2). This suggests that some of these cases
involved chronic hepatitis, possibly reflecting HBV infection during
infancy, when hepatitis becomes chronic in 90–95% of modern cases
compared to only 2–6% in adult infections. An intriguing early case
of a possible coinfection was found in a Mesolithic hunter-gatherer
from Russia (Sidelkino, 11,336–11,181 cal. BP). This individual showed
evidence of the respiratory pathogen *C. diphtheriae* and *Helicobac-
ter pylori*, usually restricted to gastric infections; however, rare con-
temporary examples of bacteraemia have been reported for both[47,48].
Overall, our results show that coinfections can be detected using
ancient metagenomic screening but are probably underestimated
given methodological limitations such as differences in pathogen load,
tissue availability and other factors affecting detectability of ancient
microbial DNA.

## Temporal dynamics and drivers of incidence

Understanding the factors affecting the dynamics of past epidemics is
a main aim of paleoepidemiology. Our dataset allows us to address this
question using direct molecular evidence for ancient pathogens across
prehistory. To investigate changes in pathogen incidence over time, we
performed Bayesian change-point detection and time series decom-
position on two pathogens with high detection rates, *Y. pestis* (plague)
and *B. recurrentis* (LBRF), using the detection rate of the respective
pathogen as a proxy for its incidence (Methods). For plague, we inferred
a gradual rise in detection rate starting from roughly 6,000 BP, about
1,000 years after the estimated time to the most recent common ances-
tor of now-known ancient strains (7,100 cal. BP)[33]. It reached a first peak
around 5,000 BP across Europe and the Eurasian Steppe, coinciding
with the emergence and early spread of the LNBA− strains, believed
to have had limited flea-borne transmissibility[15,49] (Fig. 4). Detection
remained high with extra peaks for a roughly 3,000-year period, until
an abrupt change around 2,800 BP led to a roughly 800-year period in
which plague was only detected in one sample (VK522, Oland, Sweden
2,343–2,154 cal. BP). Starting at roughly 2,000 BP, plague reappeared
in three samples from central Asia (DA92, DA101, DA104, Kazakhstan

and Kyrgyzstan; Extended Data Fig. 7 and Supplementary Table 2),
just before the first historically documented plague pandemic (Fig. 4).
Another hiatus of roughly 600 years led to a rise and peak associated
with the second plague pandemic roughly 600 BP (European late
medieval cases, Denmark and previously published cases; Fig. 4). This
pattern of change coincides with the extinction of the LNBA− strains
roughly 2,700 BP (ref. 49) and the second *Y. pestis* diversification event
starting roughly 3,700 BP, which gave rise to an extinct Bronze Age
lineage (RT5, LNBA+)[50] and present-day lineages; these had increased
flea-mediated transmission adaptations favouring bubonic plague and
led to all known later plague pandemics[51]. The adaptations included
acquiring two plasmids: one with the *ymt* gene for survival in the flea
midgut and another with the *pla* gene for invasiveness after trans-
mission[52]. The lack of detection during both periods is also seen in
publicly available ancient *Y. pestis* genomes from other Eurasian sites,
suggesting that sampling bias is unlikely to substantially influence the
observed dynamics.

The inferred temporal dynamics of LBRF show a first peak in detec-
tion around 5,500 BP, slightly more recent than for plague, but with
more sporadic occurrences and sharper peaks during the first roughly
2,000 years (Fig. 4). The geographic extent during the early period
ranges from Scandinavia (NEO29, Denmark, 5,647–5,471 cal. BP) to the
Altai mountains (RISE503, Russia, 3,677–3,461 cal. BP) (Fig. 3 and Sup-
plementary Table 2). From roughly 2,800 BP, LBRF was detected more
consistently, peaking around 2,000 years ago, predominantly in the
Eurasian Steppe region (Fig. 3). This change from epidemic outbreaks to
endemicity overlaps in time with the estimated emergence of a distinct
*B. recurrentis* Iron Age clade[34] (Supplementary Information 5). The
period of high LBRF detection coincided with a time without detectable
plague activity (Fig. 4), reinforcing that the absence of plague is not
due to sample size limitations or poor DNA preservation. This opposing
pattern is unlikely to result from any cross-immunity between *Y. pestis*
and *B. recurrentis* but could plausibly, in part, be caused by population
size decreases and behavioural and societal adjustments during plague
epidemics. LBRF remained detectable until the end of the time series,
particularly in Europe; the continued presence might have facilitated
the emergence of a medieval *B. recurrentis* clade roughly 600 years BP
(ref. 34) (Supplementary Information 5) (Figs. 3 and 4).

A striking feature shared in the temporal dynamics of plague and
LBRF was the absence of detectable cases before roughly 6,000 BP, coin-
ciding with a transition of individuals in predominantly hunter-gatherer

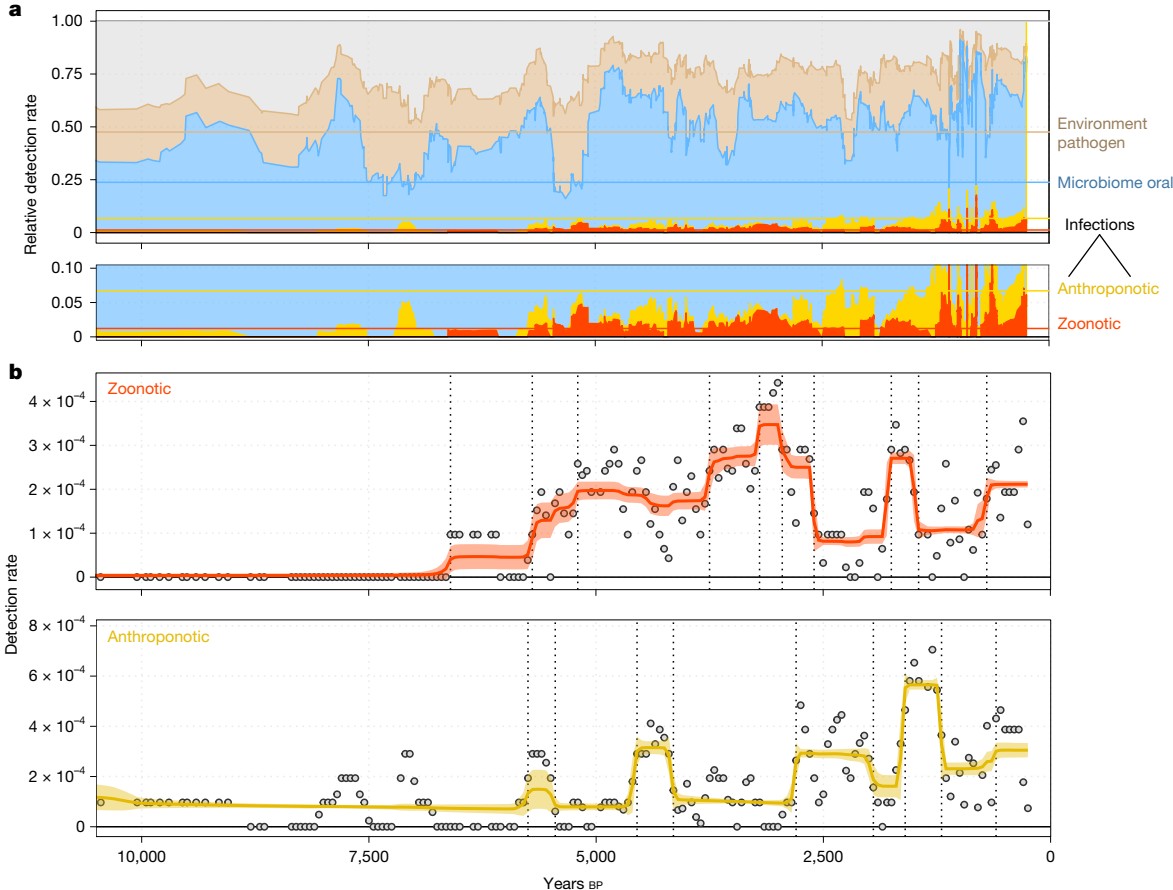

**Fig. 5 | Time series of ancient microorganisms by microbial source.**
**a**, Timeline of relative detection rates in sliding windows of *n* = 21 temporally consecutive samples for different ancient microbial species classes. Coloured horizontal lines indicate the expected rates if species in all classes would be detected at equal rates based on the total number of distinct species in each class. **b**, Trendlines for detection rates inferred using Bayesian change-point detection and time series decomposition for ancient microbial species in the zoonotic (top) and anthroponotic (bottom) reservoir class.

contexts to those in farming or pastoralist cultural contexts. It has been proposed that this transition led to a higher risk of zoonotic disease transmission and facilitated the spread of both old and new pathogens[3]. Our dataset allows us to test this hypothesis using molecular evidence for infectious disease burden. To increase power to detect changes in the load of different pathogen types, we focused on grouped ancient microbial hit categories (Supplementary Table 4).

We found that species associated with the ancient oral microbiome showed the highest relative detection rate, accounting for up to 50% of ancient hits across various periods (Fig. 5a and Extended Data Fig. 9). Species in the 'environmental' classes of probably exogenous origins were also detected at consistent rates throughout time. Species in the 'infection' classes occurred at low detection rates throughout (mostly less than 10%). We found that species in the 'zoonotic' reservoir classes were not detected until around 6,500 BP (Fig. 5a). Using Bayesian time series decomposition, we inferred an overall increase in the detection rates of the zoonotic reservoir classes from roughly 6,000 BP, remaining at elevated levels until the medieval period (Fig. 5b and Extended Data Fig. 9a). Whereas species in the 'anthroponotic' reservoir classes also occur earlier (predominantly species with human-to-human transmission, Extended Data Fig. 10a), we observe increased detection rates from roughly 2,500 BP onwards (Fig. 5b and Extended Data Fig. 9). Our results thus provide direct evidence for an epidemiological transition of increased infectious disease burden after the onset of agriculture through to historical times.

We used Bayesian spatiotemporal modelling to investigate possible drivers of the observed ancient microbial incidences. We modelled the presence or absence of either individual microbial species or combined species groups using sets of putative covariates, including spatiotemporal variables (longitude, latitude and sample age), paleoclimatic variables (mean annual temperature and precipitation), human mobility and ancestry, sample material (tooth or other) and a proxy for 'detectability' (the number of human-classified reads). In the models for the zoonotic or anthroponotic infection species classes, sample age was an important predictor, consistently negatively associated with incidence, and high effect sizes in the individual species models for *B. recurrentis* and *L. interrogans* (Extended Data Fig. 11 and Supplementary Table 6). Longitude was another important factor in the infection classes; it was positively associated with incidence rates for the combined anthroponotic class, and in individual models for *Y. pestis* and *B. recurrentis*. The positive effect of longitude suggests a higher detection rate in the eastern part of our spatiotemporal range, where samples from the Eurasian Steppe predominate.

The increased infection incidence in Steppe populations could reflect an increased genetic susceptibility or a higher risk of acquiring diseases associated with the pastoralist lifestyle. The latter suggestion seems more plausible as continued exposure to selective pressures from certain infectious diseases probably would reduce susceptibility in these populations. Human ancestry showed small but consistent positive effects in some models, particularly the infection classes, for the Caucasus hunter-gatherers. Across all models, the incidence of ancient microorganisms was positively associated with teeth as sample material; the highest effect sizes were found in the oral microbiome and infection classes (Extended Data Fig. 11). Teeth preserved ancient oral microbiome and pathogen DNA better than petrous bones (the source of 86% of our samples), probably because of oral cavity exposure and

better access to microbial DNA in the bloodstream[53]. These results support the notion that species detected in those classes are predominantly of endogenous origin.

## Conclusions

During the Holocene, human lifestyles changed considerably as agriculture, animal husbandry and pastoralism became key practices but the impact on infectious disease incidence is debated. Our study represents a large-scale characterization of ancient pathogens across Eurasia, providing clear evidence that identifiable zoonotic pathogens emerged around 6,500 years ago and were consistently detected after 6,000 years ago. Although zoonotic cases probably existed before 6,500 years ago, the risk and extent of zoonotic transmission probably increased with the widespread adoption of husbandry practices and pastoralism. Today, zoonoses account for more than 60% of newly emerging infectious diseases[54].

We observed some of the highest detection rates at roughly 5,000 BP, a time of substantial demographic changes in Europe due to the migration of Steppe pastoralists and the displacement of earlier populations[4,5]. Steppe pastoralists, through their long-term continuous exposure to animals, probably developed some immunity to certain zoonoses and their dispersals may have carried these diseases westwards and eastwards. Consequently, the genetic upheaval in Europe could have been facilitated by epidemic waves of zoonotic diseases causing population declines, with depopulated areas subsequently being repopulated by opportunistic settlers who intermixed with the remaining original population. This scenario would mirror the population decline of Indigenous people in the Americas following their exposure to diseases introduced by European colonists[55,56]. Our findings support the interpretation of increased pathogen pressure as a likely driver of positive selection on immune genes associated with the risk of multiple sclerosis in Steppe populations roughly 5,000 years ago[57], and immune gene adaptations having occurred predominantly after the onset of the Bronze Age in Europe[9].

Expanding our analyses to the broader pathogen landscape allowed us to infer and contrast incidence patterns between different species and types of pathogens to a greater extent than previously possible. If ancient pathogen DNA of a single species is not detected in a particular region or period, asserting whether this is due to low disease incidence or confounding factors such as differential DNA preservation between different periods and environments is challenging. Our analyses counter these limitations; we demonstrate that pathogens with known epidemic potential and high detection rates, such as *Y. pestis* (plague) and *B. recurrentis* (LBRF), show notable differences in their detection rate over time, suggesting that low detection rates in these cases represent an actual reduction in incidence. During the early period (roughly 5,700–2,700 years ago), the continuous detection of *Y. pestis* is suggestive of endemic disease. The succeeding pattern of distinct waves and periods without detection indicate epidemic outbreaks; these detection peaks match the historically described plague pandemics. This shift from endemic to epidemic is concurrent with important changes in the *Y. pestis* genome, particularly increased flea-transmissibility and pathogenicity[15,50]. The pattern for *B. recurrentis* is almost entirely the opposite, with narrow peaks and long periods without detection, suggesting local epidemics before roughly 2,700 years ago and consistent detection afterwards. This later endemicity of LBRF could be driven by changes in the bacterial genome and by human and environmental factors known to increase the risk of louse infestation[34,58,59]. Experimental studies have demonstrated that *Y. pestis*, like *B. recurrentis*, can infect body lice in the midgut and, sometimes, also the Pawlowsky gland, a putative salivary gland[59]. Body lice infected in the Pawlowsky gland can transmit *Y. pestis* in concentrations sufficient to initiate disease in humans, possibly contributing to transmission during plague outbreaks. Infected body lice have higher mortality than uninfected lice, and it remains unknown whether coinfection of body lice with *Y. pestis* and *B. recurrentis* is possible.

Our study has some important limitations. Whereas ancient shotgun metagenomic data offer direct evidence of past infections, their usefulness depends on having a high pathogen load and the right tissue samples. Our ancient tooth and bone samples are well suited to detect high-load bloodstream infections such as *Y. pestis* and *B. recurrentis*, but pathogens with lower loads or different tissue preferences are underrepresented. Moreover, differentiating ancient infections from those arising from environmental sources, such as the necrobiome, is challenging. Finally, our dataset lacks information on RNA viruses, therefore underestimating the zoonotic disease burden. However, the timing is probably accurate as the conditions favouring zoonotic transmission of RNA viruses are similar to those of other zoonotic pathogens[60].

Our findings demonstrate how the nascent field of genomic paleoepidemiology can create a map of the spatial and temporal distribution of diverse human pathogens over millennia. This map will develop as more ancient specimens are investigated, as will our abilities to match their distribution with genetic, archaeological and environmental data. Our current map shows clear evidence that lifestyle changes in the Holocene led to an epidemiological transition, resulting in a greater burden of zoonotic infectious diseases. This transition profoundly affected human health and history throughout the millennia and continues to do so today.

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

## Methods

### Dataset

We compiled a dataset of aDNA shotgun-sequencing data from 1,313 ancient individuals previously sequenced for studies of human population history (references for previous publications describing laboratory procedures and sample and/or site descriptions in Supplementary Table 1). To facilitate ancient microbial DNA authentication, we excluded sequencing libraries subjected to uracil-DNA glycosylase (UDG) treatment that removes characteristic aDNA damage patterns from further analyses. Samples sequenced across several libraries were combined into single analysis units to maximize sensitivity for detection of ancient microbial DNA present in low abundance.

### Ancient microbial DNA screening

We carried out screening for ancient microbial DNA using a computational workflow combining $k$-mer-based taxonomic classification, read mapping and aDNA authentication. We first performed taxonomic classification of the sequencing reads (minimum read length 30 base pairs (bp)) using KrakenUniq[61], against a comprehensive database of complete bacterial, archaeal, viral, protozoan genomes in the RefSeq database (built with default parameters of $k$-mer size 31 and low-complexity sequences masked). To increase sensitivity for ancient viral DNA, we reran the classification on a viral-specific database of complete viral genomes and neighbour assemblies from RefSeq (https://www.ncbi. nlm.nih.gov/genome/viruses/about/assemblies/), using all reads classified as non-human from the previous run.

Following this initial metagenomic classification, a subset of genera was further processed in the genus-level read mapping and authentication stages. For bacterial pathogens, we selected genera with two or more established species of human pathogens from a recent survey of human bacterial pathogens[2] ($n = 125$ genera). Genera with a single pathogenic species were not included to balance between including genera responsible for substantial human pathogenic burden and computational feasibility. We further included genera including human protozoan pathogens ($n = 11$ genera), as well as all viral genera ($n = 1,356$).

For each genus of interest showing more than or equal to 50 unique $k$-mers assigned, all sequencing reads classified were collected and aligned in parallel against a representative reference assembly for each individual species within the genus. We selected the assembly with the most unique $k$-mers assigned as the representative reference genome for each species in a particular sample. If no reads were assigned to any assembly of the species in KrakenUniq, we selected the first assembly for mapping. Read mapping against the selected assembly was carried out using bowtie2 (ref. 62), using the 'very sensitive' preset and allowing one mismatch in the seed (-N 1 option). Mapped BAM files were subjected to duplicate marking using 'samtools markdup'[63], and filtered for mapping quality MAPQ $\geq$ 20. The aDNA damage rates were estimated using metaDMG[64].

### Authentication of ancient microbial DNA

To authenticate ancient microbial DNA, we calculated sets of summary statistics quantifying expected molecular characteristics of true positive ancient microbial DNA hits[65]:

**Similarity to the reference assembly.** Summary statistics in this category measure how similar sequencing reads are to a particular reference assembly, with true positive hits expected to show higher similarity than false-positive hits. Summary statistics used include the following.
**Average edit distance.** This describes the average number of mismatches in sequencing reads mapped to a particular reference (lower means more similar to the reference).
**ANI.** The ANI is the average number of bases in a mapped sequencing read matching the reference assembly, normalized by the read length (higher means more similar to the reference).

**Number of unique $k$-mers assigned.** The number of unique $k$-mers that are assigned to a particular reference assembly from the KrakenUniq classification (higher means more similar to the reference).

**aDNA characteristics.** Summary statistics in the aDNA characteristics category measure the evidence for sequencing reads deriving from an aDNA source. Summary statistics used include the following.
**Average read length.** This describes the average length in base pairs of sequencing reads mapped to a particular reference (shorter means more likely to be ancient).
**Terminal aDNA substitution rates.** The terminal aDNA substitution rate is the frequency of C>T (G>A) substitutions observed at the 5′ (3′) terminal base across all sequencing reads mapped to a particular reference (higher means more likely to be ancient).
**Bayesian $D_{max}$.** Bayesian $D_{max}$ is an estimator of the aDNA damage rate from metaDMG (higher means more likely to be ancient).
**Bayesian $Z$.** Bayesian $Z$ is an estimator of the significance of evidence for the aDNA damage rate from metaDMG (higher means more likely to be ancient).

**Evenness of genomic coverage.** Summary statistics in this category measure how evenly mapped sequencing reads are distributed across a reference assembly. Summary statistics used include the following.
**Average read depth.** The average read depth is the average number of reads covering a base in the reference assembly.
**Breadth of coverage.** The breadth of coverage describes the fraction of the reference assembly that is covered by one or more sequencing reads.
**Expected breadth of coverage.** The breadth of coverage expected for a particular average read depth is calculated[66] as

$$1 - e^{-(\text{average read depth})}$$

**Ratio of observed to expected breadth of coverage.** This is the ratio of the breadth of coverage observed in the mapping to the breadth of coverage expected given observed average read depth (higher means more even coverage).
**Relative entropy of read start positions.** This relative entropy is a measure for the information content of the genomic positions of mapped reads. To obtain this statistic, we calculate the frequency of read alignments with their start positions falling within windows along the reference assembly, using two different window sizes (100 and 1,000 bp). The obtained frequency vector is converted into Shannon information entropy, and normalized using the maximum entropy attainable if the same total number of reads were evenly distributed across the windows (higher means more even coverage).

### Filtering of putative ancient microbial hits

From this initial screening, we then selected a subset of putative microbial 'hits' (sample–species combinations) for further downstream analysis based on a set of aDNA authentication summary statistics:

- number of mapped reads greater than or equal to 20
- 5′ C>T deamination rate greater than or equal to 0.01
- 3′ G>A deamination rate greater than or equal to 0.01
- ratio of observed to expected breadth of coverage greater than or equal to 0.8
- relative entropy of read start positions greater than or equal to 0.9
- ANI > 0.965
- rank of number of unique $k$-mers assigned less than or equal to 2

For this initial filtered list of putative microbial hits, we ran metaDMG using the full Bayesian inference method to obtain $Z$ scores measuring the strength of evidence for observing aDNA damage.

The final list of putative individual ancient microbial hits was then obtained using the filtering cutoffs

- metaDMG Bayesian $D_{max} \geq 0.05$
- metaDMG Bayesian $Z \geq 1.5$
- rank of number of unique $k$-mers assigned = 1

For authentication of viral species, we used the same filtering cutoffs described above, except for a lower ANI cutoff (greater than 0.95), as well as a lower cutoff for relative entropy of read start positions (greater than 0.7) for short viral genomes (fewer than 10 kilobases).

The result of this filtering is a single best-matching species hit for each sample and genus of interest (Supplementary Table 2). We note that this approach will miss potential cases in which aDNA from many species of the same genus are present in the sample. However, because of the considerable challenges involved in distinguishing this scenario from false positives due to cross-mapping of ancient reads from a single source of DNA to reference assemblies of a closely related species (for example, *Y. pestis*/*Yersinia pseudotuberculosis*), we opted for the conservative option of retaining only the best hit for each genus.

To further authenticate putative hits with low read counts ($n \leq 100$ final reads), we carried out a BLASTn analysis. We extracted the reads for a species hit from the final filtered BAM files and queried them against the 'nt' database (downloaded 18 August 2024) using 'blastn -task blastn -max_hsps 1'. For the reads of each putative ancient microbial hit, we then tabulated the number of times and proportion of the highest scoring BLAST hits matched either the genus or species inferred from our workflow (Supplementary Table 3).

### Simulations of ancient microbial DNA
We simulated aDNA fragments from microbial reference genomes in silico using gargammel[67]. We chose nine species representing pathogens of interest, and for each selected an assembly not present in the pathogen screening workflow database:
- *Brucella melitensis* (GCF_027625455.1)
- *H. pylori* (NZ_CP134396.1)
- *M. tuberculosis* (NZ_CP097110.1)
- *Salmonella enterica* (NZ_CP103966.1)
- *Y. pestis* (NZ_CP064125.2)
- *Y. pseudotuberculosis* (NZ_CP130901.1)
- *P. vivax* (GCA_900093555.2)
- variola virus (GCA_037113635.1)
- human betaherpesvirus 5 (GCA_027927465.1)

For each reference genome, we simulated 5 million single-end sequencing reads (100 bp read length) with adapter sequences, with read length distribution and damage patterns from a mapDamage2 results of a previously published ancient pathogen genome (RISE509, *Y. pestis*[15]). The full-length simulated reads were then adapter-trimmed using AdapterRemoval[68]. To investigate the ability of the workflow to detect low abundance ancient microorganisms, we randomly down-sampled the full read set for each reference genome using seqtk (https://github.com/lh3/seqtk) (50, 100, 200, 500 reads; ten replicates each).

### Topic model analysis
We carried out topic model analysis on taxonomic classification profiles for each sample using the R package fastTopics[69] (https://github.com/stephenslab/fastTopics). We used the number of unique $k$-mers assigned to non-human genera from KrakenUniq as the observed count data for each sample, excluding genera with fewer than 50 unique $k$-mers assigned. The analysis was carried out for $L = 2$ and $L = 3$ topics to capture broad structures in the classification profiles.

### Ancient microbial groups
For combined analyses, we grouped the ancient microbial hits into three categories on the basis of the likely source of the microbial DNA (Supplementary Table 4):

1. Environmental, to capture all hits derived from environmental sources including the necrobiome (labelled environment_background, environment_pathogen, to distinguish potential pathogenic species from non-pathogenic ones)
2. Oral microbiome, including both commensal and pathogenic species (microbiome_oral)
3. Probably pathogenic infections, further distinguished into different modes of transmission (infection_anthroponotic; infection_vector_borne; infection_zoonotic).

We define zoonotic pathogens here as those transmitted from animals to humans or which made such a host jump in our sampling time frame[70].

### Time series
To infer temporal dynamics of ancient microbial species, we calculated detection rates in a sliding window of $k = 21$ temporally consecutive samples across the entire timeline of the 1,266 samples with dating information. For individual species, the detection rate for each window corresponds to the proportion of the 21 samples in each window that were positive for the species of interest. For analyses of species combined in classes, we calculated the detection rate as the ratio of the total number of hits within a class in the window over the total number of possible hits across all species in a window (21 samples × 258 species across all classes). For individual species with $n \geq 20$ hits or combined species classes, we further performed Bayesian change-point detection and time series decomposition (BEAST)[71] implemented in the R package Rbeast (https://github.com/zhaokg/Rbeast), using the detection frequencies described above as response variables. Data for previously published *Y. pestis* and *B. recurrentis* genomes were obtained from AncientMetagenomeDir (release v.23.12, https://doi.org/10.5281/zenodo.10580647)[72].

### Spatiotemporal models of species incidence
To identify possible drivers of the observed spatiotemporal ancient microbial incidence, we combined the individual microbial species and the combined species groups with palaeoclimatic variables, human mobility estimates and kriged estimates of ancestry composition for Holocene West Eurasia. Palaeoclimatic reconstructions were accessed using the CHELSA-Trace21k data, which provide global monthly climatologies for temperature and precipitation at 30-arcsec spatial resolution in 100-year time steps for the past 21,000 years (ref. 73). To pair the microbial species and groups with the palaeoclimatic reconstructions, we took the average climatic value across all the time steps that fall within the microbial species and/or group age ± standard deviation at each of the sampling locations. Palaeoclimatic variables considered were annual mean temperature (BIO01) and annual precipitation (BIO12). Human mobility values were accessed from ref. 74 and roughly represent the distance in kilometres between the burial location of the ancient human individual and its putative ancestral origin on the basis of patterns of genetic similarity derived from multidimensional scaling (MDS) analysis. Microbial species and/or groups were paired with the mobility estimate of the ancient human individual that occurs closest in space and time. Kriged ancestry estimates were extracted from ref. 75, using the spatiotemporal ancestry kriging method from Racimo et al.[76], and paired to the closest spatiotemporal location of the ancient human remain where the corresponding microbial species and/or groups were sampled.

To determine the influence of the covariates on the microbial incidence, we used a hierarchical Bayesian model implemented in the inlabru R package[77,78], in which ancient microbial presence or absence follows a binomial distribution and the spatiotemporal variables (latitude, longitude and sample age), number of human-classified reads, sample material, palaeoclimatic variables, human mobility and human ancestry constitute the linear predictors. The sample material is a

categorical variable indicating whether the material used for sequencing was a tooth or not (bone), which inlabru treats as a random effect variable. We followed the default inlabru priors, in which distributions are distributed as a Gaussian variable with mean $\mu$ and precision $\tau$. The prior on the precision $\tau$ is a Gamma with parameters 1 and 0.00005. The mean is a linear combination of the covariates. By default, the prior on the intercept of the linear combination is a uniform distribution, whereas the priors on the coefficients are Gaussian with zero mean and precision 0.001. All covariates were normalized before the analyses. For each microbial species and group, we tested many models with different sets of covariates: (1) palaeoclimate + mobility + ancestry, (2) palaeoclimate + mobility, (3) palaeoclimate + ancestry, (4) only climate, (5) mobility + ancestry, (6) only mobility, (7) only ancestry and (8) no climate, mobility nor ancestry. Spatiotemporal variables, number of human-classified reads and sample material were included in all models. Because covariates were normalized, results indicate deviations from the mean. The effect size is interpreted in units of standard deviation. We used a criterion (DIC score) that prioritizes both fit and simplicity (low number of effective parameters) for evaluating the performance of hierarchical Bayesian models. Parameter distributions of the full model tended not to differ by much to those of the best-performing models under this score, when those parameters were included in the model. Results for both the full and best-performing models (that is, models with the lowest DIC score for each microbial species or combined species group) are shown in Extended Data Fig. 11. DIC scores as well as Watanabe–Akaike information criterion for all models we tested can be found in the Supplementary Table 6.

## Geographic maps

All maps were created using the R project for statistical computing[79], using the sf package[80]. Shape files for coastlines, rivers and lakes were obtained from Natural Earth (https://www.naturalearthdata.com) using the rnaturalearth package[81]. Elevation data for Fig. 1 were obtained from AWS Open Data Terrain Tiles (https://registry.opendata.aws/terrain-tiles) using the elevatr package[82].

## Reporting summary

Further information on research design is available in the Nature Portfolio Reporting Summary linked to this article.

## Data availability

Data for the 907 individuals for whom sequencing data as trimmed read files (FASTQ) are released in this study have been deposited at the European Nucleotide Archive under accession no. PRJEB65256. Accession numbers for sequencing read data of the remaining individuals are provided in Supplementary Table 1. Processed analysis files including the KrakenUniq database file and metagenomic profiling results, microbial species read alignments (BAM format) as well as per-sample summary tables and plots from the screening pipeline are available at https://doi.org/10.17894/ucph.f0f75211-7bc3-445d-90c0-b72a22ba0930.

## Code availability

A snakemake workflow implementing the computational screening pipeline is available at https://github.com/martinsikora/pathopipe.

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

**Acknowledgements** The Lundbeck Foundation GeoGenetics Centre is supported by the Lundbeck Foundation (grant nos. R302-2018-2155, R155-2013-16338), the Novo Nordisk Foundation (grant no. NNF18SA0035006), the Wellcome Trust (grant no. UNS69906), Carlsberg Foundation (grant no. CF18-0024), the Danish National Research Foundation (grant nos. DNRF94, DNRF174), the University of Copenhagen (KU2016 programme) and Ferring Pharmaceuticals A/S to E.W. Extra support was provided by Germany's Excellence Strategy (EXC-2077), project no. 390741603 'The Ocean Floor – Earth's Uncharted Interface'. We thank A. Razeto and P. Selmer Olsen, for administrative and technical assistance. We thankfully acknowledge Illumina Inc. for collaboration. E.W. thanks St. John's College, Cambridge, for providing a stimulating environment of discussion and learning. This work was further supported by the Swedish Foundation for Humanities and Social Sciences grant (Riksbankens Jubileumsfond grant no. M16-0455:1) to K.K. M.S. was supported by Maritime encounters, Riksbankens Jubileumsfond, grant no. M21-0018. M.E.A. was supported by Marie Skłodowska-Curie Actions of the EU (grant no. 300554), The Villum Foundation (grant no. 10120) and Independent Research Fund Denmark (grant no. 7027-00147B). G.S. is supported by the Marie Skłodowska-Curie Individual Fellowship 'PALAEO-ENEO' (grant agreement number 751349). H.S. was supported by a Carlsberg Semper Ardens grant (no. CF19-0601) and a European Research Council (ERC) Consolidator Grant (grant no. 101045643). F.R. is supported by a Villum Young Investigator Grant (project no. 00025300), a Novo Nordisk Fonden Data Science Ascending Investigator Award (grant no. NNF22OC0076816) and by the ERC under the European Union's Horizon Europe programme (grant agreement nos. 101077592 and 951385). F.V.S. was supported by the Lundbeck Foundation (grant no. R322-2019-2610). N.O., R.Å., L.H. and B.N. are financially supported by Knut and Alice Wallenberg Foundation as part of the National Bioinformatics Infrastructure Sweden at SciLifeLab. A.K.N.I. and L.F. thank the OAK Foundation.

**Author contributions** M.S. and E.W. conceptualized the study. M.S., E.C., A.F.-G., S.H.N., A.K.N.I. and F.V.S analysed data. M.S., E.C., A.F.-G., N.O., R.Å., L.H., E.K.I.-P., B.M., S.H.N. and H.S. were involved in method development and implementation. G.S., M.E.A., F.V.S., H.S., C.G., J.S. and L.V. were involved in data generation. M.S., M.E.A., K.-G.S. and K.K. curated bioarchaeological data. M.S. T.C.J., B.N., J.P., L.F., F.R. and E.W. supervised the research. M.S., A.K.N.I. and E.W. wrote the first draft of the paper. M.S., A.K.N.I., L.F., J.P. and E.W. were involved in reviewing drafts and editing. M.S., K.-G.S. and A.K.N.I. wrote the Supplementary Information. All co-authors read, commented on and agreed on the submitted manuscript.

**Competing interests** The authors declare no competing interests.

## Additional information

**Correspondence and requests for materials** should be addressed to Martin Sikora, Astrid K. N. Iversen or Eske Willerslev.

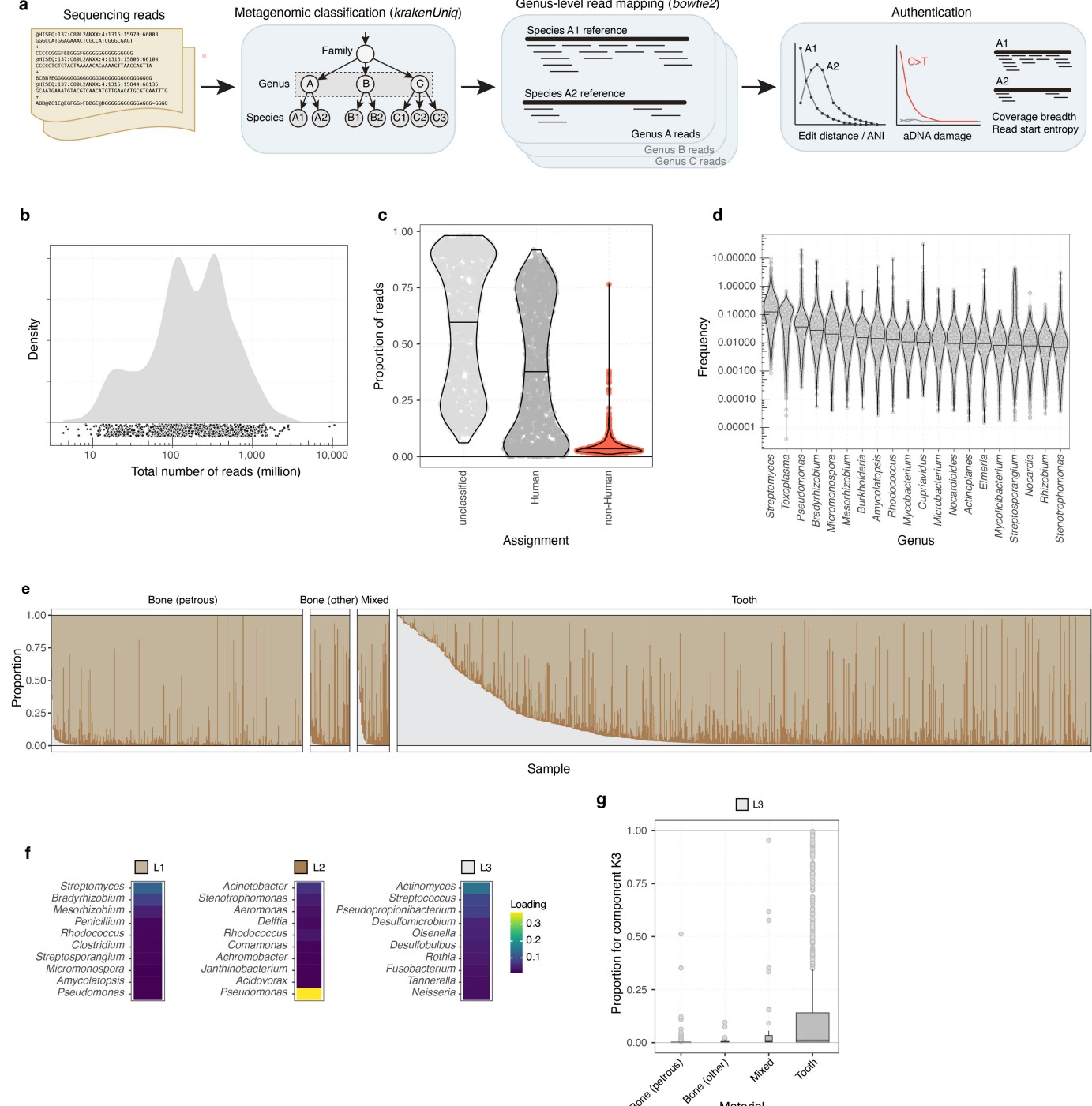

**Extended Data Fig. 1 | Workflow overview and metagenome composition.**
a, Workflow overview. b, Distribution of total number of sequencing reads screened across the n = 1,313 study samples. c, Violin plots showing distributions of proportions of reads classified as human, non-human or not classified for the study samples. Median values for each genus are indicated by horizontal lines. d, Violin plots showing fraction of reads classified on the taxonomic level of genus, for the top 20 most abundant genera. e, Barplots showing inferred proportions for L = 3 topics (indicated by fill colour) from topic model analysis for n = 1,272 study samples with sample material information. f, Factor loadings for the 10 highest loading genera for each of the L = 3 topics from the topic model analysis. g, Boxplots showing distributions of proportions for topic K3 (associated with oral microbiome taxa) in different sample materials.

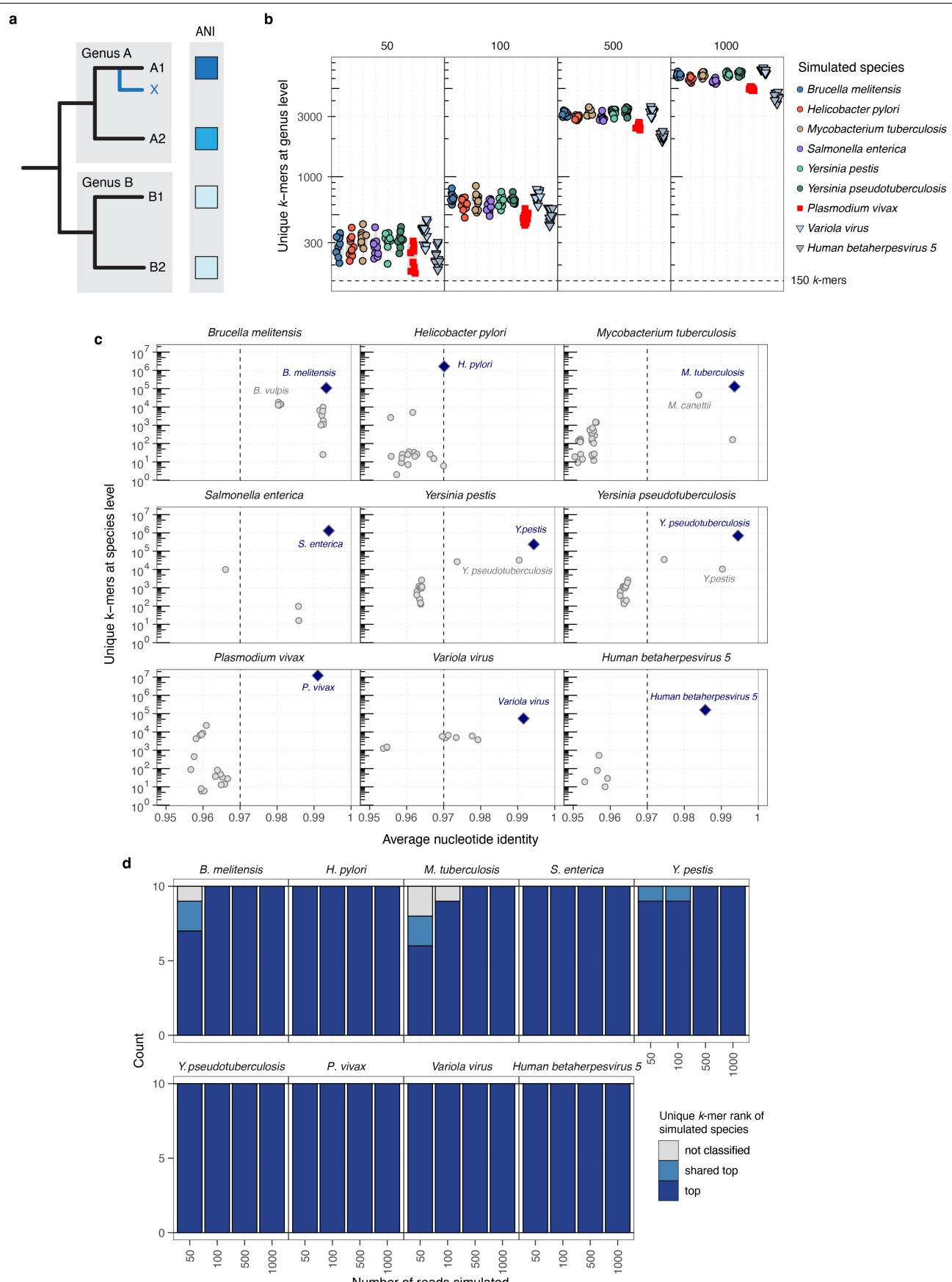

**Extended Data Fig. 2 |** See next page for caption.

**Extended Data Fig. 2 | Reference genome similarity in simulated ancient microbial data.** a, Illustration showing phylogenetic context and expected average nucleotide identity (ANI) for a hypothetical sampled microbial species X and four genomes (A1, A2; B1, B2) of two genera (A, B) present in the reference database. b, Number of unique $k$-mers classified at the level of genus using KrakenUniq for replicates of different read numbers across all simulated species. Dashed line indicates cutoff used in analysis of real data (150 unique $k$-mers). c, Number of unique $k$-mers classified at the level of species as a function of average nucleotide identity for mappings against all individual species reference genomes in the genus of reads simulated for a particular species. Blue diamonds indicate results for the mapping against a reference genome from the same species as the simulated read data, whereas grey circles indicate reference genomes of other species. Selected individual species results are highlighted by species name. Dashed line indicates ANI ≥ 0.97 cutoff value. d, Barplots showing number of replicates where the true positive species reference genome was highest ranking in numbers of unique $k$-mers classified at level of species.

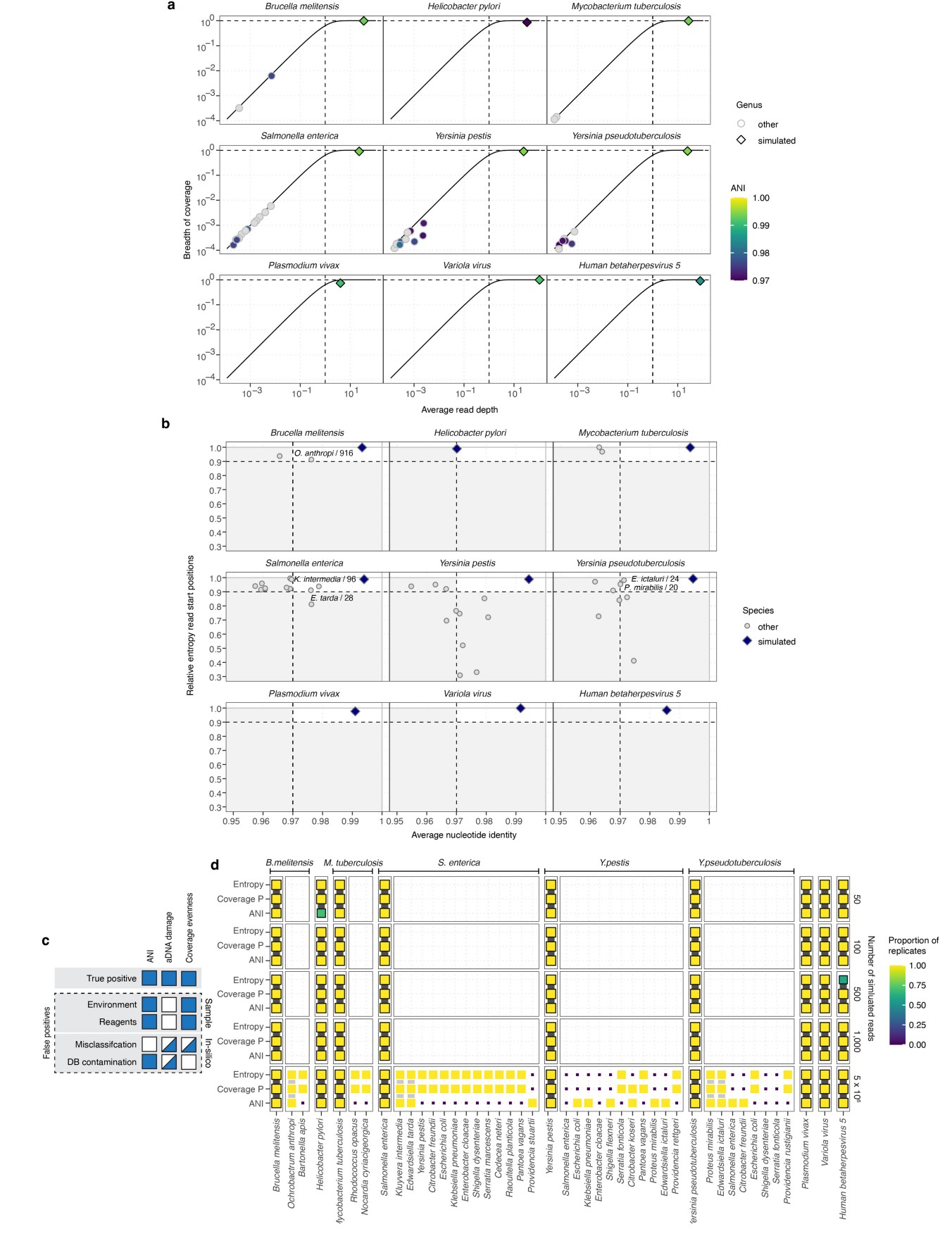

**Extended Data Fig. 3** | See next page for caption.

**Extended Data Fig. 3 | Read mappings across genera in simulated ancient microbial data.** a, Observed breadth of genomic coverage as a function of average read depth for distinct species hits (i.e., mappings with highest number of unique *k*-mers at species level for a genus; n ≥ 20 reads mapped). Each panel shows results for reads simulated from species indicated. Results for mappings against the simulated species are indicated by diamond shape, whereas mappings against species from other genera are indicated with circles. Symbol fill colour indicates average nucleotide identity for mapped reads (grey symbols ANI < 0.97). Solid black line shows theoretical expected breadth of coverage for a given average read depth. Vertical dashed line indicates 1X average read depth. b, Relative entropy statistic (1000 bp window size) as a function of average nucleotide identity. Blue diamonds indicate results for the mapping against reference genome from the same species as the simulated read data, whereas grey circles indicate reference genomes for species hits in other genera. Dashed lines indicate cutoffs used in analyses of real data (ANI ≥ 0.97, entropy ≥ 0.9). False positive hits of reads mapped to a reference genome from a different genome passing cutoffs and their final number of mapped reads (out of 5 million total simulated reads) are labelled. c, Illustration showing potential sources of false positive hits and expected results for authentication summary statistics. d, Matrix plot showing all microbial hits with n ≥ 20 reads mapped and their authentication statistics, for all simulated species and read numbers. Symbol colour and size indicates the number of replicates passing the cutoff for each of three summary statistics shown (ANI ≥ 0.97, ratio of observed/expected coverage breadth ≥ 0.8, entropy ≥ 0.9). Hits passing cutoffs for all three statistics are indicated with coloured outline and background lines (black - true positives; grey - cross-genus false positive mappings).

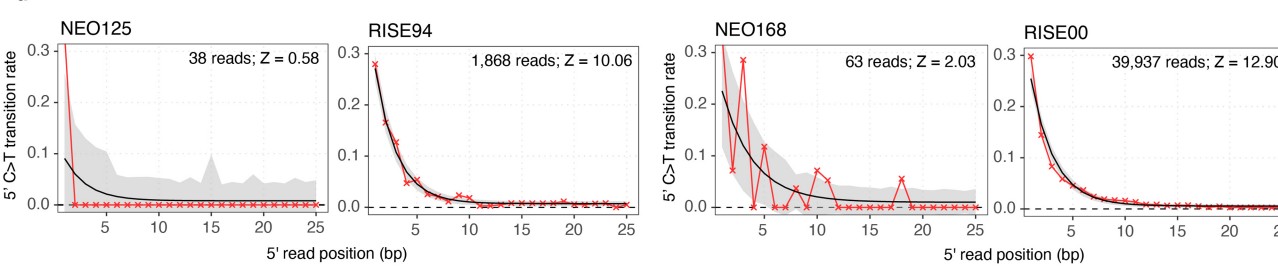

**Extended Data Fig. 4 | Examples of authentication for microbial hits.**
a, Observed breadth of genomic coverage as a function of average read depth.
Coloured symbols indicate hits in species *Toxoplasma gondii* (left panel) and
*Yersinia pestis* (right panel), with symbol colour indicating relative entropy of
read start positions. Solid black line shows theoretical expected breadth of
coverage for a given average read depth[66]. b, Lengths of contigs in the reference
genome of *Toxoplasma gondii* and number of samples showing n ≥ 20 reads
mapped. Symbol colour indicates the average number of reads mapped to a
specific contig across samples. c, Bayesian estimator of aDNA damage (D max)

and significance (Z-score) obtained from metaDMG, for hits in species
*Clostridium botulinum* (left) and *Yersinia pestis* (right). Error bars indicate
±1 standard deviation, and symbol fill colour indicates average read depth for
mapped reads. Samples used as examples in aDNA damage curves (d) are labelled
and indicated with black circles. d, aDNA damage patterns for four example hits
in species *Clostridium botulinum* and *Yersinia pestis*. Plots show observed
nucleotide misincorporation frequencies (red symbols and line) and metaDMG
fit (black line) and 68% credible intervals (shaded region) for C > T transitions as
a function of distance from the 5′ read end.

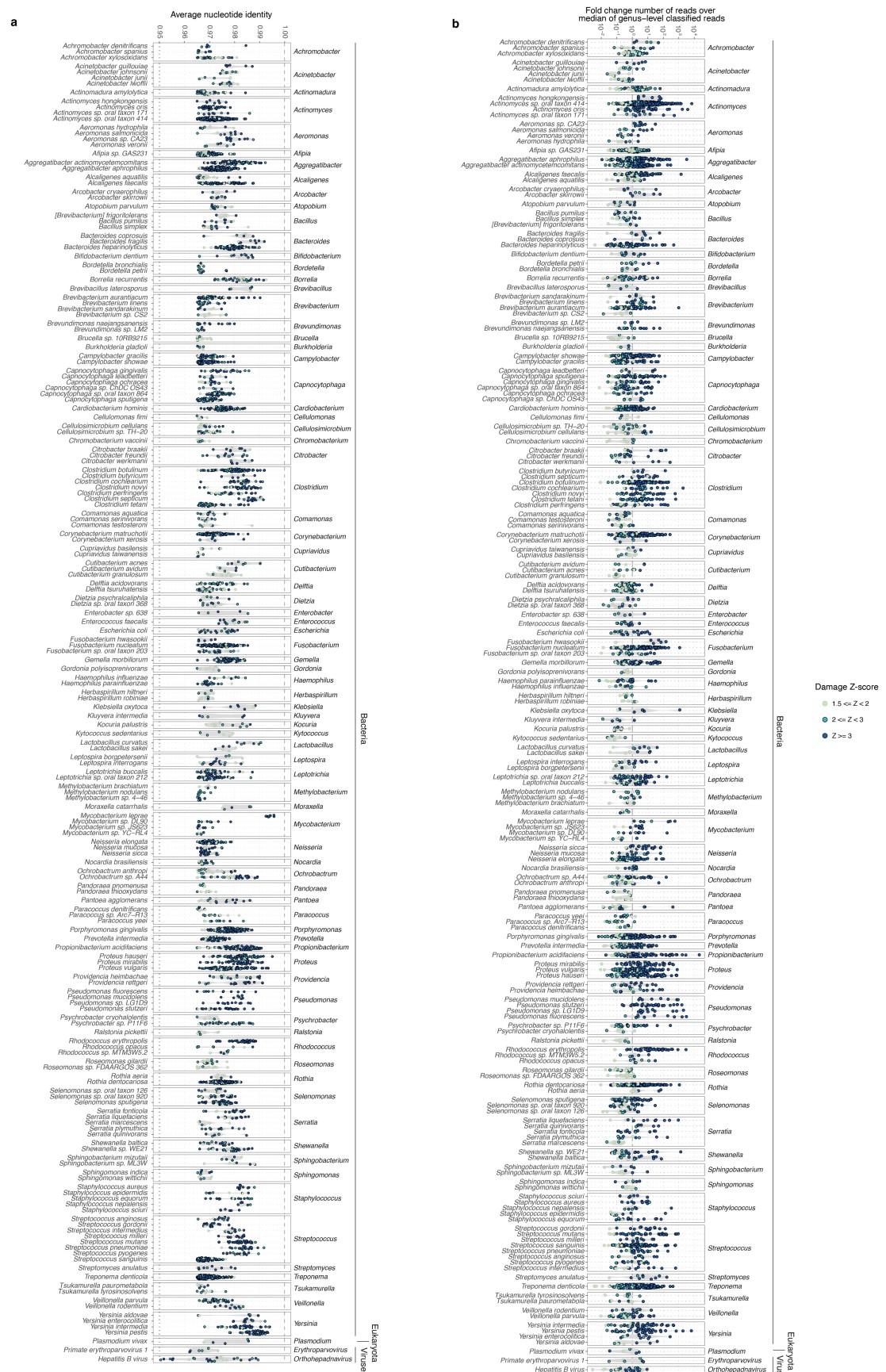

**Extended Data Fig. 5 | Ancient microbial hit ANI and read recruitment.** a, b, Distributions of ANI (a) and log10-fold change of mapped reads over median of reads classified at taxonomic rank of genus per sample (b) for individual species hits detected in n ≥ 5 samples. Symbol colour indicates species hit category.

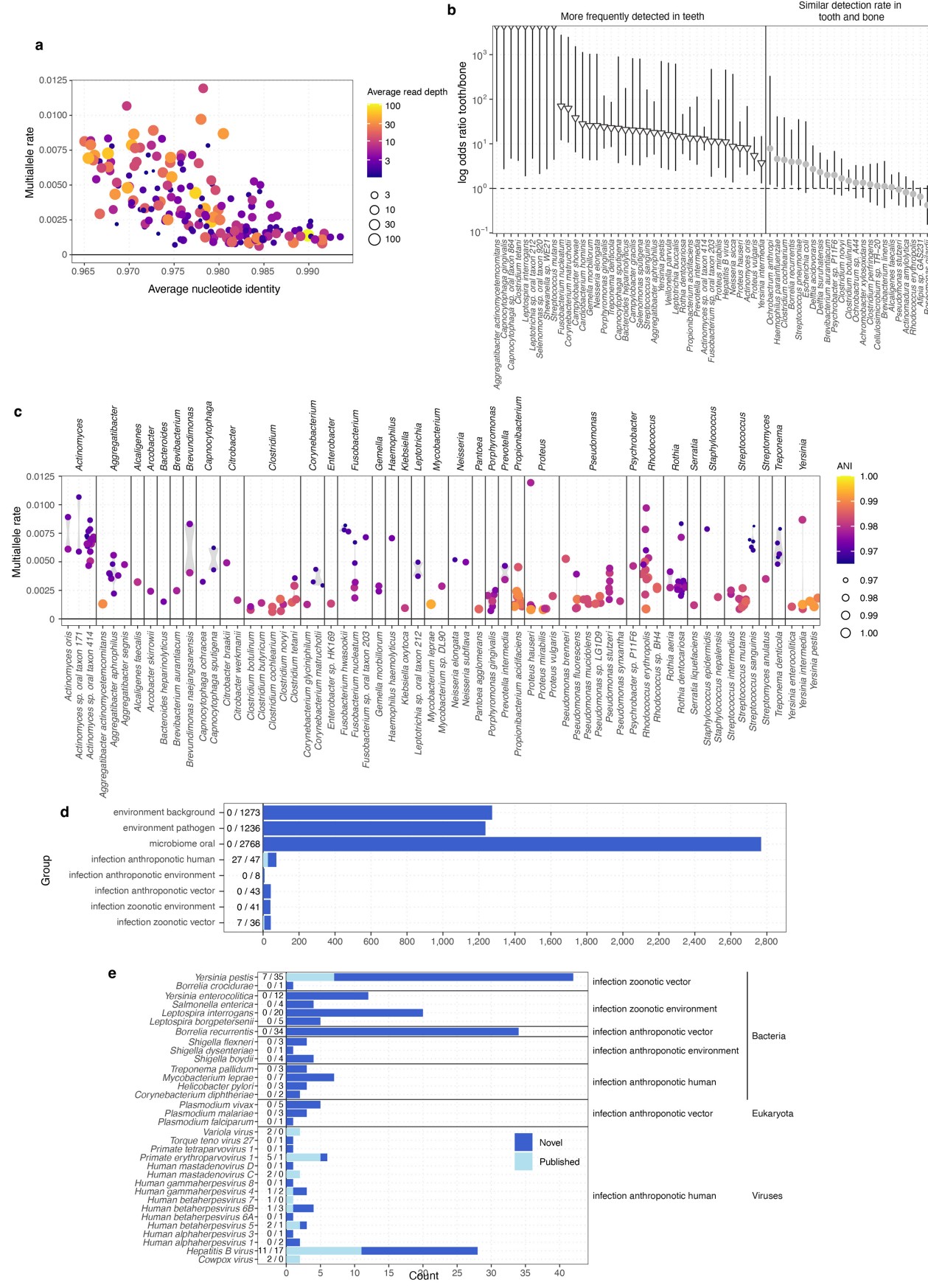

**Extended Data Fig. 6** | See next page for caption.

**Extended Data Fig. 6 | Ancient microbial hit characteristics.** a, Odds ratios for association of ancient hits with sample material (tooth or bone) across n = 61 species with ≥ 20 ancient hits. Symbols indicate significance of association (p ≤ 0.01, Fisher's exact test; white triangles - more frequently identified in tooth; grey circles - no significant association). Error bars indicate 95% confidence interval of odds ratio b, c, Rates of observing multiple alleles in 2 randomly sampled sequencing reads at genomic sites in n = 190 ancient hits (average read depth ≥ 1X) across n = 120 samples. b, Multi-allele rate as a function of ANI. Symbol colour indicates average read depth. c, Distribution of multi-allele rate across species hits. Symbol colour indicates ANI. d, e Barplots showing number of hits identified in each microbial species group (d) or each species within groups of likely infections (e). Novel and previously reported ancient pathogen hits are distinguished by bar colour, with total number in each category labelled.

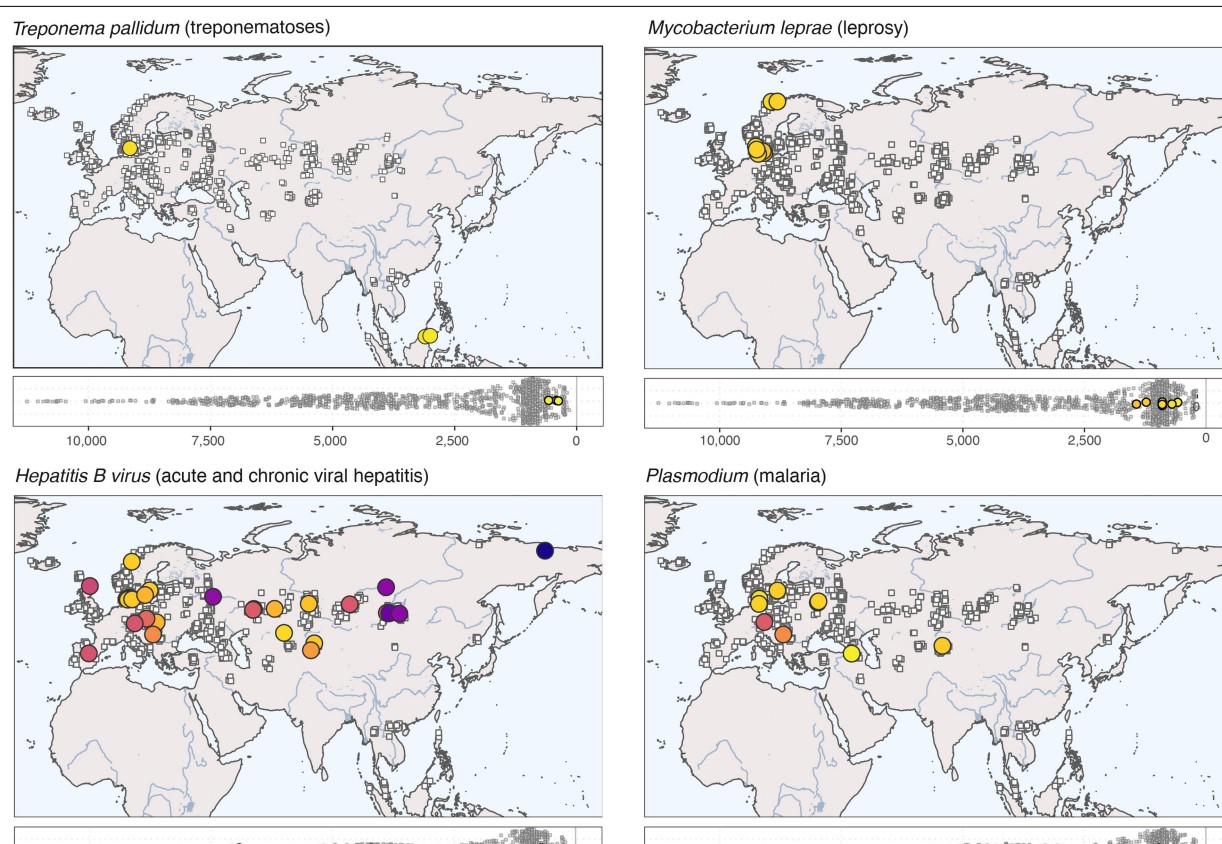

**Extended Data Fig. 7 | Spatiotemporal distribution of selected ancient pathogens.** Each panel shows geographic distribution (top) and timeline (bottom) for identified cases of the respective pathogen (indicated by coloured circle). Geographic locations and age distributions of all n = 1,313 study samples are shown in each panel using white squares. The panel for *Plasmodium* combines the three species detected (*P. vivax* n = 5; *P. malariae* n = 3; *P. falciparum* n = 1).

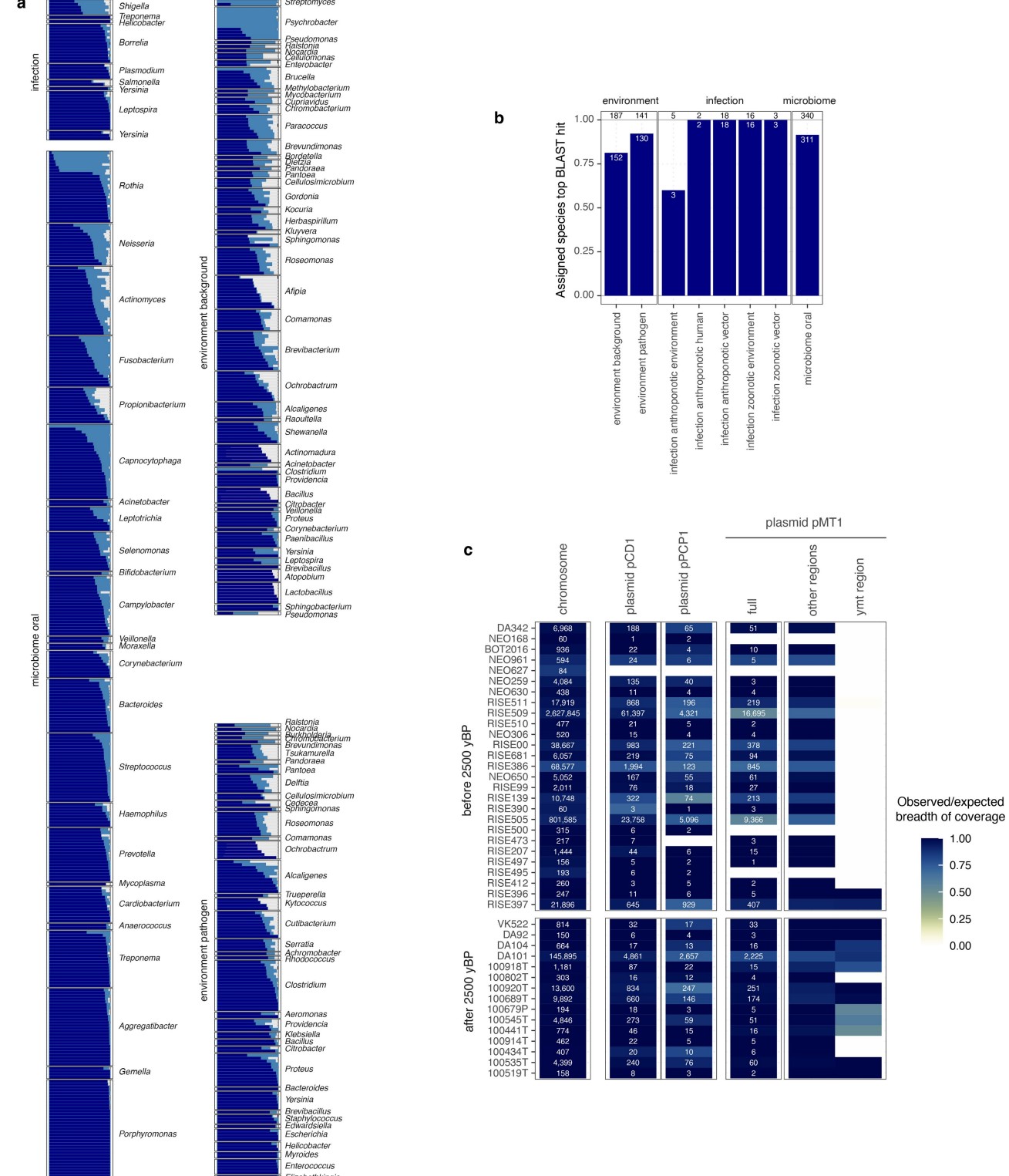

**Extended Data Fig. 8 | Additional ancient microbial hit authentication.**
a, Bar plots showing proportion of reads assigned to same species (dark blue) or genus (light blue) using BLASTn for all hits with n ≤ 100 final reads (n = 712), stratified by genus and microbial source groups b, Bar plots showing the proportion of ancient microbial hits with n ≤ 100 final reads matching the species with most reads assigned using BLASTn, stratified by microbial source group.

c, Heatmap showing number of reads mapped to *Yersinia pestis* CO92 chromosome and plasmids, for n = 42 *Yersinia pestis* hits. Cell color indicates ratio of observed over expected breadth of coverage. Results for plasmid pMT1 are shown for full plasmid, as well as separately for the 19 kb region containing the *ymt* gene absent in the LNBA- strains. Samples are ordered by decreasing age from top to bottom.

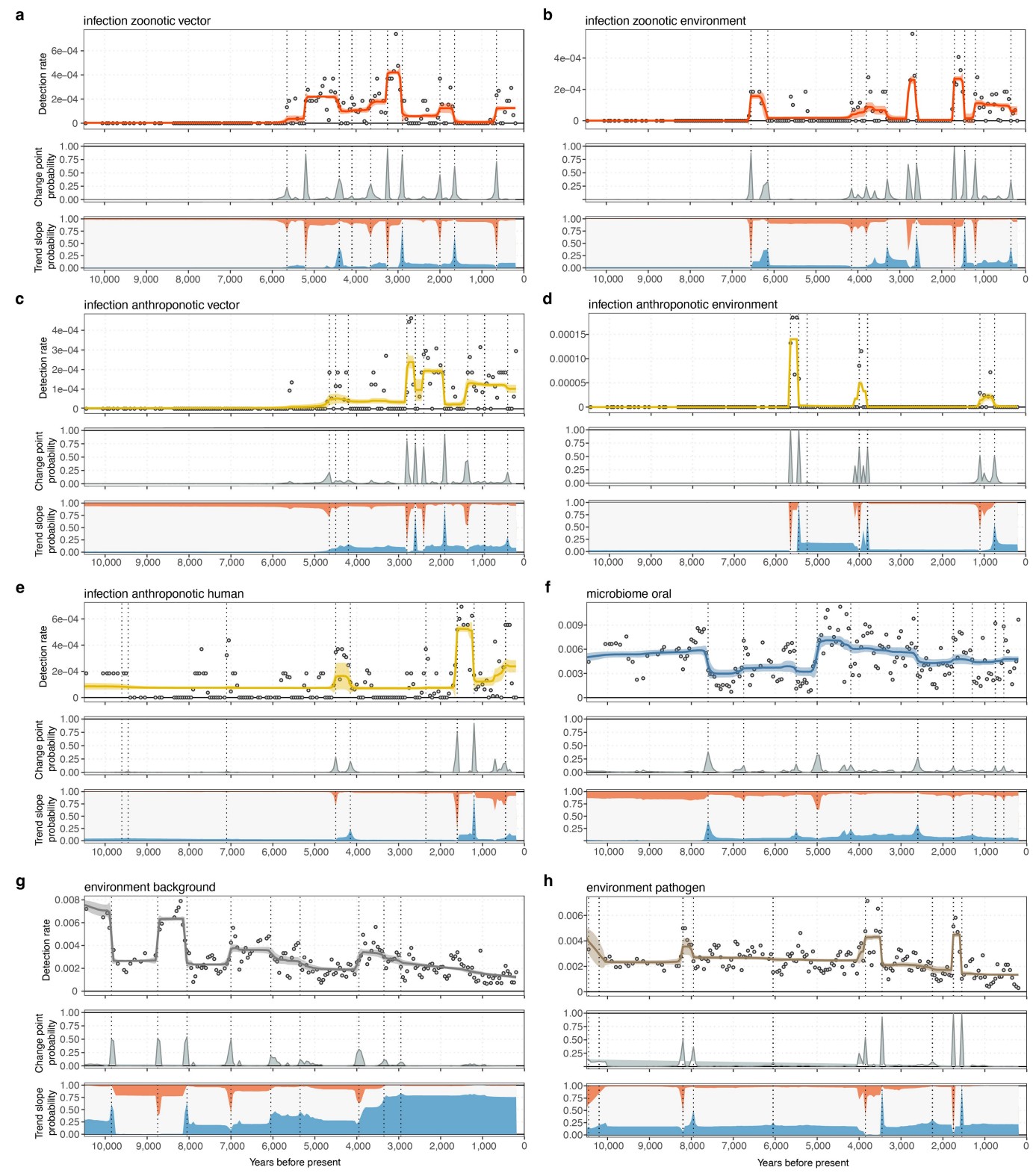

**Extended Data Fig. 9 | Time series of detection rates for ancient microbial groups.** a-h, Panels show estimated trendlines and 95% credible interval for detection rates (top), probability distributions and locations (dotted lines) for change points (middle) and probability of trend slope (bottom) being positive (red), negative (blue) or zero (white), inferred using Bayesian change-point detection and time series decomposition.

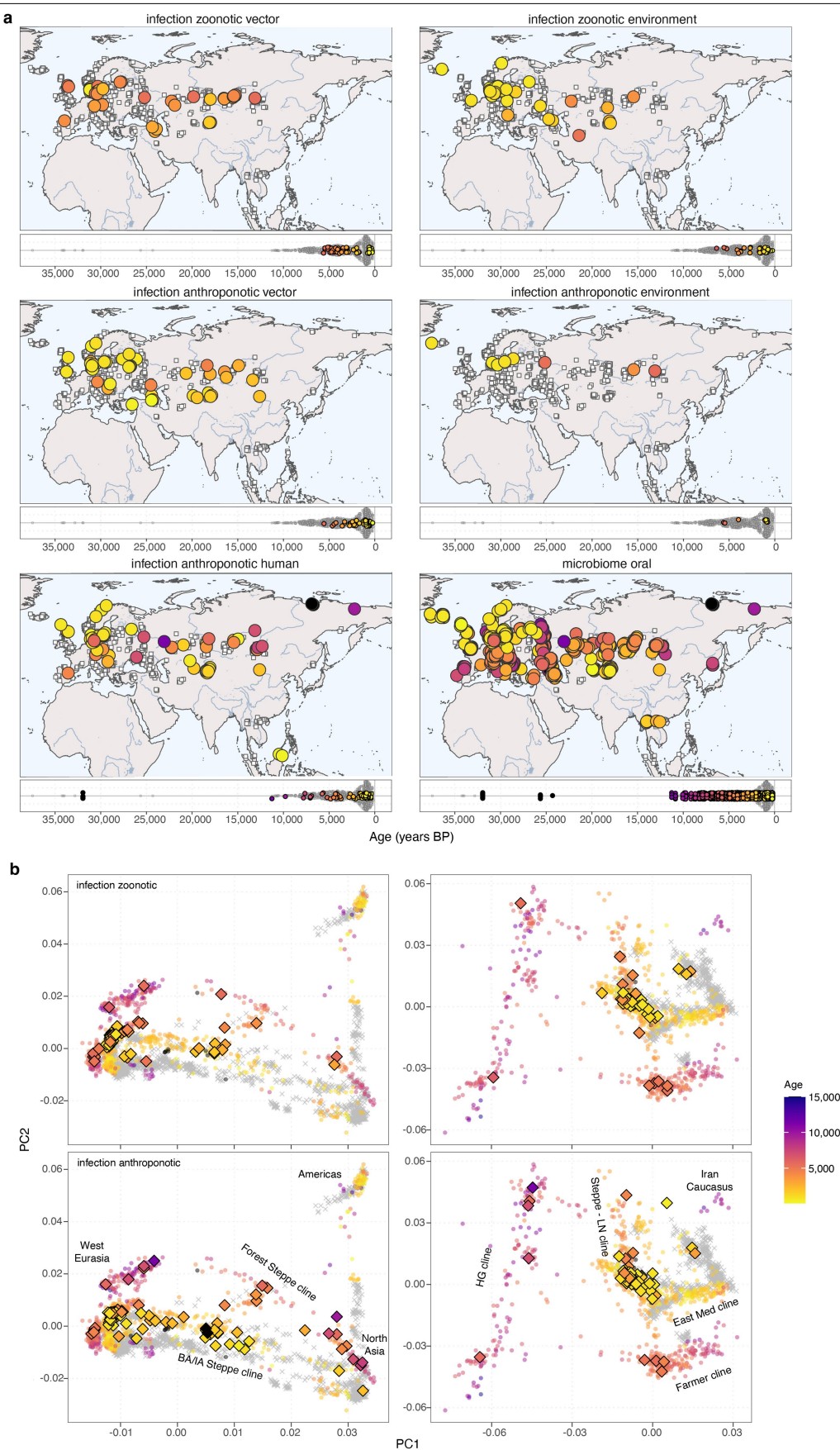

**Extended Data Fig. 10** | See next page for caption.

**Extended Data Fig. 10 | Spatiotemporal distribution and host genetic structure for ancient microbial groups.** a, Panels showing geographic distributions (top) and timelines (bottom) for identified cases of ancient microbial hits in the oral microbiome and infection groups classes (indicated by coloured circle). Geographic locations and age distributions of all 1,313 study samples are shown in each panel using white squares. b, Principal component analyses showing ancient and modern human genetic population structure in non-African (left panels) and west Eurasian (right panels) individuals. Grey crosses indicate present-day individuals, whereas coloured symbols indicate ancient individuals (coloured by sample age). Diamonds with black outlines indicate position in PCA space for samples with hits in combined infection groups. Major clines of known ancient and modern human ancestry groups are indicated with labels.

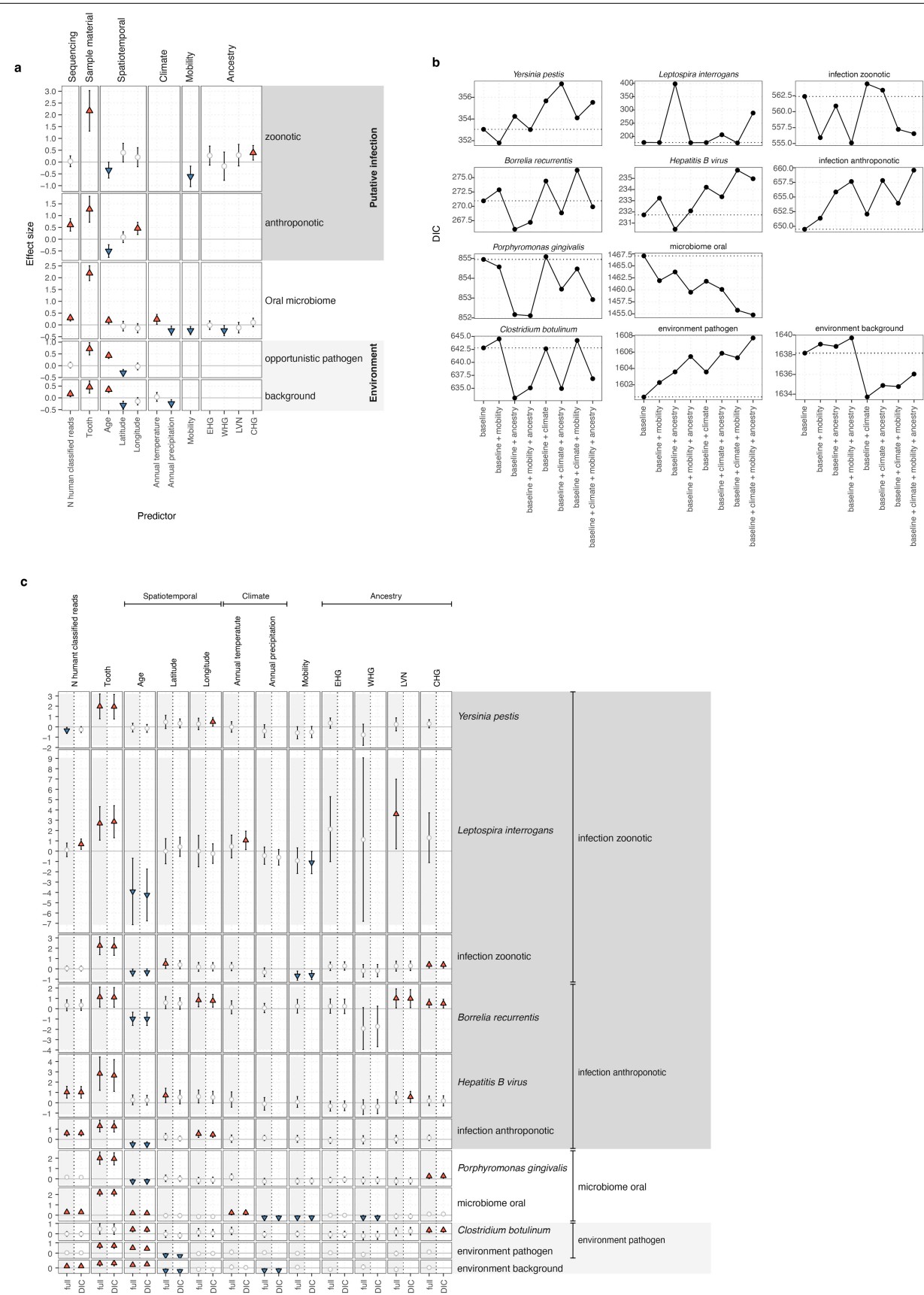

**Extended Data Fig. 11 |** See next page for caption.

**Extended Data Fig. 11 | Predictors of ancient microbial species incidence.** a, Matrix showing effect sizes and of n = 12 potential predictors (columns) for presence of selected combined ancient microbial species groups inferred from spatiotemporal modelling. Predictors with positive effect (defined here as 2.5% and 97.5% posterior quantiles both positive) are shown as red triangles, whereas predictors with negative effect (defined here as 2.5% and 97.5% posterior quantiles both negative) are shown as blue inverted triangles. Predictors where the 95% posterior quantile range spans zero are indicated using white circles. Posterior 95% quantiles are indicated by the error bars. b, Deviance information criterion values for each model and response variable. c, Matrix as in (a), showing effect sizes and of n = 12 potential predictors (columns) for presence of combined ancient microbial species groups and individual species from spatiotemporal modelling. For each class, estimates for the full model (gray background) and the model with lowest deviance information criterion (white background) is shown. Symbols and colors as in (a).

Martin Sikora
Astrid K. N. Iversen
Eske Willerslev

# Reporting Summary

## Statistics

For all statistical analyses, confirm that the following items are present in the figure legend, table legend, main text, or Methods section.

| n/a | Confirmed | |
|---|---|---|
| ☐ | ☒ | The exact sample size (*n*) for each experimental group/condition, given as a discrete number and unit of measurement |
| ☒ | ☐ | A statement on whether measurements were taken from distinct samples or whether the same sample was measured repeatedly |
| ☐ | ☒ | The statistical test(s) used AND whether they are one- or two-sided<br>*Only common tests should be described solely by name; describe more complex techniques in the Methods section.* |
| ☐ | ☒ | A description of all covariates tested |
| ☐ | ☐ | A description of any assumptions or corrections, such as tests of normality and adjustment for multiple comparisons |
| ☐ | ☒ | A full description of the statistical parameters including central tendency (e.g. means) or other basic estimates (e.g. regression coefficient) AND variation (e.g. standard deviation) or associated estimates of uncertainty (e.g. confidence intervals) |
| ☒ | ☐ | For null hypothesis testing, the test statistic (e.g. *F*, *t*, *r*) with confidence intervals, effect sizes, degrees of freedom and *P* value noted<br>*Give P values as exact values whenever suitable.* |
| ☒ | ☐ | For Bayesian analysis, information on the choice of priors and Markov chain Monte Carlo settings |
| ☒ | ☐ | For hierarchical and complex designs, identification of the appropriate level for tests and full reporting of outcomes |
| ☒ | ☐ | Estimates of effect sizes (e.g. Cohen's *d*, Pearson's *r*), indicating how they were calculated |

*Our web collection on statistics for biologists contains articles on many of the points above.*

## Software and code

Policy information about availability of computer code

| Data collection | No specific software was used for data collection |
|---|---|

| Data analysis | krakenuniq (1.0.4)<br>mawk (1.3.4)<br>seqtk (1.3-r106)<br>seqkit (2.3.0)<br>bowtie2 (2.5.2)<br>samtools (1.17)<br>picard (2.27.5)<br>bedtools (2.30.0)<br>datamash (1.5)<br>metaDMG (0.2-41-gc867207)<br>snakemake (7.20.0)<br>gargammel (1.1.4)<br>R (4.2.2)<br>R package fastTopics (0.6-142)<br>R package Rbeast (0.9.7)<br>R package tidyverse (1.3.2)<br>R package inlabru (2.8.0)<br>R package Rsamtools (2.14.0) |
|---|---|

For manuscripts utilizing custom algorithms or software that are central to the research but not yet described in published literature, software must be made available to editors and reviewers. We strongly encourage code deposition in a community repository (e.g. GitHub). See the Nature Portfolio guidelines for submitting code & software for further information.

## Data

Policy information about availability of data

All manuscripts must include a data availability statement. This statement should provide the following information, where applicable:
- Accession codes, unique identifiers, or web links for publicly available datasets
- A description of any restrictions on data availability
- For clinical datasets or third party data, please ensure that the statement adheres to our policy

Data for the 907 individuals where sequencing data as trimmed read files (FASTQ) is released in this study has been deposited at the European Nucleotide Archive under accession PRJEB65256. Accession numbers for sequencing read data of the remaining individuals are provided in Supplementary table 1. Processed analysis files including KrakenUniq database file and metagenomic profiling results, microbial species read alignments (BAM format) as well as per-sample summary tables and plots from screening pipeline are available at https://sid.erda.dk/sharelink/DrwYQeSdLJ.

## Research involving human participants, their data, or biological material

Policy information about studies with human participants or human data. See also policy information about sex, gender (identity/presentation), and sexual orientation and race, ethnicity and racism.

| Reporting on sex and gender | *Use the terms sex (biological attribute) and gender (shaped by social and cultural circumstances) carefully in order to avoid confusing both terms. Indicate if findings apply to only one sex or gender; describe whether sex and gender were considered in study design; whether sex and/or gender was determined based on self-reporting or assigned and methods used.*<br>*Provide in the source data disaggregated sex and gender data, where this information has been collected, and if consent has been obtained for sharing of individual-level data; provide overall numbers in this Reporting Summary. Please state if this information has not been collected.*<br>*Report sex- and gender-based analyses where performed, justify reasons for lack of sex- and gender-based analysis.* |
|---|---|
| Reporting on race, ethnicity, or other socially relevant groupings | *Please specify the socially constructed or socially relevant categorization variable(s) used in your manuscript and explain why they were used. Please note that such variables should not be used as proxies for other socially constructed/relevant variables (for example, race or ethnicity should not be used as a proxy for socioeconomic status).*<br>*Provide clear definitions of the relevant terms used, how they were provided (by the participants/respondents, the researchers, or third parties), and the method(s) used to classify people into the different categories (e.g. self-report, census or administrative data, social media data, etc.)*<br>*Please provide details about how you controlled for confounding variables in your analyses.* |
| Population characteristics | *Describe the covariate-relevant population characteristics of the human research participants (e.g. age, genotypic information, past and current diagnosis and treatment categories). If you filled out the behavioural & social sciences study design questions and have nothing to add here, write "See above."* |
| Recruitment | *Describe how participants were recruited. Outline any potential self-selection bias or other biases that may be present and how these are likely to impact results.* |
| Ethics oversight | *Identify the organization(s) that approved the study protocol.* |

Note that full information on the approval of the study protocol must also be provided in the manuscript.

# Field-specific reporting

Please select the one below that is the best fit for your research. If you are not sure, read the appropriate sections before making your selection.

☒ Life sciences ☐ Behavioural & social sciences ☐ Ecological, evolutionary & environmental sciences

For a reference copy of the document with all sections, see nature.com/documents/nr-reporting-summary-flat.pdf

# Life sciences study design

All studies must disclose on these points even when the disclosure is negative.

| | |
|---|---|
| Sample size | No statistical methods were used to predetermine sample size. Samples were chosen based on previously being analysed for ancient human DNA |
| Data exclusions | No data were excluded from analysis |
| Replication | Replication is not applicable due to studying ancient individuals |
| Randomization | Randomization is not applicable due to studying ancient individuals |
| Blinding | Blinding is not applicable due to studying ancient individuals |

# Reporting for specific materials, systems and methods

We require information from authors about some types of materials, experimental systems and methods used in many studies. Here, indicate whether each material, system or method listed is relevant to your study. If you are not sure if a list item applies to your research, read the appropriate section before selecting a response.

### Materials & experimental systems

| n/a | Involved in the study |
|---|---|
| ☒ | ☐ Antibodies |
| ☒ | ☐ Eukaryotic cell lines |
| ☐ | ☒ Palaeontology and archaeology |
| ☒ | ☐ Animals and other organisms |
| ☒ | ☐ Clinical data |
| ☒ | ☐ Dual use research of concern |
| ☒ | ☐ Plants |

### Methods

| n/a | Involved in the study |
|---|---|
| ☒ | ☐ ChIP-seq |
| ☒ | ☐ Flow cytometry |
| ☒ | ☐ MRI-based neuroimaging |

## Palaeontology and Archaeology

| | |
|---|---|
| Specimen provenance | This study re-analyses ancient metagenomic data from previously published archaeogenetics studies |
| Specimen deposition | This study re-analyses ancient metagenomic data from previously published archaeogenetics studies |
| Dating methods | No new dates are provided |

☐ Tick this box to confirm that the raw and calibrated dates are available in the paper or in Supplementary Information.

| | |
|---|---|
| Ethics oversight | No ethical oversight was required as the study re-analyses ancient metagenomic data |

Note that full information on the approval of the study protocol must also be provided in the manuscript.

