## [Peer Review file · Nature]

The spatiotemporal distribution of human pathogens in ancient Eurasia

Corresponding Author: Dr Martin Sikora

Version 0:

Reviewer comments:

Referee #1

(Remarks to the Author)

"The landscape of ancient human pathogens in Eurasia from the Stone Age to historical times" screens shotgun sequencing data from 1313 ancient humans for pathogens and presents the results, along with interpretation of what the broad patterns mean for the history of infectious disease. It is the largest sample yet presented which screens for ancient pathogen DNA, and it breaks methodological ground by adapting workflows to permit sequencing and analysis at a large scale. While the findings are overinterpreted in some places, and some issues need addressing (notably the interpretation of "zoonotic pathogens" and the need for much more contextual information in places), the findings are impressive and indeed of major significance. Certain specific results are completely novel - such as finding the prominence of louse-borne relapsing fever. Despite reservations/suggestions expressed below, this paper makes significant substantive and methodological contributions.

36 "the origins and past dynamics of human pathogens remain poorly understood." This is vague and understates the contributions of the last ten years from archaeogenetics, not to mention the last generation or two of skeletal anthropology and work by historians. Moreover, with very little focus on phylogenetics, this paper says almost nothing about the origins of human pathogens.

36 "the first spatiotemporal map of diverse ancient human microorganisms and parasites." This is certainly not the first such map. A google search for "atlas infectious disease" will turn up examples. Arguably, this is the first archaeogenetic-based spatiotemporal map, though it is really a large-scale screening for bacterial pathogen DNA (plus a few DNA viruses and eukaryotic parasites) for (mostly) temperate Eurasia up to some date in the last few hundred years.

50 "zoonotic pathogens, which are transmitted from living animals to humans or which have made a host jump into humans from animals in the timeframe of this study..." The paper misuses the category of a zoonotic pathogen. Here is the WHO: "a zoonosis is any disease or infection that is naturally transmissible from vertebrate animals to humans." So long as the pathogen primarily or extensively relies on non-human animals as hosts, and transmits from non-human animals to humans, it is a zoonosis.

What do the authors mean by "host jump"? If it is a sporadic infection but not an evolutionary host switch, fine - it is a zoonosis. If it is a host switch that leads to evolutionary adaptation such that humans become a primary host, and the pathogen can circulate permanently in human populations without an animal reservoir, it is no longer a zoonosis.

Also, it is not clear what is the timeframe of this study. When does it start and end? If we count as a zoonosis any pathogen that adapted to humans within the last ~25,000 years, then virtually every significant pathogen would be a zoonosis (just excepting "heirlooms" like herpes viruses).

55 reference 2 is significantly out of date

65 reference 5 is a popular website; it would be more responsible to cite a scholarly source for this major claim

66 references 6 and 7 are out of date and reflect a lack of engagement with the work of historians. Dobyns' work on the impact of European arrival has long since been shown to be problematic. Also, reference 7 lacks the author/editor (Little)

and has the year wrong and is now superseded by a range of important syntheses.

73 reference 10 is not an appropriate citation for the point that the Neolithic revolution facilitated pathogen transmission.

77 the paper should eliminate all references to the "first epidemiological transition." While the work of Armelagos had a major impact for a time, this construction never really became established. Sedentarization, animal domestication, urbanization, and long-distance exchange were very different processes, spread over 5-10,000 years (with different order and different timing in different regions).

111 the main text should give a sense of how and why these 1,313 individuals were included in the study. this is not a random sample, obviously, and seems dominated by access to Northern European archaeological samples plus Eurasian steppe samples. That's fine, of course, it's just important to be a little clearer because it seems that the contingencies of what the authors happen to have worked on have created a very particular chronological and geographical profile.

111 How many of these samples have been previously screened and how many human and/or pathogen genomes are new? In particular, we need a summary of how many new ancient pathogens have been positively identified. We learn deep in the paper that some of these are confirming previous findings.

139 here or elsewhere it needs to be clearer that these are exclusively DNA viruses

214 so are all of the hits for *Y. pestis* new?

230 "medieval cemeteries in Denmark" = late medieval, correct? Should be clear, and give specific date ranges associated with this claim.

238 the findings re: louse-borne relapsing fever are significant and striking. This is one of those important but underrated diseases. We know next to nothing of its evolutionary history and human history. This needs a discussion for context in a supplemental file, with more context about what is known of the phylogeny, evolution, ecology, and history of the pathogen. Really, this could be a separate paper, but at least needs sufficient context here.

262 the discussion of *E. coli* and *Salmonella* needs to be much clearer
shigella spp are lineages of *E. coli*
can we be sure that with these low ANI these are really *shigella*?
possible to distinguish the other strains of *E. coli*? are these enterotoxigenic?
is the *S. enterica* typhoidal (including paratyphoid strains) or non-typhoidal?
also, what are dates of all of these pathogens in this sample? could be very interesting.

271 Food-borne infections? This is confusing and probably misleading. Are these not the fecal-oral pathogenic strains?

286 the "non-finding" of *V. cholerae* is unsurprising and should probably not be the example used. The lack of *M. tuberculosis* is certainly interesting (to the extent that it is not an artefact somehow - I'm a bit skeptical). Could mention any number of major bacterial diseases that are not evidenced. Maybe most interesting of all is typhus - which has been of little interest to archaeogeneticists unfortunately!

330 why no previous mention of diphtheria. is this the oldest reported case? would be very interesting if it's high-confidence.

348-9 the patterns of plague are significant, but more detail is needed to engender confidence. for instance, can we exclude that the high incidence of *Y. pestis* from 5000 bp to 2800 bp (the paper is a bit unclear on the window), followed by a hiatus, is not influenced by sampling, at least in part? for instance, what would the incidence be in Europe and in central Asia, separately, for 1000-year time windows? Plague reappears around 2000 bp where - central Asia? Are these strains ancestral to the first pandemic? Then, does the subsequent peak really represent the first pandemic as we know it (from ca. 541-750 CE)? Is the following hiatus really 800 years? If so, that implies that the hiatus started around the time the first pandemic should launching. This whole discussion is frustratingly opaque.

364 again, more detail on the spatial dynamics of LBRP would be helpful.

381 checking the classifications in S3, why is *B. quintana* zoonotic - is it not a human pathogen?

B. recurrentis is not a zoonotic disease. It is transmitted from human to human. If the authors have evidence for the adaption from an animal to a human host during this time, they should present it. But other pathogens they categorize as anthroponotic also adapted to humans in this time frame (smallpox, TB, for sure, and probably others).

R. prowazekii is not a zoonosis. It does circulate in squirrel populations in the Eastern US but only a handful of cases have transmitted from animal to human. It is simply a human disease, but we have absolutely no idea how it adapted to humans or when.

R. typhi, *S. enterica* (if typhoidal), etc. are not zoonoses.

The four malaria parasites are not zoonoses. Their ancestors were primate pathogens. *P. vivax* was a human pathogen

already in the Pleistocene. *P. falciparum* probably did adapt to human hosts in the last few thousand years, but this is debated.

393 reference 54 is inappropriate - out of date and perhaps the source of the paper's confusion about transmission patterns and evolutionary history.

400 given the confusion and deep inconsistency about what counts as a zoonotic pathogen, it is impossible to tell if the claims about the patterns of zoonoses can be salvaged. Even so, one would want stronger tests of robustness against preservation and sampling biases (e.g. are most of the pre-6500 ybp hits not potentially from environmental sources?).

Once LBRP is removed, I think most of the zoonoses will be *Y. pestis* and leptospirosis. So the claim then perhaps becomes about *Y. pestis* and leptospirosis, not zoonoses.

S1 it would be helpful if there was a sheet (S1?) that allowed you to see which archaeological samples had which pathogen species - it is hard to flip back and forth.

S1 what is Barrie 2023?

S1 make clear that these dates are years BP (1950?)

S3 viruses are not a domain, so maybe another term?

S3 In any case, we can agree that the smallpox virus is not a bacterium.

Did I miss the archaeology supplement? I sincerely apologize if so. If not, where is it? Detailed site context and individual grave context should be standard practice in archaeogenetics. We don't know, for instance, if none/some/many/most of these samples are from contextually ordinary burials. We don't really know if the dating is good. I assume "direct" dating is radiocarbon? Are the upper and lower ages 2-sigma?

Referee #2

(Remarks to the Author)

###Overview:

This is a remarkably impressive effort to screen and report the microbial/pathogen content found in the off-target reads of 1,313 ancient human remains covering 35,000 years of Eurasian history. While a growing field, ancient pathogen genomics has tended to focus on specific species, reconstructing their histories. A more holistic assessment of disease burden through time has been lacking and major uncertainties surround how the human experience of infectious disease has changed through time.

The authors provide results that challenge the strongly held view that zoonotic diseases took off during the Neolithic, instead suggesting that while zoonotic pathogens may be linked to the domestication of animals it isn't until the period of Steppe migrations that zoonotic pathogens are detected to increase in prevalence (detection), hence supporting a wider role of human migrations (approximated by Steppe ancestry) in the distribution of modern pathogens.

The intention to create the first spatiotemporal map of human infections is ambitious, innovative and exciting and the authors should be commended for making all fastq reads available, including well annotated code and metadata. I have no doubt this project will spur further research.

While I am excited by this project, I do have some concerns over the interpretations. Some very strong claims are made, and I feel quite a bit more could be done to discuss the potential limitations of the input data and resulting insights. Even addressing these questions using modern day metagenomes (which would be an interesting comparator) would be fraught with the challenge of accounting for sampling bias and research effort. Furthermore, the classification of microorganisms into pathogenic or non-pathogenic or zoonotic or opportunistic etc. is non-trivial.

I outline my comments below.

###Major comments:

The authors construct a database of 44 'human pathogens'. Little attention is given to how this was devised. The status of a species as a human pathogen or not is often not that clear cut, particularly for species where the life histories through time and reservoir hosts are unknown or poorly characterised. What do the authors consider as a 'pathogen'? Databases do exist of 'human pathogens' – were these considered? Eg. work from Bartlett et al. discuss and release a much larger database of 1513 bacterial pathogens <https://www.microbiologyresearch.org/docserver/fulltext/micro/168/12/mic001269.pdf?expires=1695037757&id=id&accname=sgid025015&checksum=EE127AD22A68080B022BA26644ABBEA0>. The selection of genera to be considered as pathogens may well colour resulting observations. Similarly, the classification into zoonotic, anthroponotic, sapronotic etc is non-trivial but not explained. There certainly may be anthroponotic species which

are in fact zoonotic, though the animal reservoirs have been poorly characterised. Eg. some argue HBV is not anthroponotic <https://www.wjgnet.com/1007-9327/full/v20/i24/7665.htm>. I would recommend carefully evaluating host-pathogen association databases eg. CLOVER to provide justification for some of the classifications. This is very important because the major observation that zoonotic pathogens are not detected before 6,500BP is based entirely on how the authors chose to classify their detected species.

aDNA has amazing potential to provide unexpected insights. I have no doubt that the distribution of infectious diseases was likely different in the past, but some examples in text do need additional discussion. An example is the claimed IIV-31 human infection. The authors comment that this is a virus of woodlouse but without citation, and it does seem like a virus of invertebrates rather than one you would find in human teeth. Does the virus receptor allow entry to vertebrate cells? Can we believe this? In addition, other unexpected observations such as Proteus virus Isfahan may be genuine but this species has also implicated as a possible contaminant in DNA extraction kits/lab environments. Are the damage profiles strong in this case? These are not the only unexpected observations. Amongst the provided supplementary tables there are hits to citrus plant viruses, cattle parasites, and fish pathogens. How confident can we be that the results presented relate to genuine ancient human infections? This does bring into question some of the observations and these should be discussed with more care.

A high % assignment to toxoplasma is evident across the dataset. This is likely due to eukaryotic contamination in the reference database rather than toxoplasma being present at such a high frequency. I would be concerned, particularly for parasite species, that contamination in the RefSeq reference genomes or eg. retained adapter content etc. could lead to spurious hits and inflated prevalence estimates. What steps were taken to mitigate for this?

Previously published datasets are not really considered and observational data (place and time) is readily available for many human pathogens from eg. the SPAMM initiative <https://github.com/spaam-community/AncientMetagenomeDir>. While I appreciate exhaustive inclusion is a big ask, I think for *Y. pestis* and LBRF (I note this is only one previously published case) these should be considered in the assessment of temporal dynamics. Certainly for *Y. pestis* this would add a good amount of data. If this data can't be included given the need for a 'rate' of detection then could some of the claims be validated given published datasets. Eg. is it true that there is an 800 year period of no published observations in published *Y. pestis* data also? And why does an absence of plague support a prevalence of LBRF....? Are the authors claiming some cross-immunity or the such like? This feels a strong statement without further expansion.

The authors should be careful about overstating what can be gleaned from this dataset. It is of course very impressive in scale but even the best efforts are likely to be only a partial proxy to "infectious disease load over time". There are a number of potential issues with using this data to generalise to all human infectious disease. This includes that the data is Eurasian centric rather than global, not all species are considered, the work is restricted to DNA pathogens, there is differential preservation and survival of different microbial species (each species likely has a different LOD and some are very unlikely to be found despite being highly significant eg. *Mtb*). Perhaps some species, suggested as prevalent in the past, have high asymptomatic rates and are under sampled and studied today meaning their current distribution is poorly known. No systematic screen has been conducted of contemporary metagenomes so for many pathogens it is difficult to comment on their true distribution and prevalence today. These points should be more carefully discussed.

###Minor comments:

Introduction: The statement 'the burden of infectious diseases has also left lasting impressions on human genomes, where the selective pressure exerted by pathogens has continuously shaped human genetic variation' feels too strong. In fact, it could be argued that given the high postulated impact of infectious diseases in human history, efforts to identify confident susceptibility loci in human populations are surprisingly unproductive, aside from archetypal cases such as *Plasmodium*. The cited study (Karlsson et al. Nat Rev Gen 2014) has very few compelling examples for instance. I would modify this statement.

The introduction comments on 1313 individuals but only 896 samples are discussed in the abstract. 1272 individuals are presented in extended data 1. Could these discrepancies be clarified?

Non-human read classification is highly variable and implies marked differences in endogenous content. To contextualise, could the distribution be given for the % of unassigned reads in the text?

For the tooth associated metagenomes there is a mixture of the metagenome being modelled as 2 or 3 components – the authors stating the latter relating to the abundance of co-occurring oral microbiome species. Is there anything different about the sampling of these teeth? For example, use of root or calculus etc. Does sampling lead to a difference in microbial profile given the biota of teeth is likely highly heterogenous. Are there cases where different teeth have been sampled from the same individual? How similar or different are the resulting taxa profiles?

Were fungal species considered? If not, why not?

How should 'strict' and 'lenient' metaDMG be interpreted?

It seems there is quite a lot of scope to comment on the necrobiome or shared environmental microbiota across this dataset but this is not discussed. A characterisation of these components could be of high relevance to distinguishing signatures of pathogenic infection from commensal colonisation. Did the authors consider this or can they comment?

More nuance is needed in places. For example if anything, given the need for high infectious loads, tropism to bone and reduced degradation rates, co-infections are likely underestimated in this dataset rather than "readily detected".

Some strong claims are made without accompanying evidence eg. that the processes supporting zoonotic bacteria are the same for zoonotic RNA viruses.

Fig 5 – the 'other' category shows the highest detection rate across time. What are these? Contaminants? Commensals? Do they share similar properties.

Referee #3

(Remarks to the Author)

Sikora Willerslev and colleagues screened shotgun sequencing data from 1,313 ancient human remains, covering 35,000 years of Eurasian history, for ancient DNA derived from bacteria, viruses, and parasites. The manuscript reports identifying from 2,400 individual microbial species hits in 896 samples including the food borne pathogens *Yersinia enterocolitica*, the animal-borne *Leptospira interrogans*, malaria-causing parasite *Plasmodium vivax*, *Yersinia pestis*, the causative agent of plague, Hepatitis B virus, and *Borrelia recurrentis*, the cause of louse-borne relapsing fever (LBRF). They conclude that their paleoepidemiologic analyses document an increased burden of zoonotic infectious diseases following the domestication of animals.

Major issue. Sample contamination.

Given the number of papers that have been called into question due to possible sample or reagent contamination – including by the authors of this paper – I would have expected more explicit analysis to control for contamination. These samples have all the issues that have made other studies be published to great fanfare only to fall into question when data is examined by the scientific genomic community. The authors are re-analyzing publicly available data that was generated to examine human evolution. As such great care was taken in the original studies to ensure that there was no contamination with current human DNA. However that is a very different process than what one would embark upon to characterize microbial DNA. There are no negative controls for these samples such as soil DNA or reagent sequencing.

Major issue. Specificity of 'hits'

The 'hits' explored by the authors are often as little as 20 reads. Extremely little detail is provided to understand how the authors make the claim that these are 'hits' rather than random reads from another microbial species that has a weak hit to a pathogen in the database.

The authors did a comprehensive analysis on the ancient human bone and tooth samples but did not report the results for controls (e.g., corresponding soil samples). Since the biomass of microbial DNA in ancient human samples is low and the positive hits based on read mapping are sometimes based on as few as 20 reads, the authors need to demonstrate that those hits were not false positives or sourced from contaminations.

In line 196, authors further concluded that lower ANI and genome mixture suggested environmental sources. However, multiple strains can be present on one individual. The issue is complicated by the approach taken by the authors when identifying "a single best-matching species hit for each sample and genus of interest", meaning that the multiple alleles could also be due to a mixture of species. Authors should address the effects of multiple species/strains on their results.

'Identifying eukaryotic pathogens is challenging, as sequence contamination from other organisms 291 regularly occurs in reference genomes deposited in public databases' Seems a little insincere to reference Salzberg's landmark study at this point in the manuscript given the arguments Steinegger and Salzberg are making about all of RefSeq.

Now suddenly authors evoke statistics such as 'low support from coverage evenness statistics due to the ancient reads 294 predominantly mapping to short contigs representing contamination' but were these statistics used for bacterial pathogen read mapping??

'we identified two cases of infections with the malaria-causing parasite *Plasmodium vivax*,' what constitutes a 'case'? is this still 20 reads?

Overall detection rates of plague were interesting but by this point I was so concerned about the data quality. For example *Borrelia recurrentis* rate is 0.05% but remember from Fig 2A these are mostly lenient DMG.

Out of 82 identified putative hits (the majority of 266 which were *Escherichia coli*; 55 cases, 4.1% of samples), 27 were found as co-occurring hits in a set of 267 12 individuals (Extended Data Fig. 7). Do any of these match lab strains of *E. coli*? That would be a red flag for me.

Escherichia coli; 55 cases, 4.1% of samples

E. coli is a minor component of healthy gut community and there is no reason one would expect it to be in bones so maybe this is more a concern about level of sample contamination?

Major issue. Unmapped reads.

The unmapped reads are a vast percentage but not addressed.

'The highest fraction of reads classified include soil-dwelling taxa such as *Streptomyces* and 116 *Bradyrhizobium* or taxa with broad ecological distribution, such as *Pseudomonas*, reflecting a 117 predominantly environmental source of microbial DNA in the samples.' *Bradyrhizobium* can also be found in water supplies as a contaminant so could come from reagents which means that other microbial contaminants are also there.

Additional overall issue. Microbiology of Death.

Metcalf has performed experiments and analyses to evaluate the 'microbiology of death'. This work should be read by the authors and referenced. It will change the ways in which they make assumptive statements such as 'highest relative abundance

124 of genera such as *Actinomyces*, *Streptococcus* or *Pseudopropionibacterium*, commonly associated with 125 the human oral microbiome. This component was observed at noticeable frequencies in many samples 126 derived from teeth, but only rarely in those from other tissues (Extended Data Fig. 1e, f). This suggests

127 that a substantial fraction of the identifiable microbial DNA found in many ancient tooth samples derives 128 from the endogenous oral microbiome of the human host.'

Specific comments on figures.

Figure 1 toy example of data workflow shows an idealized version. What are the actual numbers is more important; e.g. what % unmapped reads?; when is there 5x coverage as depicted in species A1 reference? Digging into the data this seems like highly idealized data. Would be much better to show realistic data and accurate data flow.

Fig 2. DMG never defined but used as key to legend? What does DMG mean? Seems central to argument but also seems like a term the authors made up? DMG = DNA damage microbial genome??

High ANI strict DMG should be shown on the bottom of the bar chart if these are the higher confidence data reads to facilitate making comparisons between genera rather than method used by authors. This would for example show that *Mycobacterium* has few high ANI samples. Same for *Bordetella*. Since both of these genera become main stories, the data is misleading in presentation. By comparison *Yersinia* and *Staph* are both high ANI.

Figure 2. what is a 'hit' that is being counted in 2A?

Fig 2d. really low confidence data is being shown. Total of 115 reads and read depth of either >1 or >2. Can't really tell from legend. What does it really mean to look for multiallele rates with 1 X depth?? sample with higher multi-allele rate ANI is low.

Methods.

'We identified 39 cases of *Y. pestis*' based on what criteria??

Same question for 'The leprosy-causing bacterium

248 *Mycobacterium leprae* was identified in eight individuals (0.6% detection rate),' it's not that 0.6% of the reads matched *M. leprae*. It's that 0.6% of aDNA samples had at least 20 reads that matched *M. leprae* with low ANI and low aDMG.

Figure 6. what is the 'sequencing' feature used in Fig 6 as a co-variate?

Methods. Dataset is not described more than '515 We compiled a dataset of ancient DNA shotgun-sequencing data from 1,313 ancient individuals, of which

516 130 are newly reported in this study.' Should at least provide BioProject, BioSample IDs for samples used in study.

These 130 new samples – how were they treated in this study? Were appropriate controls collected for them to monitor where microbial contamination occurs in the process?

We selected the 527 assembly with the most unique k-mers assigned as the representative reference genome for each species in 528 a particular sample. If no reads were assigned to any assembly of the species in KrakenUniq, we selected 529 the first assembly for mapping.'

Does this mean that even if only 1 or 2 reads were assigned to a species, this was used?

As this is a major part of this article, some additional explanation should be provided: 'Authentication of ancient microbial DNA 535 To authenticate ancient microbial DNA, we calculated sets of summary statistics quantifying expected 536 molecular characteristics of true positive ancient microbial DNA hits63:' in addition to referencing a review written 7 years ago. How was sequencing done on different platforms at different institutes controlled for?

e.g. putative microbial "hits" includes 605 - Number of mapped reads \geq 20. Mapped with what ANI over what length? How long are these reads? What k-mer size is used?

Minor comment. In line 399, authors claimed that the "sapronotic" class occurred "at lower detection rates (mostly below 10%)." However, in Fig 5a, the "sapronotic" class seems to be at relatively higher detection rate and certainly not below 10%.

Please clarify such inconsistency.

Question. What about AMR genes? Were these identified in any of the samples?

Referee #4

(Remarks to the Author)

* The landscape of ancient human pathogens in Eurasia from the Stone Age to historical times

* Downloads/ancientPathogensREV.pdf

This paper represents an important and interesting contribution. The analyses are useful. The authors are commended for sharing a code pipeline and being clear about availability of inputs.

In my opinion, the statistical explanations, and perhaps methodologies, need to be improved a little.

The authors need to discuss their Bayesian priors. Even if they simply used inlabru defaults, this should be clearly stated. Ideally, they should describe a brief plan (we planned to use defaults no matter what; we planned to follow a certain procedure to tighten priors under certain conditions; ...)

The analysis shown in F6 is inadequately described. There is no context that can be used to understand the meaning of the word "positive" at L425, or the ancestry effects in F6. What is the baseline? There is also no basis to understand what is meant by "effect size".

Using WAIC to exclude variables from a Bayesian model does not seem like best practice. The authors should consider simply using their full Bayesian model, or else explain their choice.

The authors need to describe how they put pathogens into five classes. Did they take any steps to avoid selection biases arising from already-observed patterns when they did so?

MINOR COMMENTS

LBRF is spelled two different ways on L45

Qualifying word needed on L47 (e.g., "inferred" form of transmission)

L80 "small number" is not ideal, maybe "small proportion"

L82 "pathologies" is the wrong word

s at L107 needs to be clarified

The use of k to mean two completely different things in the topic analysis is not ideal (L120)

The reason for requiring damage to be present should be mentioned briefly in the workflow section (~L148), maybe also in the Intro (~L107).

The use of on upper and one lower bound, and the switch between percentage and reads per million, around L155 is awkward. Maybe just do median, or else median and range. And maybe do everything as reads per million reads (0.0x% is a confusing idiom for many, I think)?

I vote against the artificial categorization in F2b. There is no need to explain "significance" or (more confusingly) "similarity" or to divide the plot into arbitrary groups. The ordering and the CIs convey what needs to be conveyed.

"such as" is a tiny bit confusing in L180

Authors should briefly explain what is meant by "cal. BP", and also the associated tags (e.g., NEO168). I was unable to understand the latter.

s at L287 needs to be edited for clarity

The interjection about vivax vectors seems poorly contextualized and extraneous.

L302 might be stronger without the word "ancient" -- a serious investigation of this question would focus on remains from 100 or so years ago, I think.

The s. at L335 seems extraneous and poorly supported.

At L347 it would be good to say briefly how the properties of extinct strains are known. Alternatively, the authors could say

"believed to have had" if they don't want to get into it.

L434. "No effect" should virtually never be used to describe a statistical inference. This case is not by any means one of the exceptions.

Text around L470 is well qualified, but still comes across as too strong. I suggest dropping "sweeping through the continent".

¶ at L483 is also a bit too strong. Certainly this analysis is far better supported than earlier analyses, but there are presumably other possible explanations for the detection patterns.

Authors need to say more about the topic-model analysis. Was number of topics the only choice they needed to make? How did they make it?

s at L478 comes across a little strong. Which part of this interpretation is supported?

Jonathan Dushoff

Version 1:

Reviewer comments:

Referee #1

(Remarks to the Author)

This is an excellent paper and the revised version is much improved. It will make several original contributions, as highlighted in my previous comments. Given the potential significance of this paper, I would advise the following further suggestions be taken into consideration by the authors and editors.

The classification of diseases is much improved. A few issues remain. The new classification compresses (1) reservoir and (2) mode of transmission into one dimension, "Microbial origin group." That is invalid. Plague is both a zoonotic disease (animal reservoir) and a vector-borne disease (highly dependent on flea-borne transmission). The malaria spp discussed are vector borne but also adapted to humans. So maybe two dimensions: reservoir and transmission (oral-fecal, respiratory, vector-borne, sexual). This is something that has to be done right for the paper to make sense.

I also question the placement of shigella, typhoid, and related diseases as 'environmental.' That is misleading. They depend on human hosts (and possibly other animals, esp earlier in their evolutionary history) to persist.

The supplement on LBRF is helpful though still not quite adequate, given that this paper could become a reference for the study of this neglected disease. Can more please be said about epidemiological factors? This is really a disease of 'misery' and crowding? It is more common today in Africa but might colder winters have contributed in the past in temperate climates? Some of the historical claims are pretty bunk - the MacArthur paper.

line number in pdf

61-2 The authors of the paper have replaced an outdated reference to the history of infectious disease with something equally bad, note 7. It's a weakness of this paper that there is no engagement with history. There are lots of serious papers and books. I work in the field and have never seen the paper they cite. It is written by a clinical psychologist in upstate New York with no apparent training in history or archaeology, and it shows, because the paper is riddled with sophomoric errors. Please, look up a credible source on the history of infectious disease and its impact on societies!

117 Fig. 1 is great, but in the text, they don't seriously and earnestly address the main concern about sampling bias, which is not within-cemetery bias. Rather, how were these 1313 individuals assembled? the very fact that they are Eurasian means it excludes tropical (and most subtropical) regions, all of Africa, Australia, and the Americas. In other words, it is biased against the most densely populated parts of Eurasia, plus all the other continents. Eurasian steppe populations account for what percent of historical humanity - it would be possible to do a back-of-the-napkin calculation, and it's going to be less than 1%. But here they are what percent? This sample is exceedingly biased in that respect.

Just as importantly, what kinds of archaeological work generated this sample? We all know that anomalous burials and mass burials are disproportionately screened, so what is the role of that in the background of this sampling?

What percent of the sample is "urban," for each period?

283 the reported mortality for LBRF is problematic. first, i assume this is case fatality rate? be clear. second, this does not look reliable and is unsupported by an objective reading of the paper they cite. The paper says this "Case fatalities between 30% and 70% have been reported in untreated patients during major historic epidemics, but in treated cases, on average, 2–6% will die." To support this claim, that paper cites the 1970 paper "Louse-Borne Relapsing Fever; A Clinical and Laboratory Study of 62 Cases in Ethiopia and a Reconsideration of the Literature." I read that. It reports case fatality from .5% to over 50%, in some highly suspicious reported outbreaks from early 20C central Africa. LBRF may have been a serious

disease, and it has been very underconsidered. But rehashing bad work to dramatize it will undermine the important findings of this paper. I really hope the authors will work to get this right.

334 it's weird to be stubborn about V cholerae. The reason you don't find it is that it emerged in the late 18th century and only reached the sampled areas from the 1830s, and then very sporadically. And the point about M tuberculosis is fine, except we might have expected a much higher burden of advanced disease. What percent of the sample of 1313 is urban?

408 Starting at ~2,000 BP, plague reappeared in our dataset (samples
409 DA92, DA101, DA104, Kazakhstan and Kyrgyzstan, Central Asia; Fig. 3; Supplementary table S2),
410 including the period before the first historically documented plague pandemic (~1500-1300 BP,
411 VK522, Sweden and previously published cases

This is really interesting, but some things have to be clearer.

The plague starts showing up again around 2000 BP. But only in samples from plague's enzootic region, Kazakhstan and Kyrgyzstan. As best I can tell, there are only a couple of individuals sampled from the 'blank period' of plague from the enzootic region. Some discussion of the sampling is in order. It is essential to be clear that the positive plague samples prior to the historically documented First pandemic are from central Asia. This will strengthen the important point raised in the conclusion that "The pattern in the later periods is consistent with an epidemic period, distinct waves matching the historically described plague pandemics sweeping through Europe separated by periods without detection of the pathogen."

Also, where do these four Y pestis genomes sit on the phylogeny? I would presume the three central asian cases are basal (one has been published, I think).

Finally, I am pretty sure there is a grave error here. They report an individual positive for Y pestis from Sweden. VK522. From Oland. They cite Margyaran 2020. I go there and look for VK522. I go to the Supplement of Margyaran 2020. I see VK522 is equated with Oland 1052 and given a radiocarbon date 386 ± 80 CE. That aligns with the date in the supplement of this paper and used in all the analysis. An individual positive for Y pestis in Sweden from the fourth-fifth century AD is a huge find. The people who work on the first pandemic will immediately try to rewrite the history of everything we know about it! But honestly I am skeptical plague would show up there, so I dig a little. The Margyaran supplement cites Wilhemson 2017. There, in the supplement to this Lund PhD thesis, I read that this individual (Oland 1052) was radiocarbon dated to 386 ± 80 BC. This thesis uses BC and CE. That's 700-800 years different. Now, this is a grotesque mistake, with massive consequences (because others will not dig, as apparently the authors of this paper did not dig, to do minimal due diligence on a finding that is pretty surprising), so I have to think maybe it is me who has misunderstood something. If so, I sincerely apologize. If not, dear goodness please get this correct and double check things so that plague historians do not get misled.

In all there would be a lot of value in clarifying what this paper says about the history of plague by more phylogenetic analysis and more clarity on sampling. I know that a phylogenetic paper on the plague is another study, but I'm not sure it's responsible to give this much info here without just going further.

Again, this paper has the potential to be a blockbuster, so it is crucial to address these remaining issues before hopefully publishing what will be a major contribution.

Referee #3

(Remarks to the Author)

The authors have provided a robust revision. While ideally soil and other negative controls would be included in a prospective study, I accept that the authors are unable to obtain samples like this when retrospective data collected for aDNA studies is used.

The only concern I have is with their handling of unclassified reads. They treated these separately from non-human classified reads, but unclassified reads can significantly impact relative abundances especially considering the high percentage of unclassified reads. I would suggest authors focus solely on the presence/absence of pathogens, omitting discussions on relative abundances.

Overall, I think this work can act as a pioneer in the field of ancient microbial DNA research by reanalyzing ancient human samples and proposing a robust authentication framework.

Referee #4

(Remarks to the Author)

The authors' revisions are appreciated, and they have addressed most of my concerns adequately.

I have one remaining concern.

Using WAIC-based selection is presumably a good way to choose a model for posterior predictive accuracy (as the authors discuss in their rebuttal, but not in their MS). The authors don't even make an argument that it's appropriate for Bayesian `_inference_` (i.e., parameter estimation), and I don't believe that it is. What kind of "overfitting" are the authors worried about here? They seem to focus on the `_effects_` of covariates, not on any predictions.

Referee #5

(Remarks to the Author)

This is a very significant project, the first of its kind, which makes full scale screening result publicly available, thoroughly analyses the data and uses a well tested screening workflow developed for the project. I commend the author on undertaking such a big task, which involves the work with a very large set of taxa and the reuse of previously published data. Additionally, I was happy to see that additional data and code was or will be made available (although I would have liked to see an example). The workflow itself is well-designed and filtering parameters well-chosen, although some aspects such as the limitation to a single species in each genus are not ideal, and I was somewhat confused why genera were only included if more than 2 established human pathogens were present. I also appreciate the use of presence/absence data on infectious agents, which is sadly mostly missing in the literature. Since the knowledge of the presence of a disease in a population at a given time can already be precious knowledge, as this study clearly demonstrates.

Reviewer #2 comment response:

All of the minor comments from reviewer #2 were adequately addressed.

R#2: "While I am excited by this project, I do have some concerns over the interpretations. Some very strong claims are made, and I feel quite a bit more could be done to discuss the potential limitations of the input data and resulting insights. Even addressing these questions using modern day metagenomes (which would be an interesting comparator) would be fraught with the challenge of accounting for sampling bias and research effort. Furthermore, the classification of microorganisms into pathogenic or non-pathogenic or zoonotic or opportunistic etc. is non-trivial".

I have similar concerns. While the project is indeed novel and highly relevant, I am concerned about some conclusions drawn in this article. I think the methodological work is very good and while not without issues, constitute a very good basis for a spatio-temporal analysis. However, when it comes to the broader interpretation of results beyond the description of large scale trends (not based on specific species), I am more sceptical. The analytical results are present/absence data, meaning that we know nothing of the genomic structures or phylogenies of these organisms. I appreciate that it is indeed challenging to work with such a large bulk of taxa and that much of the authentication process has been well automated, so to speak, however, particularly when such important conclusions are being drawn, final identifications should be double-checked anyway and for any specific conclusions relating to function and adaptation, full genomes should be available. As the bulk of your pathogenic hits are reduced to a handful of species, this should be doable.

R#2: "The authors construct a database of 44 'human pathogens'. Little attention is given to how this was devised. The status of a species as a human pathogen or not is often not that clear cut, particularly for species where the life histories through time and reservoir hosts are unknown or poorly characterised. What do the authors consider as a 'pathogen'? Databases do exist of 'human pathogens' – were these considered? Eg. work from Bartlett et al. discuss and release a much larger database of 1513 bacterial pathogens (link). The selection of genera to be considered as pathogens may well colour resulting observations. Similarly, the classification into zoonotic, anthroponotic, sapronotic etc is non-trivial but not explained. There certainly may be anthroponotic species which are in fact zoonotic, though the animal reservoirs have been poorly characterised. Eg. some argue HBV is not anthroponotic <https://www.wjgnet.com/1007-9327/full/v20/i24/7665.htm>. I would recommend carefully evaluating host-pathogen association databases eg. CLOVER to provide justification for some of the classifications. This is very important because the major observation that zoonotic pathogens are not detected before 6,500BP is based entirely on how the authors chose to classify their detected species."

This was adequately addressed.

R#2: "aDNA has amazing potential to provide unexpected insights. I have no doubt that the distribution of infectious diseases was likely different in the past, but some examples in text do need additional discussion. An example is the claimed IIV-31 human infection. The authors comment that this is a virus of woodlouse but without citation, and it does seem like a virus of invertebrates rather than one you would find in human teeth. Does the virus receptor allow entry to vertebrate cells? Can we believe this? In addition, other unexpected observations such as Proteus virus Isfahan may be genuine but this species has also implicated as a possible contaminant in DNA extraction kits/lab environments. Are the damage profiles strong in this case? These are not the only unexpected observations. Amongst the provided supplementary tables there are hits to citrus plant viruses, cattle parasites, and fish pathogens. How confident can we be that the results presented relate to genuine ancient human infections? This does bring into question some of the observations and these should be discussed with more care"

While this is good, I do believe that particularly for the pathogens which play a larger role in the discussion (and especially the ones which are not really expected to be found within the sampled tissue) and the conclusion, a more thorough species validation should be presented in the SI.

R#2: "A high % assignment to toxoplasma is evident across the dataset. This is likely due to eukaryotic contamination in the reference database rather than toxoplasma being present at such a high frequency. I would be concerned, particularly for

parasite species, that contamination in the RefSeq reference genomes or eg. retained adapter content etc. could lead to spurious hits and inflated prevalence estimates. What steps were taken to mitigate for this?"

Sikora et al: We agree with the reviewer that *Toxoplasma* hits can be caused by database contamination, in fact we are presenting them as examples of contamination, and how our authentication approach guards against such false positive results (e.g., coverage evenness statistics, Extended Data Fig. 4). As a consequence, our final authenticated results do not contain any hits for *Toxoplasma* (Supplementary Table S2).

This was adequately addressed. But please see my remarks below regarding the simulations and *Plasmodium* species.

R#2: "Previously published datasets are not really considered and observational data (place and time) is readily available for many human pathogens from eg. the SPAMM initiative <https://github.com/spaam-community/AncientMetagenomeDir>. While I appreciate exhaustive inclusion is a big ask, I think for *Y. pestis* and LBRF (I note this is only one previously published case) these should be considered in the assessment of temporal dynamics. Certainly for *Y. pestis* this would add a good amount of data. If this data can't be included given the need for a 'rate' of detection then could some of the claims be validated given published datasets. Eg. is it true that there is an 800 year period of no published observations in published *Y. pestis* data also? And why does an absence of plague support a prevalence of LBRF....? Are the authors claiming some cross-immunity or the such like? This feels a strong statement without further expansion"

The data was included (although it can be noted that within the last week multiple new *B. recurrentis* genomes were published on BiorXiv Swali et al 2024). While I agree with the conclusions regarding *Y. pestis* and *B. recurrentis* regarding lifestyle and hygiene, I think that any speculation regarding body lice transmission is out of scope here. You have no data on the genomes of these cases, merely presence absence data. Additionally, while the study did indeed detect large amounts of cases of *Y. pestis* and *B. recurrentis*, both these pathogens are very well suited for molecular detection in teeth and are highly adapted human pathogens. Many are not, be it because of tropism, low viral/bacterial load, preservation or non-DNA genomes. Correlating them in such a way is not really warranted as the whole picture is still not available, and you cannot exclude genome variation or ecological aspects. While the absence of cases is certainly interesting to note and would be interesting to investigate further, the data presented here is not an adequate basis for a hypothesis on vectors or very specific cases of adaptation.

R#2: "The authors should be careful about overstating what can be gleaned from this dataset. It is of course very impressive in scale, but even the best efforts are likely to be only a partial proxy to "infectious disease load over time". There are a number of potential issues with using this data to generalise to all human infectious disease. This includes that the data is Eurasian centric rather than global, not all species are considered, the work is restricted to DNA pathogens, there is differential preservation and survival of different microbial species (each species likely has a different LOD and some are very unlikely to be found despite being highly significant eg. *Mtb*). Perhaps some species, suggested as prevalent in the past, have high asymptomatic rates and are under sampled and studied today meaning their current distribution is poorly known. No systematic screen has been conducted of contemporary metagenomes so for many pathogens it is difficult to comment on their true distribution and prevalence today. These points should be more carefully discussed"

The authors have included references to limitations in the manuscript. However, I believe the manuscript would benefit from more details, particularly regarding tropism, LOD, and how disease phenotype can affect recovery.

Beyond the issues raised above, here are my comments:

A lot of this work depends on the databases used, so I would have liked it to be described in a bit more detail. Were Univec/Emvec type databases included? Since you used KrakenUniq, databases should be dusted by default, but how much is the mapping in bowtie impacted by sequence complexity issues? While *H. pylori* and MTB do have skewed GC content, most other genomes that were tested during the simulation do not, and I am wondering how it fares with viruses, which can exhibit repeats over large percentages of their genomes.

I am very happy that simulation was included in the analysis. However, except for the two organisms noted above, the tested species are mostly comparable in genome length, GC and plasticity. I am a bit concerned about the analysis of *Plasmodium* hits and those with similarly large genomes, which are not accounted for during simulation (the best hit covers 0.15% of the reference sequence). These genomes differ significantly from 3-5Mb bacterial genomes and would need a lot more data to validate. Considering your issues with *T. gondii*, including larger protozoan genomes and viruses would be advantageous. Additionally, regarding your criteria for a positive call, depending on the genome plasticity, available diversity and sequence divergence within the species, the use of ANI, for example, which is of course a very important metric, can need slightly different cut-offs. At least one of the chosen species *Y. pestis* evolves clonally and similarly to *M. leprae* could require higher ANI cut-offs, and the opposite can be true for very plastic/divergent species or ancient strains that fall distantly basal to the modern diversity. These would also often display increased mean edit distances. So it would be good to show, with your simulated data, that these differences don't make an impact with a wider choice of species and the inclusion of more divergent strains.

While I appreciate that the simulation data has shown the overall suitability and effectivity of the workflow for pathogen detection in ancient DNA datasets, false positive cannot be completely excluded. Databases are still heavily biased towards pathogenic species and many environmental/commensal organisms have not yet been sequenced or assembled. Most of the hits reported in this study (130 out of 195) are based on ca. $\leq 5\%$ of genome coverage of the reference genomes. This a) doesn't allow you to know anything about their genomes beyond their potential presence (which I realise is limited by the available shotgun data to begin with) and b) mostly limits the identification to "probable" cases, as it is unlikely the species can actually be fully validated with this amount of data. While you do point this out early on in the manuscript, it is mostly omitted later. Talking of "probable/likely" cases would be more appropriate in most instances.

That being said when reporting on species, which based on standard tropism and disease phenotype should not or only rarely be present in the sampled tissue (e.g. *Y. enterocolitica*) (and species which have not been reported previously), I would expect a more thorough analysis on species validation to be described in the SI. E.g. whether all the plasmids needed show coverage or if specific sites are present.

(Remarks on code availability)

Code has been made available, and additional output and databases will be made available publicly as well. I did not review the full code itself, however, a more extensive README file would be needed.

Version 2:

Reviewer comments:

Referee #1

(Remarks to the Author)

In the revised version of the manuscript and accompanying tables, the authors have thoughtfully responded to the reviewers' suggestions and comments. The authors have now corrected the classification of pathogens, and they have added helpful material such as the discussion of LBRF in Supp. Inf. 3.

This is now a very clean paper. It is a major contribution both substantively and methodologically. I wholeheartedly recommend publication.

Referee #4

(Remarks to the Author)

Thanks to the authors for their revisions.

My point about "WAIC-based selection" was apparently insufficiently clear. I agree that DIC is appropriate for comparing these models, but I don't object to WAIC, either. My point was that – in the context of Bayesian `_inference_` – the authors should avoid doing selection at all if possible, and to provide a clear justification if they feel that selection is necessary. Selection is popular in this context because it provides sharper confidence bounds than would be supported by the original assumptions, but this is not actually a supportable reason to use it.

Referee #5

(Remarks to the Author)

The authors have addressed the majority of questions raised by me adequately and have provided extensive additional work. However, some questions remain. Please see below:

"We selected a set of 136 bacterial and protozoan genera (11,553 species total) containing at least two established species of human pathogens² to screen for ancient microbial DNA in the genus-level read mapping and authentication stages." I had ask for some clarification regarding this in my first review. This should be included in the text as it is unclear. Why were genera excluded based on the number of pathogenic species they carry?

Regarding comment 1b/3b:

I appreciate the addition of further validation for pathogenic species, although my reservations about functional and adaptive inferences in the manuscript remain.

For *C. diphtheriae* as far as I am aware the *tox* gene is not specific to *C. diphtheriae* and is present in homologous copies in other species of the genus, which should probably be noted.

In Table S5, the KrakenUniq statistics are the same for each header, which makes sense, but this should be specified in the legend somehow.

Regarding comment 5a/5b:

While I appreciate the addition of the paragraph in the conclusion, it is never directly tied to this discussion nor does it come

up within the discussion paragraph themselves, making it feel rather detached. In fact there has been little to no change to this section.

In the end sentences such as this one:

“Strikingly, this period of high LBRF detection coincided with a period without detectable plague activity (Fig. 4), further supporting the notion that the absence of plague represents a true reduction in incidence during this period rather than sample size limitations or poor DNA preservation”

Are speculative in nature, based on correlations discerned in the current data. While such a hypothesis can of course be proposed it should be made abundantly clear in the actual discussion why the observation could wrong or apply to a different set of pathogens, which the methods in ancient DNA and in this study cannot recover etc. This is especially true since you also further expanded the discussion on vectors, and particularly on non-flea vectors for *Yersinia pestis*, which have been and still are under considerable debate.

Regarding comment 11:

Regarding cases like NEO627/*Y.pestis* and NEO29/*B. recurrentis*. Are cases like these removed from the list of validated hits? I might have missed it but I did not see it made clear whether there were criteria for verification based on your additional analysis? While they might prove to be real with additional data, they are currently lacking coverage for a validation.

Note that i could not access this link without logging in and thus could not review its content (https://www.dropbox.com/s/clfi/8xbzgn5r5gnvjqi5dd/readsim.hum_microbe.summary.tsv.gz?rlkey=qa7sitz4dv03lz63yh59z9ja0&dl=0)

Version 3:

Reviewer comments:

Referee #4

(Remarks to the Author)

When I asked for a “clear justification”, I meant in the MS, not in a rebuttal. Not clear whether the authors thought I was the only person who could possibly want that. In any case, the authors' justification is not responsive to my concerns.

I tried to be very clear that I was not challenging the general idea of information criteria (which would be ridiculous), but instead referring specifically to the context of inference (written as `_inference_` in two of my reviews). The authors' primary citations are Gelman...13, which is explicitly about prediction (not inference), and vdS...21, which seems to be about inference, but does not seem to mention information criteria at all!

The authors don't make even an a priori case for why reducing model complexity (in this context, not some other context) is preferable. And of course their statement that lower IC values indicate better “fit” is technically wrong. Best practice would be to report parameter values and CIs from their full Bayesian model. Since they have produced a blizzard of information (table S6 and the rebuttal figure) in support of an argument that this would yield similar conclusions, they should not object.

Referee #5

(Remarks to the Author)

The authors have addressed all my comments satisfactorily. I think the manuscript would benefit from including the reply to comment 11 in the supplementary information, as this is pertinent information. However, I leave this up to the authors. And lastly I want to note, in case the new link will eventually be included in the publication, that the link provided in the last comment worked but the file had no extension and I could only use it because I had seen the extension on the old dropbox link.

We have structured this response in the following way: In the first section “Major changes to the manuscript”, we outline substantial changes and updates in the analyses and manuscript aimed at addressing broader concerns and/or suggestions from multiple reviewers. Following that, the section “Individual reviewer responses” provides replies to specific remaining concerns of individual reviewers. Comments quoted from the reviewer concerns are formatted in *italics*, and quoted text from the updated manuscript is shown in blue font.

Major changes to the manuscript

We have carried out the following major changes to improve the manuscript.

Updated analysis workflow

The revised manuscript presents results from an update to the pathogen screening workflow. In this new version, we are using *bowtie2* instead of *minimap2* for the mapping stage. While this change increases computation time slightly, we found that overall sensitivity was improved. With this update we see increases in the number of mapped reads for the ancient microbial hits reported in the first submission, and additional cases for many species which did not pass cut-offs previously due to low read numbers with the previous approach.

Broad extension in number of screened genera

In this significant update, we have substantially extended the list of genera screened, providing a much broader perspective on the ancient microbial landscape. In the analyses presented in the initial submission, we had screened 44 genera known to contain human pathogen species. For this resubmission, we have increased this number 4-fold, to 136 different genera. We have chosen this list of genera based on a recently published curated list of human pathogens by Bartlett et al 2022 (<https://doi.org/10.1099/mic.0.001269>; as suggested by reviewer 2), including all genera which contain at least two species of “established” human pathogens in their nomenclature. Importantly, the design of our workflow allows for any species within all genera to become a “hit”, irrespective of their pathogen status. We have attempted to further clarify this point throughout, to avoid the misunderstanding that only the specific pathogen species were targeted as in reviewer 2 commenting: “*The authors construct a database of 44 ‘human pathogens’*”.

In our database, the 136 genera together comprise a total of 11,553 species reference genomes, to each of which every sample could potentially be mapped to. Among the species in our final authenticated list of ancient microbial hits, 214 are “established” pathogens in Bartlett et al, whereas 278 have not been listed as pathogens.

Improved authentication

Contamination of various sorts is a major concern for us and our analyses, and understandably also for the reviewers. To further expand and clarify the measures we are taking to avoid false positives, we have now carried out simulations *in silico* of ancient DNA fragments for a set of bacterial species of interest, and applied the workflow to these simulated datasets. Our results show that we readily recover the true positive species even at very low input read numbers. The results of the simulations also allowed us to update cut-off values for summary statistics used in the authentication to further increase robustness of our results. The simulation approach is outlined in the methods section, and a new results section and accompanying figures (Extended Data Fig. 2 and 3) have been added. We have furthermore expanded the description of the workflow and authentication statistics in the initial part of the results section to further clarify how we authenticate true positive hits.

Classification of microbial species

Assigning discrete transmission categories such as “zoonotic” to a diverse set of microbes is a difficult task, and the reviewers rightly flagged this categorization as a concern. Furthermore, there was concern and/or misunderstanding about whether microbial hits represent infections (e.g., reviewer 2:

“Amongst the provided supplementary tables there are hits to citrus plant viruses, cattle parasites, and fish pathogens. How confident can we be that the results presented relate to genuine ancient human infections?” or references of reviewer 2 and 3 to the necrobiome).

To address these issues, we have modified our approach and grouped microbial species by the likely source of the ancient microbial DNA in our sample. Any species where we cannot with reasonable confidence rule out an origin of the DNA either from exogenous (e.g., soil) or endogenous (e.g., members of human microbiome communities such as skin or gut that can invade the body after death as part of the necrobiome) environmental source are grouped as “environmental”, with a further distinction as to whether they have been reported as human pathogen (“established” in Bartlett et al). Species which are known members of the human oral microbiome have now been grouped in a separate category, as we do have direct access to their tissue of origin from tooth samples. And finally, classes of “infection” which likely represent true bloodstream infections in the ancient individual. We further subdivided those into modes of transmission as previously, but have included a new category of “vector-borne” pathogens and reserve “zoonotic” for species where direct animal to human transmission is an important mode. We are aware that assignment to these infection subcategories are not trivial and their distinction can be fuzzy or even change throughout the evolutionary history of a pathogen (e.g., *Yersinia pestis* and fleas). We have thus attempted to be as transparent as possible throughout, and present results both for individual and combined categories (e.g., vector-borne and zoonotic).

Finally, we also have briefly described decomposition and the necrobiome in newly included Supplementary information 2.

Individual reviewer responses

Referee #1 (Remarks to the Author):

expertise: history of infectious diseases

"The landscape of ancient human pathogens in Eurasia from the Stone Age to historical times" screens shotgun sequencing data from 1313 ancient humans for pathogens and presents the results, along with interpretation of what the broad patterns mean for the history of infectious disease. It is the largest sample yet presented which screens for ancient pathogen DNA, and it breaks methodological ground by adapting workflows to permit sequencing and analysis at a large scale. While the findings are overinterpreted in some places, and some issues need addressing (notably the interpretation of "zoonotic pathogens" and the need for much more contextual information in places), the findings are impressive and indeed of major significance. Certain specific results are completely novel - such as finding the prominence of louse-borne relapsing fever. Despite reservations/suggestions expressed below, this paper makes significant substantive and methodological contributions.

We thank the reviewer for the kind words.

36 "the origins and past dynamics of human pathogens remain poorly understood." This is vague and understates the contributions of the last ten years from archaeogenetics, not to mention the last generation or two of skeletal anthropology and work by historians. Moreover, with very little focus on phylogenetics, this paper says almost nothing about the origins of human pathogens.

We agree with the Reviewer's comment, and have modified the part of the sentence to:

"Infectious diseases have had devastating impacts on human populations throughout history, but important questions about their origins and past dynamics remain"
(lines 44-45)

36 "the first spatiotemporal map of diverse ancient human microorganisms and parasites." This is certainly not the first such map. A google search for "atlas infectious disease" will turn up examples. Arguably, this is the first archaeogenetic-based spatiotemporal map, though it is really a large-scale screening for bacterial pathogen DNA (plus a few DNA viruses and eukaryotic parasites) for (mostly) temperate Eurasia up to some date in the last few hundred years.

We have highlighted the archaeogenetic-based aspect of our findings to clarify our approach as:

"To create the first archaeogenetic-based spatiotemporal map of ancient human pathogens..."
(lines 45-46)

50 "zoonotic pathogens, which are transmitted from living animals to humans or which have made a host jump into humans from animals in the timeframe of this study..." The paper misuses the category of a zoonotic pathogen. Here is the WHO: "a zoonosis is any disease or infection that is naturally transmissible from vertebrate animals to humans." So long as the pathogen primarily or extensively relies on non-human animals as hosts, and transmits from non-human animals to humans, it is a zoonosis.

What do the authors mean by "host jump"? If it is a sporadic infection but not an evolutionary host switch, fine - it is a zoonosis. If it is a host switch that leads to evolutionary adaptation such that humans become a primary host, and the pathogen can circulate permanently in human populations without an animal reservoir, it is no longer a zoonosis.

Also, it is not clear what is the timeframe of this study. When does it start and end? If we count as a zoonosis any pathogen that adapted to humans within the last ~25,000 years, then virtually every significant pathogen would be a zoonosis (just excepting "heirlooms" like herpes viruses).

We hope that our updated groupings described in the "Major changes" section address and clarify these points. The WHO definition of zoonoses used by the referee goes on to say "A zoonosis is an infectious disease that has jumped from a non-human animal to humans.", and cites HIV and SARS-CoV-2 as zoonoses, both of which have made a jump into the human species and subsequently circulate amongst humans (<https://www.who.int/news-room/fact-sheets/detail/zoonoses>). We therefore believe our use of zoonosis is consistent with others, and more importantly captures both historical host jumps and ongoing zoonotic and sometimes also human-to-human transmissions, which are both relevant to our historical analysis of transmission of pathogens between animals and humans. We have been explicit in how we use the term when we introduce the groupings in order to avoid confusion:

“We define zoonotic pathogens here as those transmitted from animals to humans or which made such a host jump in our sampling time frame”.
(lines 244-246)

55 reference 2 is significantly out of date

We fully agree that this is an old paper. However, it was the first paper to hypothesise that an epidemiological transition took place, and the idea has been under debate ever since. As such, we would suggest that it is appropriate to cite the original source.

65 reference 5 is a popular website; it would be more responsible to cite a scholarly source for this major claim

We have now included a new reference for this statement:

Infant and child death in the human environment of evolutionary adaptation. *Evol. Hum. Behav.* **34**, 182–192 (2013).

66 references 6 and 7 are out of date and reflect a lack of engagement with the work of historians. Dobyens' work on the impact of European arrival has long since been shown to be problematic. Also, reference 7 lacks the author/editor (Little) and has the year wrong and is now superseded by a range of important syntheses.

We have replaced these references with a more recent synthesis:

Huremović, D. Brief History of Pandemics (Pandemics Throughout History). *Psychiatry of Pandemics* 7–35 (2019).

73 reference 10 is not an appropriate citation for the point that the Neolithic revolution facilitated pathogen transmission.

We have removed the citation.

77 the paper should eliminate all references to the "first epidemiological transition." While the work of Armelagos had a major impact for a time, this construction never really became established. Sedentarization, animal domestication, urbanization, and long-distance exchange were very different processes, spread over 5-10,000 years (with different order and different timing in different regions).

We have toned down the hypothesis and stated that the idea is under debate, and refer to the reply above for our motivation to keep the reference included in the manuscript

111 the main text should give a sense of how and why these 1,313 individuals were included in the study. this is not a random sample, obviously, and seems dominated by access to Northern European archaeological samples plus Eurasian steppe samples. That's fine, of course, it's just important to be a little clearer because it seems that the contingencies of what the authors happen to have worked on have created a very particular chronological and geographical profile.

We agree with the Reviewer. As burial practices varied across cultures and time, our samples might, to varying extent, represent select groups within past societies and remains where burial practices favoured preservation of relevant material. Nevertheless, the pathogens present in these samples were

most likely widespread within each population, as social structures in relatively unhygienic living conditions are less effective in controlling the spread of diseases. Moreover, despite our efforts to obtain samples from as broad a geographical area as possible, present economic conditions, societal structures and politics create limitations in sample access in addition to those imposed by burial practices and the environment.

As the Reviewer suggests, we have highlighted that the samples are likely not random samples from a population in the text, and also noted that the detected pathogens likely were not restricted to those groups in a given population:

“As burial practices varied across cultures and time, these samples likely, to varying extents, represent select groups within past societies. Nevertheless, the detected pathogens were probably present throughout the entire population, given that diseases tend to spread easily in communities with poor sanitation and hygiene.”
(lines 119-122)

111 How many of these samples have been previously screened and how many human and/or pathogen genomes are new? In particular, we need a summary of how many new ancient pathogens have been positively identified. We learn deep in the paper that some of these are confirming previous findings.

With a few exceptions, the majority of the ancient pathogens identified in this study constitute novel results. We have now added references in Supplementary Table S2 for previously published findings, and provide a breakdown of the number of novel hits in each category in Extended Data Fig. 6d, e.

139 here or elsewhere it needs to be clearer that these are exclusively DNA viruses

We have emphasised that we are testing for DNA viruses as per the Reviewer's suggestion

214 so are all of the hits for Y. pestis new?

Please see the answer to comment 111 above. Of the 42 cases of *Y. pestis* detected in this study, 35 have not been detected previously. The remaining are from Rasmussen et al., 2015 (6) and Damgaard et al., 2018 (1).

230 "medieval cemeteries in Denmark" = late medieval, correct? Should be clear, and give specific date ranges associated with this claim.

We have clarified now to

“Finally, 11 out of 39 cases were identified in late mediaeval and early modern period individuals (800 - 200 BP) from two cemeteries in Denmark (Aalborg, Randers), highlighting the high burden of plague during this time in Europe.”
(lines 272-274)

238 the findings re: louse-borne relapsing fever are significant and striking. This is one of those important but underrated diseases. We know next to nothing of its evolutionary history and human history. This needs a discussion for context in a supplemental file, with more context about what is

known of the phylogeny, evolution, ecology, and history of the pathogen. Really, this could be a separate paper, but at least needs sufficient context here.

We agree with the reviewer that our results on LBRF are exciting. We have now expanded the descriptions of the spatial dynamics of LBRP in the results section and discuss implications of the observed patterns in more detail in the discussion. We have also added a supplementary file (Supplementary information 3) discussing the phylogeny, evolution, ecology, history and disease as suggested by the reviewer.

*262 the discussion of E coli and Salmonella needs to be much clearer
shigella spp are lineages of E coli
can we be sure that with these low ANI these are really shigella?
possible to distinguish the other strains of E coli? are these enterotoxigenic?
is the s enterica typhoidal (including paratyphoid strains) or non-typhoidal?
also, what are dates of all of these pathogens in this sample? could be very interesting.*

We have reduced the discussion of Salmonella / E. coli and Shigella results here, as the updated workflow and cutoffs result in only higher ANI / confidence hits to be reported. We still report cases of each of them in Supplementary Table S2, but unfortunately a deeper exploration of the strains is not feasible with the limited amount of data available for the individual hits.

271 Food-borne infections? This is confusing and probably misleading. Are these not the fecal-oral pathogenic strains?

We thank the reviewer for this helpful comment. Only fecal-oral route is used now.

286 the "non-finding" of V. cholerae is unsurprising and should probably not be the example used. The lack of M. tuberculosis is certainly interesting (to the extent that it is not an artefact somehow - I'm a bit skeptical). Could mention any number of major bacterial diseases that are not evidenced. Maybe most interesting of all is typhus - which has been of little interest to archaeogeneticists unfortunately!

While we agree that not finding *V. cholerae* is not surprising, we believe it is still valid to mention as it is a human epidemic pathogen of major importance. As for *M. tuberculosis*, to our knowledge all known ancient genomes to date have been isolated from mummified tissue, with one recent study also having success with bones in skeletons with lesions (Vagene et al 2022). In immunocompetent tuberculosis patients, the *M. tuberculosis* load in blood is low and notoriously difficult to test unless the patients are in an advanced disease state. Consequently, detecting *M. tuberculosis* in ancient remains becomes even more challenging. As the vast majority of our samples either constitute petrous bones or teeth, we don't believe this to be an artifact, but rather a limitation in these sample materials for preserving *M. tuberculosis* DNA in sufficient amounts for detection, even in the presence of infection.

330 why no previous mention of diphtheria. is this the oldest reported case? would be very interesting if it's high-confidence.

The diphtheria cases are now mentioned in the novel findings section. Our oldest case is from the Mesolithic, approximately 11,000 years ago, and we have been unable to find older published cases; consequently, this appears to be the oldest reported case.

We have now highlighted this in the main text:

We also report two cases of *Corynebacterium diphtheriae*, the causative agent of diphtheria, the earlier of which dates back to the Mesolithic before 10,000 years ago and appears to be the oldest reported case (Sidelkino, 11,336-11,181 cal. BP) (lines 328-330)

348-9 the patterns of plague are significant, but more detail is needed to engender confidence. for instance, can we exclude that the high incidence of Y pestis from 5000 bp to 2800 bp (the paper is a bit unclear on the window), followed by a hiatus, is not influenced by sampling, at least in part? for instance, what would the incidence be in Europe and in central Asia, separately, for 1000-year time windows? Plague reappears around 2000 bp where - central Asia? Are these strains ancestral to the first pandemic? Then, does the subsequent peak really represent the first pandemic as we know it (from ca. 541-750 CE)? Is the following hiatus really 800 years? If so, that implies that the hiatus started around the time the first pandemic should launching. This whole discussion is frustratingly opaque.

The reviewer is correct that plague reappears in central Asia (Kyrgyzstan and Kazakhstan) approximately 2000 BP. The peak around the 541-750 pandemic represents a new plague case from Sweden (VK522) and previously published cases (as seen in figure 4). The reviewer is correct in stating that an 800-year hiatus between the first (541-750 CE) and second (1346-1353 CE) is incorrect - we have now changed it to ‘~600 years’. We have modified the text in the main manuscript to clarify the discussion:

“Starting at ~2,000 BP, plague reappeared in our dataset (DA92, DA101, DA104, Kazakhstan and Kyrgyzstan, Central Asia; Fig. 3; Supplementary table S2), including the period before the first historically documented plague pandemic (~1500-1300 BP, VK522, Sweden and previously published cases). Another hiatus of ~600 years led to a rise and peak associated with the second plague pandemic ~600 BP (European late mediaeval cases, Denmark and previously published cases; Fig. 4)” (lines 409-412)

364 again, more detail on the spatial dynamics of LBRP would be helpful.

We have now also provided more detail on the spatial dynamics of LBRP in the main manuscript to clarify the patterns (lines 420-441).

381 checking the classifications in S3, why is B. quintana zoonotic - is it not a human pathogen?

B. quintana is now removed from the supplementary table as we now only include species that we have detected.

B. recurrentis is not a zoonotic disease. It is transmitted from human to human. If the authors have evidence for the adaption from an animal to a human host during this time, they should present it. But other pathogens they categorize as anthroponotic also adapted to humans in this time frame (smallpox, TB, for sure, and probably others).

The reviewer is correct, *B. recurrentis* is transmitted from human to human through human body lice. We have changed the classification to vector-borne diseases.

R. prowazekii is not a zoonosis. It does circulate in squirrel populations in the Eastern US but only a handful of cases have transmitted from animal to human. It is simply a human disease, but we have absolutely no idea how it adapted to humans or when.

R. prowazekii is now removed from the supplementary table as we now only include species that we have detected.

R. typhi, *S. enterica* (if typhoidal), etc. are not zoonoses.

R. typhi is now removed from the supplementary table as we now only include species that we have detected. For *S. enterica*, we have cautiously re-classified the species group as “environment_pathogen”, as we don’t have enough resolution to distinguish different serovars and cannot rule out an ancient environmental source rather than an infection.

The four malaria parasites are not zoonoses. Their ancestors were primate pathogens. *P. vivax* was a human pathogen already in the Pleistocene. *P. falciparum* probably did adapt to human hosts in the last few thousand years, but this is debated.

We have now changed the classification for the malaria parasites to vector-borne diseases.

393 reference 54 is inappropriate - out of date and perhaps the source of the paper's confusion about transmission patterns and evolutionary history.

We have removed this reference in the revised version.

400 given the confusion and deep inconsistency about what counts as a zoonotic pathogen, it is impossible to tell if the claims about the patterns of zoonoses can be salvaged. Even so, one would want stronger tests of robustness against preservation and sampling biases (e.g. are most of the pre-6500 ybp hits not potentially from environmental sources?).

We agree that differential preservation and sampling bias is a major concern. In our updated analysis, we have significantly expanded the set of species tested, in particular now with many more species included that form part of the oral microbiome. We found high detection rates for those species throughout the whole sampling period. If pre-6,500 ybp data would mostly represent environmental sources, we would have expected preservation of endogenous oral microbiome species also to significantly drop, which is not what we observe.

Once LBRP is removed, I think most of the zoonoses will be *Y. pestis* and leptospirosis. So the claim then perhaps becomes about *Y. pestis* and leptospirosis, not zoonoses.

Please see the section on “Major changes”.

SI it would be helpful if there was a sheet (SI?) that allowed you to see which archaeological samples had which pathogen species - it is hard to flip back and forth.

We have now included relevant sample-level metadata (age, location) to the main table showing all ancient microbial hits (Supplementary Table S2). We hope this addition will help in better parsing sample-pathogen links.

S1 what is Barrie 2023?

Correctly referenced now as Barrie et al., Elevated genetic risk for multiple sclerosis emerged in steppe pastoralist populations, Nature, doi: 10.1038/s41586-023-06618-z.

S3 viruses are not a domain, so maybe another term?

We have used the generic term “microbial type” instead now throughout.

S3 In any case, we can agree that the smallpox virus is not a bacterium.

We fully agree with the reviewer that smallpox is caused by a virus and apologise for the typo and confusion in the text.

S1 make clear that these dates are years BP (1950?)

Did I miss the archaeology supplement? I sincerely apologize if so. If not, where is it? Detailed site context and individual grave context should be standard practice in archaeogenetics. We don't know, for instance, if none/some/many/most of these samples are from contextually ordinary burials. We don't really know if the dating is good. I assume "direct" dating is radiocarbon? Are the upper and lower ages 2-sigma?

We have now updated Supplementary Table S1 including extended metadata on dating for each sample. As the samples in this study have been published as part of previous studies, we now refer to the original publication in the supplementary table for the detailed site and sample context descriptions. For the small number of samples that have not been included in the previous publications, we provide a new supplementary text (Supplementary information 1) containing brief site descriptions.

Referee #2 (Remarks to the Author):

expertise: ancient pathogen genomics

###Overview:

This is a remarkably impressive effort to screen and report the microbial/pathogen content found in the off-target reads of 1,313 ancient human remains covering 35,000 years of Eurasian history. While a growing field, ancient pathogen genomics has tended to focus on specific species, reconstructing their histories. A more holistic assessment of disease burden through time has been lacking and major uncertainties surround how the human experience of infectious disease has changed through time. The authors provide results that challenge the strongly held view that zoonotic diseases took off during the Neolithic, instead suggesting that while zoonotic pathogens may be linked to the domestication of animals it isn't until the period of Steppe migrations that zoonotic pathogens are detected to increase in prevalence (detection), hence supporting a wider role of human migrations (approximated by Steppe ancestry) in the distribution of modern pathogens.

The intention to create the first spatiotemporal map of human infections is ambitious, innovative and exciting and the authors should be commended for making all fastq reads available, including well annotated code and metadata. I have no doubt this project will spur further research.

We thank the reviewer for the enthusiastic response.

While I am excited by this project, I do have some concerns over the interpretations. Some very strong claims are made, and I feel quite a bit more could be done to discuss the potential limitations of the input data and resulting insights. Even addressing these questions using modern day metagenomes (which would be an interesting comparator) would be fraught with the challenge of accounting for sampling bias and research effort. Furthermore, the classification of microorganisms into pathogenic or non-pathogenic or zoonotic or opportunistic etc. is non-trivial.

I outline my comments below.

###Major comments:

The authors construct a database of 44 'human pathogens'. Little attention is given to how this was devised. The status of a species as a human pathogen or not is often not that clear cut, particularly for species where the life histories through time and reservoir hosts are unknown or poorly characterised. What do the authors consider as a 'pathogen'? Databases do exist of 'human pathogens' – were these considered? Eg. work from Bartlett et al. discuss and release a much larger database of 1513 bacterial pathogens <https://www.microbiologyresearch.org/docserver/fulltext/micro/168/12/mic001269.pdf?expires=1695037757&id=id&accname=sgid025015&checksum=EE127AD22A68080B022BA26644ABBEA0>. The selection of genera to be considered as pathogens may well colour resulting observations. Similarly, the classification into zoonotic, anthroponotic, sapronotic etc is non-trivial but not explained. There certainly may be anthroponotic species which are in fact zoonotic, though the animal reservoirs have been poorly characterised. Eg. some argue HBV is not anthroponotic <https://www.wjgnet.com/1007-9327/full/v20/i24/7665.htm>. I would recommend carefully evaluating host-pathogen association databases eg. CLOVER to provide justification for some of the classifications. This is very important

because the major observation that zoonotic pathogens are not detected before 6,500BP is based entirely on how the authors chose to classify their detected species.

We thank the reviewer for the very helpful suggestion to use the work by Bartlett et al. to classify human pathogens. We hope that the reviewer's concerns are now adequately addressed as outlined in the section "Major changes to the manuscript".

aDNA has amazing potential to provide unexpected insights. I have no doubt that the distribution of infectious diseases was likely different in the past, but some examples in text do need additional discussion. An example is the claimed IIV-31 human infection. The authors comment that this is a virus of woodlouse but without citation, and it does seem like a virus of invertebrates rather than one you would find in human teeth. Does the virus receptor allow entry to vertebrate cells? Can we believe this? In addition, other unexpected observations such as Proteus virus Isfahan may be genuine but this species has also implicated as a possible contaminant in DNA extraction kits/lab environments. Are the damage profiles strong in this case? These are not the only unexpected observations. Amongst the provided supplementary tables there are hits to citrus plant viruses, cattle parasites, and fish pathogens. How confident can we be that the results presented relate to genuine ancient human infections? This does bring into question some of the observations and these should be discussed with more care.

We agree with the reviewer that many of the hits we report are not representing actual human infections. We have attempted to clarify this throughout the manuscript and with our update species grouping by likely source of the DNA (section "Classification of microbial species" above). Taking the mentioned *Proteus virus Isfahan* as example, in the revised version we have one highly confident hit (VK139), with 5,265 reads aligned (~5.4X genomic coverage) and strong evidence for aDNA damage (metaDMG estimated deamination rate $D_{max} = 0.15$, Z-score 10.3; plot pasted below).

As part of the publication we are making damage plots and other screening pipeline results available as supplementary files in a Zenodo public repository, so each individual hit can be evaluated by the interested reader.

A high % assignment to toxoplasma is evident across the dataset. This is likely due to eukaryotic contamination in the reference database rather than toxoplasma being present at such a high frequency. I would be concerned, particularly for parasite species, that contamination in the RefSeq

reference genomes or eg. retained adapter content etc. could lead to spurious hits and inflated prevalence estimates. What steps were taken to mitigate for this?

We agree with the reviewer that *Toxoplasma* hits can be caused by database contamination, in fact we are presenting them as examples of contamination, and how our authentication approach guards against such false positive results (e.g., coverage evenness statistics, Extended Data Fig. 4). As a consequence, our final authenticated results do not contain any hits for *Toxoplasma* (Supplementary Table S2).

*Previously published datasets are not really considered and observational data (place and time) is readily available for many human pathogens from eg. the SPAMM initiative <https://github.com/spaam-community/AncientMetagenomeDir>. While I appreciate exhaustive inclusion is a big ask, I think for *Y. pestis* and LBRF (I note this is only one previously published case) these should be considered in the assessment of temporal dynamics. Certainly for *Y. pestis* this would add a good amount of data. If this data can't be included given the need for a 'rate' of detection then could some of the claims be validated given published datasets. Eg. is it true that there is an 800 year period of no published observations in published *Y. pestis* data also? And why does an absence of plague support a prevalence of LBRF....? Are the authors claiming some cross-immunity or the such like? This feels a strong statement without further expansion.*

We have now added the temporal distribution of all published *Y. pestis* and *B. recurrentis* (e.g., a single genome) from the SPAAM AncientMetagenomeDir to Figure 4. Encouragingly, both periods without noticeable *Y. pestis* detection in our dataset are also evident when including all published ancient genomes. We do not claim that there is any cross-immunity between *Y. pestis* and *B. recurrentis* as this seems very unlikely. We think a more plausible explanation could be related to population density and living conditions. Due to the high mortality during plague epidemics, the population size will decrease, and living conditions will be less crowded. Moreover, it is possible that the risk of plague would cause dispersals of some part of the population within larger settlements/cities to more sparsely populated areas, resulting in a further decrease in population density. Increased efforts to improve hygiene might also occur. Together, these factors and other unknown behavioural and societal adjustments to a plague epidemic, likely decrease the transmission of human body lice and, consequently, the risk of LBRF. Moreover, a recent study has demonstrated that the body louse can be infected by *Y. pestis* in the midgut (like *B. recurrentis*) and in a putative salivary gland, the Pawlowsky gland (Bland et al. PLoS Biol 2024; <https://doi.org/10.1371/journal.pbio.3002625>). Thus, transmission could occur through bacteria-containing louse faeces in addition to when the louse feeds on the human host. It was hypothesised that infected body lice might have fueled the outbreaks initiated by infected rat-rat flea-human transmission. Infected body lice have more than six-fold higher mortality than uninfected body lice, and it is unknown to what extent, if at all, co-infection of body lice with *Y. pestis* and *B. recurrentis* is possible. If a large proportion of body lice were infected by one or the other bacteria, body lice might not have been available as a transmission vector for the other.

We have now added these points to the main text:

*“This opposing pattern is unlikely to be due to any cross-immunity between *Y. pestis* and *B. recurrentis* but could plausibly, in part, be caused by population size decreases and behavioral and societal adjustments during plague epidemics. Moreover, a recent study has demonstrated that *Y. pestis* can infect body lice in the midgut, like *B. recurrentis*, and in some, also the putative salivary*

gland, the Pawlowsky glands (PG). Body lice infected in the PG were experimentally shown to transmit *Y. pestis* in concentrations sufficient to initiate disease in humans, and it was suggested that these possibly contributed to transmission during plague outbreaks. Infected body lice have more than six-fold higher mortality than uninfected body lice, and it is unknown to what extent, if at all, co-infection of body lice with *Y. pestis* and *B. recurrentis* is possible. If a large proportion of body lice were infected by one or the other bacteria, body lice might not have been available as a transmission vector for the other, which would help explain our observation.” (lines 440-450).

The authors should be careful about overstating what can be gleaned from this dataset. It is of course very impressive in scale but even the best efforts are likely to be only a partial proxy to “infectious disease load over time”. There are a number of potential issues with using this data to generalise to all human infectious disease. This includes that the data is Eurasian centric rather than global, not all species are considered, the work is restricted to DNA pathogens, there is differential preservation and survival of different microbial species (each species likely has a different LOD and some are very unlikely to be found despite being highly significant eg. Mtb). Perhaps some species, suggested as prevalent in the past, have high asymptomatic rates and are under sampled and studied today meaning their current distribution is poorly known. No systematic screen has been conducted of contemporary metagenomes so for many pathogens it is difficult to comment on their true distribution and prevalence today. These points should be more carefully discussed.

We have attempted to qualify that our results are Eurasian-centric throughout the manuscript, and clarified that only DNA pathogens are targeted.

###Minor comments:

Introduction: The statement ‘the burden of infectious diseases has also left lasting impressions on human genomes, where the selective pressure exerted by pathogens has continuously shaped human genetic variation’ feels too strong. In fact, it could be argued that given the high postulated impact of infectious diseases in human history, efforts to identify confident susceptibility loci in human populations are surprisingly unproductive, aside from archetypal cases such as Plasmodium. The cited study (Karlsson et al. Nat Rev Gen 2014) has very few compelling examples for instance. I would modify this statement.

In light of more recent evidence from ancient DNA (e.g., Barrie et al 2024, Kerner et al 2023) we believe this statement is still valid, and we have updated the citation.

The introduction comments on 1313 individuals but only 896 samples are discussed in the abstract. 1272 individuals are presented in extended data 1. Could these discrepancies be clarified?

The 896 samples in the abstract were the number of individuals with an ancient pathogen hit (1,005 in the updated analysis). Supplementary Table S1 provides the metadata for all 1,313 study samples.

Non-human read classification is highly variable and implies marked differences in endogenous content. To contextualise, could the distribution be given for the % of unassigned reads in the text?

We would like to thank the reviewer for this helpful suggestion. We have now added:

“In the initial metagenomic classification step, the fraction of reads classified as originating from non-human taxa ranged from 1.1% to 76.5% per sample, with a median of 3.5% (non-classified reads 6.1%-98.1%; median 62.2%).”
(lines 122-124)

For the tooth associated metagenomes there is a mixture of the metagenome being modelled as 2 or 3 components – the authors stating the latter relating to the abundance of co-occurring oral microbiome species. Is there anything different about the sampling of these teeth? For example, use of root or calculus etc. Does sampling lead to a difference in microbial profile given the biota of teeth is likely highly heterogenous. Are there cases where different teeth have been sampled from the same individual? How similar or different are the resulting taxa profiles?

The data used in this study has been generated as part of previous studies on human population history, spanning almost 10 years. While some variation in sampling technique is to be expected during this timeframe, the majority of the teeth were sampled to maximise preservation of human DNA, using the cementum layer. No calculus has been included in this study. We fully expect microbial profiles to somewhat differ between samples, which is why we restrict our analysis to the simple models with few components presented here. A deeper exploration of this is out of scope for this study and will be addressed in future work.

Were fungal species considered? If not, why not?

While fungal species are included in the database used for initial metagenomic classification, we have not included them in the targeted genera list for the mapping and authentication stage for this study as our scope was bacterial, viral, and protozoan pathogens.

How should ‘strict’ and ‘lenient’ metaDMG be interpreted?

These categories have been removed in the revised version.

It seems there is quite a lot of scope to comment on the necrobiome or shared environmental microbiota across this dataset but this is not discussed. A characterisation of these components could be of high relevance to distinguishing signatures of pathogenic infection from commensal colonisation. Did the authors consider this or can they comment?

We have attempted to incorporate these distinctions in the revised classification of species, and also more extensively comment on the possible necrobiotic origin of some of the hits we observe (see “Major changes” section above). Moreover, we have briefly described decomposition and the necrobiome in newly included Supplementary Information 2.

More nuance is needed in places. For example if anything, given the need for high infectious loads, tropism to bone and reduced degradation rates, co-infections are likely underestimated in this dataset rather than “readily detected”.

We thank the reviewer for this helpful comment and we have modified the main text accordingly:
“Overall, our results show that co-infections can be detected using ancient metagenomic screening, but are likely underestimated given methodological limitations such as differences in pathogen load, tissue availability and other factors impacting detectability of ancient microbial DNA.”

(lines 391-394)

Some strong claims are made without accompanying evidence eg. that the processes supporting zoonotic bacteria are the same for zoonotic RNA viruses.

We have modified the text and inserted a reference to a study that supports this suggestion:
“As our dataset is lacking information on RNA viruses, our zoonotic disease analyses underestimate the overall zoonotic disease burden; however, the timing is likely accurate as the conditions favouring zoonotic transmission of RNA viruses are similar to those of other zoonotic pathogens, e.g., an increase in the interaction between species and shared living spaces”.
(lines 527-531)

Fig 5 – the ‘other’ category shows the highest detection rate across time. What are these? Contaminants? Commensals? Do they share similar properties.

Categories have now been revised, please see above.

Referee #3 (Remarks to the Author):

expertise: metagenomics

*Sikora Willerslev and colleagues screened shotgun sequencing data from 1,313 ancient human remains, covering 35,000 years of Eurasian history, for ancient DNA derived from bacteria, viruses, and parasites. The manuscript reports identifying from 2,400 individual microbial species hits in 896 samples including the food borne pathogens *Yersinia enterocolitica*, the animal-borne *Leptospira interrogans*, malaria-causing parasite *Plasmodium vivax*, *Yersinia pestis*, the causative agent of plague, Hepatitis B virus, and *Borrelia recurrentis*, the cause of louse-borne relapsing fever (LBRF). They conclude that their paleoepidemiologic analyses document an increased burden of zoonotic infectious diseases following the domestication of animals.*

Major issue. Sample contamination.

Given the number of papers that have been called into question due to possible sample or reagent contamination – including by the authors of this paper – I would have expected more explicit analysis to control for contamination. These samples have all the issues that have made other studies be published to great fanfare only to fall into question when data is examined by the scientific genomic community. The authors are re-analyzing publicly available data that was generated to examine human evolution. As such great care was taken in the original studies to ensure that there was no contamination with current human DNA. However that is a very different process than what one would embark upon to characterize microbial DNA. There are no negative controls for these samples such as soil DNA or reagent sequencing.

We agree that contamination of any kind is a major concern for any study of microbial DNA. We have taken a very stringent approach in authenticating our ancient microbial hits, described much more prominently now in the revised manuscript and in the section “Improved authentication” outlining the major changes to the manuscript. In Extended Data Fig. 3c we now provide a simple overview of the different kinds of “contamination” that can occur, and how our authentication metrics would be affected by them. An advantage of using aDNA over modern metagenomics is that ruling out any modern contamination e.g., from reagents, is facilitated by the characteristic aDNA damage patterns used in the authentication. Other issues such as database contamination or ancient DNA from an environmental source present more challenging obstacles. In this revised version, we have expanded our discussion of authentication statistics and present new results from simulated datasets showing that the metrics implemented facilitate stringent authentication.

Major issue. Specificity of ‘hits’

The ‘hits’ explored by the authors are often as little as 20 reads. Extremely little detail is provided to understand how the authors make the claim that these are ‘hits’ rather than random reads from another microbial species that has a weak hit to a pathogen in the database.

Please see the reply above and the extended description of the authentication approach. We also note and re-emphasise that our screening workflow considers all reference genomes of a particular genus in the read mapping step, irrespectively of pathogen status. We do observe a low level of possible “cross-mapping” across genera in cases of a highly abundant ancient microbial source from a closely related species in our simulations, and have added a caveat to the interpretation of low ANI / read number species hits in the manuscript now:

“As analysis of real data additionally includes ancient DNA damage patterns for authentication, such false positive results are thus only expected when a closely related ancient microbial species is present in the sample at high abundance. We nevertheless suggest caution in interpreting individual ancient species hits with low ANI and read numbers, particularly in scenarios of species where all hits show these characteristics.”

(lines 163-167)

The authors did a comprehensive analysis on the ancient human bone and tooth samples but did not report the results for controls (e.g., corresponding soil samples). Since the biomass of microbial DNA in ancient human samples is low and the positive hits based on read mapping are sometimes based on as few as 20 reads, the authors need to demonstrate that those hits were not false positives or sourced from contaminations.

Please see the replies above. Corresponding soil samples, while useful for future studies, are unfortunately not available for these samples. We fully agree that many of our hits derive from environmental sources, and these have been explicitly labelled as such in the revised manuscript.

In line 196, authors further concluded that lower ANI and genome mixture suggested environmental sources. However, multiple strains can be present on one individual. The issue is complicated by the approach taken by the authors when identifying “a single best-matching species hit for each sample and genus of interest”, meaning that the multiple alleles could also be due to a mixture of species. Authors should address the effects of multiple species/strains on their results.

We have clarified this issue in the revised version, stating that:

“Alternatively, ANI can also be reduced when reads mapped to a particular reference assembly originate from multiple closely related strains or species present in a sample.”

(lines 198-199)

‘Identifying eukaryotic pathogens is challenging, as sequence contamination from other organisms 291 regularly occurs in reference genomes deposited in public databases’ Seems a little insincere to reference Salzberg’s landmark study at this point in the manuscript given the arguments Steinegger and Salzberg are making about all of RefSeq.

While Steinegger and Salzberg did analyse all of RefSeq, they do state in their article that “over 95 % of contamination occurred in eukaryotic genomes. Eukaryotic genomes tend to be much more fragmented due to their larger genome sizes and higher repetitive content (as compared to prokaryotes), and many of the smaller contigs in eukaryotic genome assemblies suffer from contamination”.

We have further clarified this issue by stating:

“Identifying eukaryotic pathogens is challenging, as sequence contamination from other organisms frequently occurs in their large and often fragmented reference genomes deposited in public databases”.

(lines 338-340)

Now suddenly authors evoke statistics such as ‘low support from coverage evenness statistics due to the ancient reads 294 predominantly mapping to short contigs representing contamination’ but were these statistics used for bacterial pathogen read mapping??

All authentication statistics form the major part of our approach and are used throughout for all species. We use *Toxoplasma gondii* as an empirical example of how our statistics detect such cases (see also the related reply to reviewer 2 above).

'we identified two cases of infections with the malaria-causing parasite Plasmodium vivax, ' what constitutes a 'case'? is this still 20 reads?

Read numbers and other authentication statistics are presented in Supplementary Table S2. In the revised version, we report five authenticated cases of *P. vivax*, with final mapped read numbers ranging between 70 and 1,710 reads.

Overall detection rates of plague were interesting but by this point I was so concerned about the data quality. For example Borrelia recurrentis rate is 0.05% but remember from Fig 2A these are mostly lenient DMG.

The labelled damage categories have been removed in the revised manuscript; we now report all hits with $Z > 1.5$ and provide three Z-score ranges for ease of inspection in the figures and Supplementary Table S2. Taking the example of *Borrelia recurrentis*, in our updated analysis we now report 34 cases, 24 of which show damage evidence with $Z > 2$.

Out of 82 identified putative hits (the majority of 266 which were Escherichia coli; 55 cases, 4.1% of samples), 27 were found as co-occurring hits in a set of 267 12 individuals (Extended Data Fig. 7).

Do any of these match lab strains of E. coli? That would be a red flag for me.

Escherichia coli; 55 cases, 4.1% of samples

E. coli is a minor component of healthy gut community and there is no reason one would expect it to be in bones so maybe this is more a concern about level of sample contamination?

In the revised analysis with more stringent ANI cutoff the number of cases of *E. coli* is reduced to 21 (Supplementary Table S2). The majority of these has strong evidence for aDNA damage ($Z > 3$) and are thus highly likely ancient *E. coli* reads. As *E. coli* is also common in the environment, and we lack the ability to resolve strain-level information with low-coverage ancient DNA, we refrain from labelling those cases as infections and group them cautiously in the “environmental” category, even though some of them could reasonably represent cases of invasive infections

Major issue. Unmapped reads.

The unmapped reads are a vast percentage but not addressed.

We have added summary statistics for numbers of unclassified reads to the manuscript (line 124). We agree with the reviewer that there is much potential for interesting findings in the “microbial dark matter” also with regards to ancient microbes, but this is unfortunately out of scope for this work.

'The highest fraction of reads classified include soil-dwelling taxa such as Streptomyces and 116 Bradyrhizobium or taxa with broad ecological distribution, such as Pseudomonas, reflecting a 117 predominantly environmental source of microbial DNA in the samples.' Bradyrhizobium can also be found in water supplies as a contaminant so could come from reagents which means that other microbial contaminants are also there.

We agree with the reviewer in that many species found in the initial metagenomic classification step are likely of modern origin, whether they originate from soil or reagents. We do not expect those to pass the subsequent read mapping and authentication stages as outlined above.

Additional overall issue. Microbiology of Death.

Metcalf has performed experiments and analyses to evaluate the 'microbiology of death'. This work should be read by the authors and referenced. It will change the ways in which they make assumptive statements such as 'highest relative abundance 124 of genera such as Actinomyces, Streptococcus or Pseudopropionibacterium, commonly associated with 125 the human oral microbiome. This component was observed at noticeable frequencies in many samples 126 derived from teeth, but only rarely in those from other tissues (Extended Data Fig. 1e, f). This suggests 127 that a substantial fraction of the identifiable microbial DNA found in many ancient tooth samples derives 128 from the endogenous oral microbiome of the human host.'

We thank the reviewer for the very helpful comment and have read the suggested studies by Metcalf *et al.* and others in the field. We have expanded our discussion on possible necrobiotic sources of our hits in the text and added the appropriate references. We fully agree that a possibly significant fraction of the ancient microbial hits could be derived from the necrobiome. However, the broad detection of oral microbiome-related species and their strong association with teeth as sample materials makes it in our view extremely unlikely that a substantial fraction of them would not originate from the endogenous microbiome of the sample, even if involved in decay processes after death. We have also briefly described decomposition and the necrobiome in new Supplementary Information 2.

Specific comments on figures.

Figure 1 toy example of data workflow shows an idealized version. What are the actual numbers is more important; e.g. what % unmapped reads?; when is there 5x coverage as depicted in species A1 reference? Digging into the data this seems like highly idealized data. Would be much better to show realistic data and accurate data flow.

We respectfully disagree with the reviewer. In our view the illustration should highlight the approaches taken in the workflow in an intuitive manner. Taking the coverage as example, reducing the "cartoon coverage" in the figure would be detrimental to the ability to visually distinguish the differences in evenness of coverage between reference A1 and A2, which forms a key component in the authentication.

Fig 2. DMG never defined but used as key to legend? What does DMG mean? Seems central to argument but also seems like a term the authors made up? DMG = DNA damage microbial genome??

DMG in the previous version referred to the characteristic ancient DNA damage patterns used for authentication, but the terms are no longer used in the revised version.

High ANI strict DMG should be shown on the bottom of the bar chart if these are the higher confidence data reads to facilitate making comparisons between genera rather than method used by authors. This would for example show that mycobacterium has few high ANI samples. Same for

Bordetella. Since both of these genera become main stories, the data is misleading in presentation. By comparison *Yersinia* and *Staph* are both high ANI.

The ANI cutoff has been increased in the revised version to only include hits previously in the “high ANI” category.

Figure 2. what is a ‘hit’ that is being counted in 2A?

A “hit” constitutes a unique sample / authenticated microbial species combination. This has now been further clarified in the manuscript:

“Based on the results of the simulations, we selected the species with the highest number of unique *k*-mers assigned by *KrakenUniq* among the set of species with $n \geq 20$ reads mapped and passing all authentication criteria (Methods) as the putative ancient microbial species “hit”, for each sample and target genus. A particular sample can thus have multiple species hits across different genera, but for each individual genus at most a single species hit is reported. A limitation of this approach is therefore that potential other true positive ancient microbial hits deriving from a different species within the same genus will be missed. Importantly however, within each genus all species irrespective of their pathogen status are included in the read mapping and can result in a hit, thus avoiding potential bias that could occur if only specific pathogen species were targeted”.

(lines 169-177)

Fig 2d. really low confidence data is being shown. Total of 115 reads and read depth of either >1 or >2 . Can't really tell from legend. What does it really mean to look for multi-allele rates with $1X$ depth?? sample with higher multi-allele rate ANI is low.

This figure has now been moved to Extended Data Fig. 6c. We have updated this analysis by quantifying the rate of observing different alleles in two randomly sampled reads across all sites with at least $2X$ coverage in the genome, to avoid issues with differing sequencing depth:

“To test for the presence of such mixtures, we quantified the rate of observing different alleles at two randomly sampled reads at nucleotide positions across the genomes of hits with read depth $\geq 1X$ ”.

(lines 199-201)

Methods.

‘We identified 39 cases of Y. pestis’ based on what criteria??

The authentication criteria and respective cutoffs are provided in the methods section, “Filtering of putative ancient microbial hits” (lines 652-684).

Same question for ‘The leprosy-causing bacterium

Please see above.

248 Mycobacterium leprae was identified in eight individuals (0.6% detection rate), it's not that 0.6% of the reads matched M. leprae. It's that 0.6% of aDNA samples had at least 20 reads that matched M. leprae with low ANI and low aDMG.

Correct, the detection rate corresponds to the fraction of individuals that had an authenticated species hit to *Mycobacterium leprae*. Reads mapping to *M. leprae* ranged from 297 to 121,450 (2.2X average read depth), and showed overall very high ANI (>99% for all hits, see Fig. 2b and Supplementary Table S2) and strong evidence for aDNA damage (all but one had $Z > 3$). *M. leprae* constitutes an illustrative example, as we only detect it within recent time periods in Europe, consistent with previous results. The relative abundance of *M. leprae* among all non-human reads was also some of the highest observed in our datasets (Fig. 2c), consistent with the high bacterial loads of infections.

Figure 6. what is the 'sequencing' feature used in Fig 6 as a co-variate?

The number of human-classified reads, used as a proxy for aDNA preservation and detectability, described in the manuscript as:

“We modelled the presence/absence of either individual microbial species or combined species groups using sets of putative covariates, including spatiotemporal variables (longitude, latitude, and sample age), paleoclimatic variables (mean annual temperature and precipitation), human mobility and ancestry, sample material (tooth or other), as well as a proxy for “detectability” (number of human-classified reads).”
(lines 485-489)

Methods. Dataset is not described more than '515 We compiled a dataset of ancient DNA shotgun-sequencing data from 1,313 ancient individuals, of which 516 130 are newly reported in this study.' Should at least provide BioProject, BioSample IDs for samples used in study. These 130 new samples – how were they treated in this study? Were appropriate controls collected for them to monitor where microbial contamination occurs in the process?

The bulk of samples were previously published for human DNA, and references for the original studies are provided in Supplementary Table S2. We further included data for 68 samples which were sequenced as part of the previous studies, but not included in the final publications. As such they were treated according to the same strict ancient DNA laboratory standards as the other samples. As part of this study, we will release sequencing reads for all samples in FASTQ format at ENA under the provided accession numbers.

We selected the 527 assembly with the most unique k-mers assigned as the representative reference genome for each species in 528 a particular sample. If no reads were assigned to any assembly of the species in KrakenUniq, we selected 529 the first assembly for mapping.'
Does this mean that even if only 1 or 2 reads were assigned to a species, this was used?

In the mapping step, each species of a particular genus is used in parallel as a reference for mapping for the same set of sequencing reads initially assigned at that genus level. If the reads could further be assigned at the level of individual assemblies within a species, we use the best-matching assembly as the reference genome. If no reads could be assigned at the assembly level, we chose the first one. The motivation is that in cases where different species are highly similar (e.g. *Yersinia pseudotuberculosis* complex or *Mycobacterium tuberculosis* complex), few, if any, reads might be assigned to the species level by *krakenUniq*, but rather at the higher taxonomic rank (e.g., *Yersinia pseudotuberculosis* complex). Mapping all genus-level assigned reads to each reference and comparing mapping statistics between them ensures that we also capture cases in those scenarios.

As this is a major part of this article, some additional explanation should be provided: 'Authentication of ancient microbial DNA 535 To authenticate ancient microbial DNA, we calculated sets of summary statistics quantifying expected 536 molecular characteristics of true positive ancient microbial DNA hits 63:' in addition to referencing a review written 7 years ago. How was sequencing done on different platforms at different institutes controlled for?

We have now provided an expanded description of the authentication statistics. Despite these reviews being a few years old, they describe authentication statistics widely used today. All samples used were sequenced on Illumina HiSeq platforms at GeoGenetics in Copenhagen. While we cannot rule out subtle platform biases over the ~10 year time span covering the data, we have no reason to expect them to have a differential impact on the detection rates of individual microbial species that would alter our results.

e.g. putative microbial "hits" includes 605 - Number of mapped reads ≥ 20 . Mapped with what ANI over what length? How long are these reads? What k-mer size is used?

We only include reads of length 30 bp or longer, which is the established cutoff in aDNA studies (methods section "Ancient microbial DNA screening", line 580)

Minor comment. In line 399, authors claimed that the "sapronotic" class occurred "at lower detection rates (mostly below 10%)." However, in Fig 5a, the "sapronotic" class seems to be at relatively higher detection rate and certainly not below 10%. Please clarify such inconsistency.

Please see the comments at the beginning on the new classification.

Question. What about AMR genes? Were these identified in any of the samples?

Identification of AMR genes would be an interesting avenue to pursue. However, as the majority of our data is low coverage, identification of individual genes would be very challenging. We thus did not include such an attempt in the scope of our study.

Referee #4 (Remarks to the Author):

expertise: infectious disease epidemiology

** The landscape of ancient human pathogens in Eurasia from the Stone Age to historical times*

** Downloads/ancientPathogensREV.pdf*

This paper represents an important and interesting contribution. The analyses are useful. The authors are commended for sharing a code pipeline and being clear about availability of inputs.

We thank the reviewer for the encouraging and supportive comments.

In my opinion, the statistical explanations, and perhaps methodologies, need to be improved a little.

We have expanded the methodological descriptions and approaches in the main text in the revised version.

The authors need to discuss their Bayesian priors. Even if they simply used inlabru defaults, this should be clearly stated. Ideally, they should describe a brief plan (we planned to use defaults no matter what; we planned to follow a certain procedure to tighten priors under certain conditions; ...)

We followed the default inlabru priors, where distributions are distributed as a Gaussian variable with mean μ and precision τ (in our particular case, we did not use a spatially-varying field because we wanted to test the effect of latitude and longitude as covariates as well). The prior on the precision τ is a Gamma with parameters 1 and 0.00005. The mean is a linear combination of the covariates. By default, the prior on the intercept of the linear combination is a uniform distribution, while the priors on the coefficients are Gaussian with zero mean and precision 0.001. We have now clarified this in the Methods.

The analysis shown in F6 is inadequately described. There is no context that can be used to understand the meaning of the word “positive” at L425, or the ancestry effects in F6. What is the baseline? There is also no basis to understand what is meant by “effect size”.

The variables used as predictors in the models were standardised, so if we look at ancestry, for example, the baseline would refer to the average ancestry across the whole spatiotemporal range. The results indicate deviations from the mean and the effect is in units of standard deviation. We have now clarified this in the main text.

Using WAIC to exclude variables from a Bayesian model does not seem like best practice. The authors should consider simply using their full Bayesian model, or else explain their choice.

We used the WAIC to select the model with the best posterior predictive accuracy out of the different models we tried, and prevent overfitting (because cross-validation or alternative measures would have required us to perform repeated model fits which would have been computationally costly, and problematic given the limited amount of data we have). We report the models with the best posterior predictive accuracy in the main text to avoid having to report all possible models we tried: the results from all the other models are also now reported in the Supplementary Material. For the model with the best WAIC, we then focus only on those variables whose posterior distributions lie outside of the 95% Bayesian credible interval.

The authors need to describe how they put pathogens into five classes. Did they take any steps to avoid selection biases arising from already-observed patterns when they did so?

We have now updated the pathogen classes in the revised version as outlined in the “Major changes” section. The groupings are now based on likely source of the DNA, and include pathogen/non-pathogen information based on the published literature (Bartlett et al 2022)

MINOR COMMENTS

LBRF is spelled two different ways on L45

We have corrected the typo and now standardised the spelling of LBRF.

Qualifying word needed on L47 (e.g., “inferred” form of transmission)

Thank you for the helpful suggestion; we have now added ‘inferred’ into L47.

L80 “small number” is not ideal, maybe “small proportion”

We have corrected the text to ‘small proportion’.

L82 “pathologies” is the wrong word

We agree and have now corrected the sentence using ‘signs’.

s at L107 needs to be clarified

This part has now been updated with an extended description of the different authentication criteria and their motivation (lines 99-106).

The use of k to mean two completely different things in the topic analysis is not ideal (L120)

We have replaced the indicator for the number of components with ‘1’.

The reason for requiring damage to be present should be mentioned briefly in the workflow section (~L148), maybe also in the Intro (~L107).

This is now mentioned in the extended description of the different authentication criteria and their motivation (lines 99-106).

The use of on upper and one lower bound, and the switch between percentage and reads per million, around L155 is awkward. Maybe just do median, or else median and range. And maybe do everything as reads per million reads (0.0x% is a confusing idiom for many, I think)?

We have edited this sentence, but decided to keep the lowest abundance sample as per million to highlight the low number.

I vote against the artificial categorization in F2b. There is no need to explain “significance” or (more confusingly) “similarity” or to divide the plot into arbitrary groups. The ordering and the CIs convey what needs to be conveyed.

We don't classify hits into ANI groups anymore in the revised version, and have now changed the “significance” groups into colour ranges with explicit Z-score ranges.

“such as” is a tiny bit confusing in L180

We have added ‘such as for example’ to avoid confusion.

Authors should briefly explain what is meant by “cal. BP”, and also the associated tags (e.g., NEO168). I was unable to understand the latter.

We have now made more explicit that the sample name is provided together with calibrated age in the first instance in the manuscript:

“(sample NEO168, 5,583-5,322 calibrated years before present (cal. BP))”
(line 261)

s at L287 needs to be edited for clarity

We have corrected and modified this sentence to:

“However, as the *M. tuberculosis* load in blood is generally low in immunocompetent patients without advanced disease and *V. cholerae* species rarely cause bacteremia, both species are unlikely to be identified using metagenomic data from tooth and bone remains sampled for ancient human DNA.”
(lines 333-336)

The interjection about vivax vectors seems poorly contextualized and extraneous.

We have modified the sentence to better contextualise it, but we are happy to remove it if the reviewer wants us to. It now reads:

“The *P. vivax* malaria vector *Anopheles atroparvus* is widespread today in Europe and surrounding areas, including e.g. North-Eastern Europe, Russia, and the Pontic Steppe. Our positive cases suggest this was also the case in the past.”
(lines 349-351)

L302 might be stronger without the word “ancient” -- a serious investigation of this question would focus on remains from 100 or so years ago, I think.

We agree with the reviewer and have removed the word ‘ancient’.

The s. at L335 seems extraneous and poorly supported.

Based on this and the comment from reviewer 2 we have modified this sentence to:

“Overall, our results show that co-infections can be detected using ancient metagenomic screening, but are likely underestimated given methodological limitations such as differences in pathogen load, tissue availability and other factors impacting detectability of ancient microbial DNA.”
(lines 390-393)

At L347 it would be good to say briefly how the properties of extinct strains are known. Alternatively, the authors could say “believed to have had” if they don't want to get into it.

We have followed the reviewer's suggestion and inserted “believed to have had” plus inserted a sentence detailing the adaptations of *Y. pestis* that increased flea-borne transmission:

“It reached a first peak around ~5,000 BP, coinciding with the emergence and early spread of the LNBA- strains, which are believed to have had limited flea-borne transmissibility”
(lines 404-406)

“The adaptations included acquiring two plasmids: one with the *ymt* gene for survival in the flea midgut and the other with the *pla* gene for invasiveness after transmission from a flea bite.”
(lines 417-419)

L434. “No effect” should virtually never be used to describe a statistical inference. This case is not by any means one of the exceptions.

The section on spatiotemporal modelling has been updated with the new results, and now does not include mentions of “no effect”.

Text around L470 is well qualified, but still comes across as too strong. I suggest dropping “sweeping through the continent”.

Thank you. We have toned it down by changing ‘sweeping’ to ‘moving’.

at L483 is also a bit too strong. Certainly this analysis is far better supported than earlier analyses, but there are presumably other possible explanations for the detection patterns.

We have toned down the claim by modifying the sentence to:

Importantly, expanding our analyses to the wider pathogen landscape allowed us to infer and contrast incidence patterns between different species and types of pathogens to a greater extent than previously possible.
(lines 553-555)

Authors need to say more about the topic-model analysis. Was number of topics the only choice they needed to make? How did they make it?

We only carried out analyses with L=2 and L=3 topics, to investigate broad structure in the data. This has now been added to the methods section.

s at L478 comes across a little strong. Which part of this interpretation is supported?

We have now toned down the sentence with “likely” as this is a fair interpretation of the results reported in the cited paper by Barrie *et al.*.

“Our findings also support the interpretation of increased pathogen pressure as a likely driver of positive selection on immune genes associated with risk of the autoimmune disease multiple sclerosis

in Steppe populations around 5,000 years ago, and genetic adaptations in genes involved in host-pathogen interactions having occurred predominantly after the onset of the Bronze Age in Europe” (lines 547-551)

We have responded to the individual reviewer's comments on a point-by-point basis below. Comments from the reviewers are formatted in *italics*, and quoted text from the updated manuscript is shown in blue font.

Referee #1 (Remarks to the Author):

This is an excellent paper and the revised version is much improved. It will make several original contributions, as highlighted in my previous comments. Given the potential significance of this paper, I would advise the following further suggestions be taken into consideration by the authors and editors.

We are delighted that the Reviewer finds the manuscript much improved and we have responded to each of the comments below.

1) The classification of diseases is much improved. A few issues remain. The new classification compresses (1) reservoir and (2) mode of transmission into one dimension, "Microbial origin group." That is invalid. Plague is both a zoonotic disease (animal reservoir) and a vector-borne disease (highly dependent on flea-borne transmission). The malaria spp discussed are vector borne but also adapted to humans. So maybe two dimensions: reservoir and transmission (oral-fecal, respiratory, vector-borne, sexual). This is something that has to be done right for the paper to make sense.

We thank the reviewer for the useful suggestions. In the revised manuscript, we have now updated the classifications of microbial species in the putative infection group to incorporate both broad reservoir and transmission categories. The nomenclature used in the revised scheme is

“infection_<reservoir>_<transmission>”

Where <reservoir> is either

- “zoonotic” (animals)
- “anthroponotic” (humans only)

and <transmission> is

- “vector” (vector-borne)
- “human” (human-human)
- “environment” (any other type environment-human transmission)

Of the species previously in the “infection_vector_borne” category, *Borrelia recurrentis* and *Plasmodium spp.* are now re-classified as “infection_anthroponotic_vector”, whereas *Borrelia crocidurae* is classified as “infection_zoonotic_vector”.

All group-level analyses and corresponding figures and tables have been updated accordingly, and the revised classification scheme is provided in Supplementary table S3. For the analyses presented in figures 5 and 6 in the main text we use merged reservoir groupings obtained by collapsing the different transmission types, in order to maximize sample sizes. All major conclusions of our study remain the same with the revised classification. Interestingly, with the revised classification we observe that

“anthroponotic” reservoir infections increase in more recent time periods than “zoonotic” (within the past ~2,500 years), mainly due to increased vector-borne disease (see revised Figure 3b, Extended Data Fig. 8).

2) I also question the placement of shigella, typhoid, and related diseases as 'environmental.' That is misleading. They depend on human hosts (and possibly other animals, esp earlier in their evolutionary history) to persist.

We thank the reviewer for pointing this out. In the revised manuscript, the species hits of *Salmonella enterica* and the genus *Shigella* have now been re-classified as infections due to their dependence on the human hosts (Supplementary table S3)

3) The supplement on LBRF is helpful though still not quite adequate, given that this paper could become a reference for the study of this neglected disease. Can more please be said about epidemiological factors? This is really a disease of 'misery' and crowding? It is more common today in Africa but might colder winters have contributed in the past in temperate climates? Some of the historical claims are pretty bunk - the MacArthur paper.

We thank the reviewer for the kind and helpful comments. As suggested, we have extended the description of possible epidemiological factors, highlighting the influence of climate even in peaceful and prosperous times in the Supplementary information. Moreover, we have included new information from an unpublished preprint about the evolution of *B. recurrentis* (Swali et al., 2024), which likely helps explain our observation of the transition from epidemic outbreak to endemicity. We have included this new information in Supplementary information 3 and the main text (lines 442-443; 456-459)

We agree with the reviewer that the MacArthur paper is old, however, it is referred to in many papers published in the last couple of years as the first paper to convincingly argue that the ‘Yellow Plague’ (Europe, 550 AD) and the famine fevers in the 17th and 18th in Ireland and elsewhere was predominantly caused by LBRF (e.g. in 2019 and 2022 (Röttgerding et al., 2022)). The American Society for Microbiology lists LBRF as a possible cause of the ‘Yellow Plague’ (*Relapsing Fever: A Two Thousand Year History*, 2023) which in 550 AD followed the Justinian Plague (which was caused by *Y. pestis*). We have now included the 2019 paper by David Warrell in the Supplementary information, along with the MacArthur paper, as the MacArthur paper appears to be the first to propose this idea, and we have changed the text from ‘Convincingly arguing’ to ‘Arguing’ to imply that this is seen as a respected suggestion by many, but not all.

4) 61-2 The authors of the paper have replaced an outdated reference to the history of infectious disease with something equally bad, note 7. It's a weakness of this paper that there is no engagement with history. There are lots of serious papers and books. I work in the field and have never seen the paper they cite. It is written by a clinical psychologist in upstate New York with no apparent training in history or archaeology, and it shows, because the paper is riddled with sophomoric errors. Please, look up a credible source on the history of infectious disease and its impact on societies!

We agree with the reviewer that this chapter from the book (Pandemics throughout History, 503 citations) is written by a clinical psychologist and is not optimal for our purpose; the book focuses on the mental health implications with respect to the unique elements of pandemic outbreaks over time. We have therefore replaced the reference with the excellent book “Plagues upon the Earth: Disease and the Course of Human History” (by Kyle Harper, Princeton, NJ : Princeton University Press, 2021).

5) 117 Fig. 1 is great, but in the text, they don't seriously and earnestly address the main concern about sampling bias, which is not within-cemetery bias. Rather, how were these 1313 individuals assembled? the very fact that they are Eurasian means it excludes tropical (and most subtropical) regions, all of Africa, Australia, and the Americas. In other words, it is biased against the most densely populated parts of Eurasia, plus all the other continents. Eurasian steppe populations account for what percent of historical humanity - it would be possible to do a back-of-the-napkin calculation, and it's going to be less than 1%. But here they are what percent? This sample is exceedingly biased in that respect.

Our study has unavoidable limitations; for example, we are more likely to acquire ancient remains from more populated areas than sparsely populated ones. Moreover, funeral rites influence the availability of remains; for instance, if the deceased were cremated during these rituals, no physical remains will exist. However, we have taken care to sample as broadly as possible in space and time.

We agree with the reviewer that more easily accessible information on sample provenance in the dataset was lacking in iterations of the manuscript. To address this concern, we have now included a more detailed breakdown of the geographic distribution and burial context in a newly revised Figure 1, and expanded the corresponding section in the main text:

“We applied this workflow to a dataset of ~405 billion aDNA sequencing reads from 1,313 ancient individuals from Western Eurasia (n=1,015; 77%), Central and North Asia (n=265; 20%) and Southeast Asia (n=33; 3%), spanning a ~37,000 year time period from the Upper Paleolithic to historical times (Fig. 1b; Supplementary table S1). The sample set represents individuals previously analysed for human demographic history, and derive from a range of contexts, the majority of which deliberate burials, but also including loose finds and to a lesser extent cave finds (Fig 1b.)”

We further now also provide DOI links for the primary publication of each sample and site in the dataset as new columns in Supplementary table 1.

6) Just as importantly, what kinds of archaeological work generated this sample? We all know that anomalous burials and mass burials are disproportionately screened, so what is the role of that in the background of this sampling?

The updated metadata provided in Supplementary table 1 now includes burial context information for 1,301 of the 1,313 samples (99%) that we could assign reliably from the primary literature on the remains. We have included a breakdown of different contexts for each geographic subregion in the newly revised Figure 1b. The majority of samples stem from burials in cemeteries or tomb contexts (e.g. passage graves, burial mounds, kurgans). Mass burials while also present only constitute ~6% of the dataset (n=81), of which the majority (n=62) stems from three Viking Age mass burials (Salme, Oxford, Dorset) with

evidence of violence related to warfare. As such we do not expect those to skew the dataset towards higher incidence of infectious diseases.

7) *What percent of the sample is "urban," for each period?*

A large part of our dataset covers the period between ~12,000 years to 2,000 years ago, with a variety of subsistence contexts including hunter-gatherers, agriculturalists and pastoralist lifestyles before the advent of urbanism. In the more recent time period from 2,000 years ago, the dataset includes predominantly Viking Age and Medieval individuals from Europe, as well as nomad pastoralist groups from the Eurasian Steppe.

Viking Age samples that are not from mass burial contexts largely originate from cemetery burial contexts, and thus are expected to represent the local settlement populations. The medieval samples are largely derived from three cemeteries in Denmark, two of which are from an urban setting (Vor Frue Kirkegaard, Aalborg; Ahlgade, Holbaek; 58 samples, ~4% of dataset), and one rural (Tjaerby, Randers; 28 samples, ~2% of dataset).

8) *283 the reported mortality for LBRF is problematic. first, i assume this is case fatality rate? be clear. second, this does not look reliable and is unsupported by an objective reading of the paper they cite. The paper says this "Case fatalities between 30% and 70% have been reported in untreated patients during major historic epidemics, but in treated cases, on average, 2–6% will die." To support this claim, that paper cites the 1970 paper "Louse-Borne Relapsing Fever; A Clinical and Laboratory Study of 62 Cases in Ethiopia and a Reconsideration of the Literature." I read that. It reports case fatality from .5% to over 50%, in some highly suspicious reported outbreaks from early 20C central Africa. LBRF may have been a serious disease, and it has been very underconsidered. But rehashing bad work to dramatize it will undermine the important findings of this paper. I really hope the authors will work to get this right.*

We are grateful for the reviewer's comment and we have now modified the main text and Supplementary information 3 stating that the untreated mortality of LBRF is 10-40% as reported by the ECDC (ECDC, 2015) . We have furthermore explained the caveats and limitations of the other studies describing a 30-70% mortality in Supplementary information 3.

9) *334 it's weird to be stubborn about V cholerae. The reason you don't find it is that it emerged in the late 18th century and only reached the sampled areas from the 1830s, and then very sporadically. And the point about M tuberculosis is fine, except we might have expected a much higher burden of advanced disease. What percent of the sample of 1313 is urban?*

We have now removed the text referring to *V. cholera* in the new version of the manuscript. It is correct that we might expect a higher burden of advanced TB; however, as far as we can judge from the literature, no good data supports or disproves this hypothesis. We have modified the description of *M. tuberculosis* in the main text to highlight this as follows

*"An examples of another major human pathogens not identified in our dataset include *Mycobacterium tuberculosis*, the cause of tuberculosis (TB) However, as the *M. tuberculosis* load in blood is generally*

low in immunocompetent patients without advanced disease(Rees et al., 2024) and latent TB develops in 60% of present day cases and can persist for decades, it is, based on current knowledge, unlikely to be readily identified using metagenomic data from tooth and bone remains sampled for ancient human DNA.”

For the discussion of urban samples see above answers to comments number 6 and 7.

10) 408 Starting at ~2,000 BP, plague reappeared in our dataset (samples 409 DA92, DA101, DA104, Kazakhstan and Kyrgyzstan, Central Asia; Fig. 3; Supplementary table S2), 410 including the period before the first historically documented plague pandemic (~1500-1300 BP, 411 VK522, Sweden and previously published cases

This is really interesting, but some things have to be clearer.

The plague starts showing up again around 2000 BP. But only in samples from plague's enzootic region, Kazakhstan and Kyrgyzstan. As best I can tell, there are only a couple of individuals sampled from the 'blank period' of plague from the enzootic region. Some discussion of the sampling is in order. It is essential to be clear that the positive plague samples prior to the historically documented First pandemic are from central Asia. This will strengthen the important point raised in the conclusion that "The pattern in the later periods is consistent with an epidemic period, distinct waves matching the historically described plague pandemics sweeping through Europe separated by periods without detection of the pathogen."

We agree that the sampling was not addressed well enough in the latest version of the manuscript. In particular, our dataset is challenged by the few European samples from the pre-roman Iron Age (2,800-2,000 BP). This sampling bias is, to some extent, driven by low sample availability due to the use of cremation in this period. Furthermore, the vast majority of ancient DNA samples that do exist from this time period are sequenced with capture enrichment strategies rendering the data unsuitable for metagenomics. Hence, this uneven sampling could, in principle, explain the lack of plague cases in Europe during the plague ‘blank’ period. However, there is no sampling bias in Asia (see revised Fig 1b), and, accordingly, if plague were present in the years 3000-2000 BP we should have detected it in this area. We have updated the description of the samples preceding the first pandemic accordingly as follows:

“Detection remained high with additional peaks for a ~3,000 year period, until an abrupt change ~2,800 BP led to a ~800 year period where plague was only detected in one sample (VK522, Oland, Sweden 2,343-2,154 cal. BP). Starting at ~2,000 BP, plague reappeared in our dataset in three samples from Central Asia (DA92, DA101, DA104, Kazakhstan and Kyrgyzstan; Fig. 3; Supplementary table S2), just before the first historically documented plague pandemic (previously published cases, Fig. 4).“

11) Also, where do these four Y pestis genomes sit on the phylogeny? I would presume the three central asian cases are basal (one has been published, I think).

Unfortunately the genomic coverages for the three newly reported *Y pestis* cases from this time period are too low to perform phylogenetic analysis (DA92 -163 reads, 0.002X; DA104 (701 reads, 0.008X);

VK522 (871 reads, 0.008X). The higher coverage sample DA101 (153,872 reads, 1.7X) was indeed previously published in (de Barros Damgaard et al., 2018); as indicated in the Supplementary table S2. It was found to diverge ancestrally within a clade including first pandemic genomes from Europe (e.g. (Keller et al., 2019))

12) Finally, I am pretty sure there is a grave error here. They report an individual positive for Y pestis from Sweden. VK522. From Oland. They cite Margyaran 2020. I go there and look for VK522. I go to the Supplement of Margyaran 2020. I see VK522 is equated with Oland 1052 and given a radiocarbon date 386 ± 80 CE. That aligns with the date in the supplement of this paper and used in all the analysis. An individual positive for Y pestis in Sweden from the fourth-fifth century AD is a huge find. The people who work on the first pandemic will immediately try to rewrite the history of everything we know about it! But honestly I am skeptical plague would show up there, so I dig a little. The Margyaran supplement cites Wilhemson 2017. There, in the supplement to this Lund PhD thesis, I read that this individual (Oland 1052) was radiocarbon dated to 386 ± 80 BC. This thesis uses BC and CE. That's 700-800 years different. Now, this is a grotesque mistake, with massive consequences (because others will not dig, as apparently the authors of this paper did not dig, to do minimal due diligence on a finding that is pretty surprising), so I have to think maybe it is me who has misunderstood something. If so, I sincerely apologize. If not, dear goodness please get this correct and double check things so that plague historians do not get misled.

We thank the reviewer for spotting this mistake, which appears to have occurred in the supplementary information of (Margaryan et al., 2020) where the genomic data of that sample was first reported. We have now corrected the age for this individual, which is 2325 ± 45 uncal BP, resulting in a calibrated age of 540-208 cal BC (95.4%), median value 394 cal BC. This is indeed somewhat earlier than the previous age, and now places this individual as our sole representative within the first period of ~800 years hiatus after the extinction of the LNBA- strains. While its low coverage prohibits phylogenetic assignment of VK522, analysis of plasmid coverage (please see the response to reviewer 5 (who also comments on reviewer 2's comments), comment number 11, below) suggests that its pMT1 plasmid contained the *ymt*-gene region (see new Extended Data Fig. 7), which would place it after the divergence of the LNBA+ and later lineages. To resolve those questions, new sampling and sequencing of the individual would be needed, which is outside the scope for this study. We generally caution against making strong interpretations on individual outlier samples in large-scale archaeogenetic studies, as sample mislabellings, swaps etc while rare can happen at any stage of the process from excavation to sequencing data analysis.

Furthermore, in light of the issues with VK522 we have also now done additional verification of the metadata, and have only found and corrected a few minor inconsistencies in previously reported metadata (shown in the provided updated metadata table with changes from the previous version tracked).

In all there would be a lot of value in clarifying what this paper says about the history of plague by more phylogenetic analysis and more clarity on sampling. I know that a phylogenetic paper on the plague is another study, but I'm not sure it's responsible to give this much info here without just going further.

We appreciate that a more in-depth phylogenetic analysis would be of interest. We have now included phylogenetic placements for the subset of *Y pestis* cases where coverage allows it ($>0.01\times$; 19 samples total) as part of the newly added Supplementary information 4. Out of the 19 samples, 3 high coverage samples were previously published and included in the reference tree (RISE505, RISE509, DA101), and the remaining ones placed onto the reference tree using epa-ng as previously described (Seersholm et al., 2024)

Again, this paper has the potential to be a blockbuster, so it is crucial to address these remaining issues before hopefully publishing what will be a major contribution.

We are grateful for the reviewer's kind and encouraging comments.

Referee #3 (Remarks to the Author):

The authors have provided a robust revision. While ideally soil and other negative controls would be included in a prospective study, I accept that the authors are unable to obtain samples like this when retrospective data collected for aDNA studies is used.

1) The only concern I have is with their handling of unclassified reads. They treated these separately from non-human classified reads, but unclassified reads can significantly impact relative abundances especially considering the high percentage of unclassified reads. I would suggest authors focus solely on the presence/absence of pathogens, omitting discussions on relative abundances.

The focus of our manuscript is as the reviewer suggests on presence/absence of pathogens, relative abundances are only briefly discussed in the context of Figure 2c / Extended Data Fig. 5b. The purpose of these discussions was to highlight broad patterns of how easy or hard ancient microbial species are to detect on average in our sample. We have therefore modified our description to use “read mapping / recruitment’ over sample median numbers to avoid discussing abundances. The updated text for Figure 2c/ ED 5b now reads:

“The rate of read mapping varied by orders of magnitude between species, from hits in species with high read recruitment such as *Mycobacterium leprae* (> 100-fold enrichment over median number of classified reads across target genera) to hits at the lower limits of detection as, e.g., for the louse-borne pathogen *Borrelia recurrentis* (lowest read recruitment ~100-fold less than median number of classified reads across target genera; Fig. 2c; Extended Data Fig. 5b).”

2) Overall, I think this work can act as a pioneer in the field of ancient microbial DNA research by reanalyzing ancient human samples and proposing a robust authentication framework.

We are grateful for the reviewer's kind and encouraging comments.

Referee #4 (Remarks to the Author):

The authors' revisions are appreciated, and they have addressed most of my concerns adequately.

We are delighted that the reviewer finds that we have satisfactorily addressed all the previous concerns except one.

I have one remaining concern.

Using WAIC-based selection is presumably a good way to choose a model for posterior predictive accuracy (as the authors discuss in their rebuttal, but not in their MS). The authors don't even make an argument that it's appropriate for Bayesian _inference_ (i.e., parameter estimation), and I don't believe that it is. What kind of “overfitting” are the authors worried about here? They seem to focus on the _effects_ of covariates, not on any predictions.

We thank the reviewer for this comment. In the revised manuscript, we have modified this analysis and now use the deviance information criterion (DIC) to compare models, which is suitable for comparison of Bayesian models. We found that with the exception of the analysis for the oral microbiome species *Porphyromonas gingivalis*, the same model was selected as when using the WAIC before. We provide both DIC and WAIC scores in the updated Supplementary table S6.

Referee #5 (Remarks to the Author):

This is a very significant project, the first of its kind, which makes full scale screening result publicly available, thoroughly analyses the data and uses a well tested screening workflow developed for the project. I commend the author on undertaking such a big task, which involves the work with a very large set of taxa and the reuse of previously published data. Additionally, I was happy to see that additional data and code was or will be made available (although I would have liked to see an example). The workflow itself is well-designed and filtering parameters well-chosen, although some aspects such as the limitation to a single species in each genus are not ideal, and I was somewhat confused why genera were only included if more than 2 established human pathogens were present. I also appreciate the use of presence/absence data on infectious agents, which is sadly mostly missing in the literature. Since the knowledge of the presence of a disease in a population at a given time can already be precious knowledge, as this study clearly demonstrates.

We appreciate the supportive comments from reviewer 5, the recognition of the significant task we have undertaken, and the acknowledgement of the effective design of our workflow.

Reviewer #2 comment response:

All of the minor comments from reviewer #2 were adequately addressed.

We are delighted that reviewer 5 found that we had satisfactorily addressed all the minor comments from reviewer 2.

1a) R#2: "While I am excited by this project, I do have some concerns over the interpretations. Some very strong claims are made, and I feel quite a bit more could be done to discuss the potential limitations of the input data and resulting insights. Even addressing these questions using modern day metagenomes (which would be an interesting comparator) would be fraught with the challenge of accounting for sampling bias and research effort. Furthermore, the classification of microorganisms into pathogenic or non-pathogenic or zoonotic or opportunistic etc. is non-trivial".

1b) I have similar concerns. While the project is indeed novel and highly relevant, I am concerned about some conclusions drawn in this article. I think the methodological work is very good and while not without issues, constitute a very good basis for a spatio-temporal analysis. However, when it comes to the broader interpretation of results beyond the description of large scale trends (not based on specific species), I am more sceptical. The analytical results are present/absence data, meaning that we know nothing of the genomic structures or phylogenies of these organisms. I appreciate that it is indeed

challenging to work with such a large bulk of taxa and that much of the authentication process has been well automated, so to speak, however, particularly when such important conclusions are being drawn, final identifications should be double-checked anyway and for any specific conclusions relating to function and adaptation, full genomes should be available. As the bulk of your pathogenic hits are reduced to a handful of species, this should be doable.

To address this concern, we have added a new Supplementary information 4 section with more detailed discussion on the evidence for individual bacterial species in the “infection” classes. The section includes more in-depth descriptions of authentication statistics for those species as well as details on reference genome and plasmid content.

We further added a new **Supplementary table S5** providing expanded summary statistics from the workflow with separate entries for individual contigs in each genomic assembly, for the “infection” group of microbial hits. The full workflow results with per-contig statistics for all samples and mapped species will be provided as part Supplementary dataset S1 (full workflow result summary table available for review at <https://www.dropbox.com/s/rvt17h5gm0017jp/diseases.summary.tsv.gz?dl=0>; to be included in zenodo repository for final publication). We hope that these additions now address these remaining concerns.

2a) R#2: “The authors construct a database of 44 ‘human pathogens’. Little attention is given to how this was devised. The status of a species as a human pathogen or not is often not that clear cut, particularly for species where the life histories through time and reservoir hosts are unknown or poorly characterised. What do the authors consider as a ‘pathogen’? Databases do exist of ‘human pathogens’ – were these considered? Eg. work from Bartlett et al. discuss and release a much larger database of 1513 bacterial pathogens (link). The selection of genera to be considered as pathogens may well colour resulting observations. Similarly, the classification into zoonotic, anthroponotic, sapronotic etc is non-trivial but not explained. There certainly may be anthroponotic species which are in fact zoonotic, though the animal reservoirs have been poorly characterised. Eg. some argue HBV is not anthroponotic <https://www.wjgnet.com/1007-9327/full/v20/i24/7665.htm>. I would recommend carefully evaluating host-pathogen association databases eg. CLOVER to provide justification for some of the classifications. This is very important because the major observation that zoonotic pathogens are not detected before 6,500BP is based entirely on how the authors chose to classify their detected species.”

2b) This was adequately addressed.

We are pleased that reviewer 5 found that we had satisfactorily addressed this comment from reviewer 2.

3a) R#2: “aDNA has amazing potential to provide unexpected insights. I have no doubt that the distribution of infectious diseases was likely different in the past, but some examples in text do need additional discussion. An example is the claimed IIV-31 human infection. The authors comment that this is a virus of woodlouse but without citation, and it does seem like a virus of invertebrates rather than one you would find in human teeth. Does the virus receptor allow entry to vertebrate cells? Can we believe this? In addition, other unexpected observations such as Proteus virus Isfahan may be genuine but this species has also implicated as a possible contaminant in DNA extraction kits/lab environments. Are the

damage profiles strong in this case? These are not the only unexpected observations. Amongst the provided supplementary tables there are hits to citrus plant viruses, cattle parasites, and fish pathogens. How confident can we be that the results presented relate to genuine ancient human infections? This does bring into question some of the observations and these should be discussed with more care”

3b) While this is good, I do believe that particularly for the pathogens which play a larger role in the discussion (and especially the ones which are not really expected to be found within the sampled tissue) and the conclusion, a more thorough species validation should be presented in the SI.

To address this concern, we have added a new Supplementary information 4 section with a more detailed discussion on the evidence for individual species in the “infection” classes.

4a) R#2: “A high % assignment to toxoplasma is evident across the dataset. This is likely due to eukaryotic contamination in the reference database rather than toxoplasma being present at such a high frequency. I would be concerned, particularly for parasite species, that contamination in the RefSeq reference genomes or eg. retained adapter content etc. could lead to spurious hits and inflated prevalence estimates. What steps were taken to mitigate for this?””

Sikora et al: We agree with the reviewer that Toxoplasma hits can be caused by database contamination, in fact we are presenting them as examples of contamination, and how our authentication approach guards against such false positive results (e.g., coverage evenness statistics, Extended Data Fig. 4). As a consequence, our final authenticated results do not contain any hits for Toxoplasma (Supplementary Table S2).

4b) This was adequately addressed. But please see my remarks below regarding the simulations and Plasmodium species.

We are delighted that reviewer 5 found that we had satisfactorily addressed this comment from reviewer 2. Please see our answers to the comments below regarding simulations and *Plasmodium* species.

*5a) R#2: “Previously published datasets are not really considered and observational data (place and time) is readily available for many human pathogens from eg. the SPAMM initiative <https://github.com/spaam-community/AncientMetagenomeDir>. While I appreciate exhaustive inclusion is a big ask, I think for *Y. pestis* and LBRF (I note this is only one previously published case) these should be considered in the assessment of temporal dynamics. Certainly for *Y. pestis* this would add a good amount of data. If this data can’t be included given the need for a ‘rate’ of detection then could some of the claims be validated given published datasets. Eg. is it true that there is an 800 year period of no published observations in published *Y. pestis* data also? And why does an absence of plague support a prevalence of LBRF....? Are the authors claiming some cross-immunity or the such like? This feels a strong statement without further expansion”*

*5b) The data was included (although it can be noted that within the last week multiple new *B. recurrentis* genomes were published on BiorXiv Swali et al 2024). While I agree with the conclusions regarding *Y. pestis* and *B. recurrentis* regarding lifestyle and hygiene, I think that any speculation regarding body lice transmission is out of scope here. You have no data on the genomes of these cases, merely presence*

absence data. Additionally, while the study did indeed detect large amounts of cases of Y. pestis and B. recurrentis, both these pathogens are very well suited for molecular detection in teeth and are highly adapted human pathogens. Many are not, be it because of tropism, low viral/bacterial load, preservation or non-DNA genomes. Correlating them in such a way is not really warranted as the whole picture is still not available, and you cannot exclude genome variation or ecological aspects. While the absence of cases is certainly interesting to note and would be interesting to investigate further, the data presented here is not an adequate basis for a hypothesis on vectors or very specific cases of adaptation.

We thank the reviewer for pointing out the Swali et al 2024 preprint, from which we have now added the samples to the “previously published” distribution of Figure 4 and we have discussed the preprint in Supplementary information 3 and the main text. We think that the addition of the published experimental results regarding body lice infection with *Y. pestis* and *B. recurrentis*, respectively, are valuable as they shed light on one of the factors that might help explain the contrasting detection patterns we observe (Bland et al., 2024). We agree with the reviewer that full genomic data combined with functional experiments would be necessary to verify this suggestion. Therefore, we have made it a priority to emphasize that this is merely a suggestion and not an established fact in the main text. In order to highlight the limitations of our study more prominently, we now also provide a restructured and expanded section of the limitations of our study touched upon by the reviewer in the discussion section of the revised manuscript. The penultimate paragraph of the manuscript is now as follows:

“We note that despite these advances, the nature of our dataset also poses some important limitations on possible inferences. While ancient shotgun metagenomic data allows for direct molecular evidence of past infections, it is contingent on sufficiently high pathogen load and access to the appropriate tissue. Our sample of ancient tooth and bone material is well suited to detect bloodstream infections with high pathogen load such as *Y. pestis*, but pathogens with lower load and/or different tissue tropism are underrepresented in our dataset. Furthermore, as our dataset is lacking information on RNA viruses, our analyses also likely underestimate the overall zoonotic disease burden. However, the timing is likely accurate as the conditions favouring zoonotic transmission of RNA viruses are similar to those of other zoonotic pathogens, e.g., an increase in the interaction between species and shared living spaces. Although zoonotic cases also existed before the transition, the risk and extent of zoonotic pathogen transmission likely increased with the adoption of more widespread husbandry practices and pastoralism. Today, zoonoses are estimated to account for over 60% of newly emerging infectious diseases”

6a) R#2: *"The authors should be careful about overstating what can be gleaned from this dataset. It is of course very impressive in scale, but even the best efforts are likely to be only a partial proxy to "infectious disease load over time". There are a number of potential issues with using this data to generalise to all human infectious disease. This includes that the data is Eurasian centric rather than global, not all species are considered, the work is restricted to DNA pathogens, there is differential preservation and survival of different microbial species (each species likely has a different LOD and some are very unlikely to be found despite being highly significant eg. Mtb). Perhaps some species, suggested as prevalent in the past, have high asymptomatic rates and are under sampled and studied today meaning their current distribution is poorly known. No systematic screen has been conducted of contemporary metagenomes so for many pathogens it is difficult to comment on their true distribution and prevalence today. These points should be more carefully discussed"*

6b) The authors have included references to limitations in the manuscript. However, I believe the manuscript would benefit from more details, particularly regarding tropism, LOD, and how disease phenotype can affect recovery.

Please see our revised limitations section in response to 5b above.

Beyond the issues raised above, here are my comments:

7) A lot of this work depends on the databases used, so I would have liked it to be described in a bit more detail. Were Univec/Emvec type databases included? Since you used KrakenUniq, databases should be dusted by default, but how much is the mapping in bowtie impacted by sequence complexity issues?

We used default parameters to build the KrakenUniq database, and low complexity regions were masked. We have now also included these in the methods section. Regarding the impact on *bowtie2*, we don't expect any adverse impacts, as our default mapping quality cutoff of $\text{MAPQ} \geq 20$ would be expected to also remove reads mapping ambiguously to low complexity regions.

8) While *H. pyroli* and *MTB* do have skewed GC content, most other genomes that were tested during the simulation do not, and I am wondering how it fares with viruses, which can exhibit repeats over large percentages of their genomes.

I am very happy that simulation was included in the analysis. However, except for the two organisms noted above, the tested species are mostly comparable in genome length, GC and plasticity. I am a bit concerned about the analysis of *Plasmodium* hits and those with similarly large genomes, which are not accounted for during simulation (the best hit covers 0.15% of the reference sequence). These genomes differ significantly from 3-5Mb bacterial genomes and would need a lot more data to validate. Considering your issues with *T. gondii*, including larger protozoan genomes and viruses would be advantageous.

In the revised version of the manuscript, we have now extended the simulations to include additional species representing a broader diversity. In particular, we have added simulations for the malaria parasite *Plasmodium vivax*, as well as two species of human DNA viruses, *Variola virus* and *Human betaherpesvirus 5*. Our results show that we can readily detect each of those species using our workflow and authentication criteria. In particular, we note for the newly added species:

- All replicates down to $n=50$ simulated reads pass our initial filtering with $n \geq 150$ unique k -mers classified at the genus level (Extended Data Fig. 2b)
- The true positive species show the highest number of unique k -mers assigned at species level in the *KrakenUniq* step across all read numbers and replicates (Extended Data Fig. 2d)
- The true positive species is correctly determined across all replicates and read numbers, and cross-genus false positives are not observed (Extended Data Fig. 3d)

9) Additionally, regarding your criteria for a positive call, depending on the genome plasticity, available diversity and sequence divergence within the species, the use of ANI, for example, which is of course a very important metric, can need slightly different cut-offs. At least one of the chosen species *Y. pestis* evolves clonally and similarly to *M. leprae* could require higher ANI cut-offs, and the opposite can be true for very plastic/divergent species or ancient strains that fall distantly basal to the modern diversity. These would also often display increased mean edit distances. So it would be good to show, with your simulated data, that these differences don't make an impact with a wider choice of species and the inclusion of more divergent strains.

We agree that choosing an appropriate ANI cutoff is not straightforward in the context of the issues mentioned by the reviewer. However, we are not using ANI as a sole criteria for determining the ancient microbial species, but rather as a guard against false positives using a minimum value cutoff. The case of *Yersinia pestis* mentioned by the reviewer is a good example, as ANI is >99% between *Y. pestis* and some *Y. pseudotuberculosis* strains. Despite this, our simulations of the two species show that we can readily identify the correct species by combining a minimum ANI cutoff with the number of unique *k*-mers assigned at the species level using KrakenUniq (Extended Data Fig. 2c).

The other extreme of divergent strains is illustrated by the highly diverse *Helicobacter pylori*, where our simulated reads show an ANI of only 97% against the closest match in the reference database, just passing our cutoffs. Nevertheless, nearly all replicates are successfully detected, the exception being the lowest read number simulations with *n*=50 reads, where 2/10 replicates fail the ANI cutoff (Extended Data Fig. 3d).

Based on these results, we are confident that our workflow can robustly detect ancient microbial hits across a wide range of genome contexts and divergence scenarios. To allow a more thorough investigation of the simulation results, we provide the full workflow results as part of Supplementary dataset S1 (full workflow summary table available for review at https://www.dropbox.com/scl/fi/8xbzgn5r5gnvjijbj5dd/readsim.hum_microbe.summary.tsv.gz?rlkey=qa7sigt4dv03lz63yh59z9ja0&dl=0; to be included in the zenodo repository for final publication) in an online repository.

10) While I appreciate that the simulation data has shown the overall suitability and effectivity of the workflow for pathogen detection in ancient DNA datasets, false positive cannot be completely excluded. Databases are still heavily biased towards pathogenic species and many environmentals/commensal organisms have not yet been sequenced or assembled. Most of the hits reported in this study (130 out of 195) are based on ca. $\leq 5\%$ of genome coverage of the reference genomes. This a) doesn't allow you to know anything about their genomes beyond their potential presence (which I realise is limited by the available shotgun data to begin with) and b) mostly limits the identification to "probable" cases, as it is unlikely the species can actually be fully validated with this amount of data. While you do point this out early on in the manuscript, it is mostly omitted later. Talking of "probable/likely" cases would be more appropriate in most instances.

We agree with the reviewer and we now use "putative cases" throughout the manuscript where appropriate.

11) That being said when reporting on species, which based on standard tropism and disease phenotype should not or only rarely be present in the sampled tissue (e.g. Y. enterocolitica) (and species which have not been reported previously), I would expect a more thorough analysis on species validation to be described in the SI. E.g. whether all the plasmids needed show coverage or if specific sites are present.

As mentioned in reply to comment 3b) above, we have added a new Supplementary information 4 section with further details on the “infection” class species. We further added a new Supplementary table 5 with expanded microbial hit summary statistics from the workflow with separate entries for individual contigs in each genomic assembly.

A new plot with coverage across the major Y pestis plasmids is also included in the revised manuscript as part of the newly added Extended Data Fig. 7, and the following text was added to the section describing plague results:

“All but one hit (NEO627, n=84 reads total) hits showed expected coverage for the virulence plasmids pCD1 and pMT1, with hits before 2,500 years BP characterized by the previously reported absence of a 19kb region on pMT1 containing the *ymt* gene”

Additionally, we have carried out further validation steps for all putative hits with low read count ($n \leq 100$ final reads). To further rule out spurious results for these lower confidence hits (n=712 in total), we carried out a BLASTn analysis of the final mapped reads using the most recent ‘nt’ database. We found that the vast majority of our low read number species hits were supported by this analysis, in particular those in the “infection” classes (new Extended Data Fig. 7). We have added the following section to the manuscript:

“To further verify hits with low read numbers, we additionally performed a BLASTn search for all reads of each hit with $n \leq 100$ final reads (n=712 hits total; Supplementary table S3). The vast majority of hits showed a high proportion ($\geq 80\%$) of reads assigned to the same species using BLASTn (Extended Data Fig. 7a), and the species with the most top ranked BLASTn hits generally matched the inferred hit species (Extended Data Fig. 7b).”

Referee #5 (Remarks on code availability):

Code has been made available, and additional output and databases will be made available publicly as well. I did not review the full code itself, however, a more extensive README file would be needed.

We are delighted that reviewer 5 is satisfied with the code availability we provided. We are updating the documentation of the code to include both a general description as well as a worked example using our simulated dataset.

References

- Bland, D. M., Long, D., Rosenke, R., & Hinnebusch, B. J. (2024). *Yersinia pestis* can infect the Pawlowsky glands of human body lice and be transmitted by louse bite. *PLoS Biology*, *22*(5), e3002625.
- de Barros Damgaard, P., Marchi, N., Rasmussen, S., Peyrot, M., Renaud, G., Korneliussen, T., Víctor Moreno-Mayar, J., Pedersen, M. W., Goldberg, A., Usmanova, E., Baimukhanov, N., Loman, V., Hedeager, L., Pedersen, A. G., Nielsen, K., Afanasiev, G., Akmatov, K., Aldashev, A., Alpaslan, A., ... Willerslev, E. (2018). 137 ancient human genomes from across the Eurasian steppes. *Nature*, *557*(7705), 369–374.
- Keller, M., Spyrou, M. A., Scheib, C. L., Neumann, G. U., Kröpelin, A., Haas-Gebhard, B., Pääffgen, B., Haberstroh, J., Ribera I Lacomba, A., Raynaud, C., Cessford, C., Durand, R., Stadler, P., Nägele, K., Bates, J. S., Trautmann, B., Inskip, S. A., Peters, J., Robb, J. E., ... Krause, J. (2019). Ancient genomes from across Western Europe reveal early diversification during the First Pandemic (541-750). *Proceedings of the National Academy of Sciences of the United States of America*, *116*(25), 12363–12372.
- Margaryan, A., Lawson, D. J., Sikora, M., Racimo, F., Rasmussen, S., Moltke, I., Cassidy, L. M., Jørsboe, E., Ingason, A., Pedersen, M. W., Korneliussen, T., Wilhelmson, H., Buš, M. M., de Barros Damgaard, P., Martiniano, R., Renaud, G., Bhérer, C., Víctor Moreno-Mayar, J., Fotakis, A. K., ... Willerslev, E. (2020). Population genomics of the Viking world. *Nature*, *585*(7825), 390–396.
- Rees, C. E., Swift, B. M., & Haldar, P. (2024). State-of-the-art detection of *Mycobacterium tuberculosis* in blood during tuberculosis infection using phage technology. *International Journal of Infectious Diseases: IJID: Official Publication of the International Society for Infectious Diseases*, *141S*, 106991.
- Relapsing Fever: A Two Thousand Year History*. (2023, February 2). ASM.org.
<https://asm.org:443/Articles/2023/Relapsing-Fever-A-Two-Thousand-Year-History>
- Röttgerding, F., Njeru, J., Schlüfter, E., Latz, A., Mahdavi, R., Steinhoff, U., Cutler, S. J., Besier, S.,

Kempf, V. A. J., Fingerle, V., & Kraiczy, P. (2022). Novel approaches for the serodiagnosis of louse-borne relapsing fever. *Frontiers in Cellular and Infection Microbiology*, *12*, 983770.

Seersholm, F. V., Sjögren, K.-G., Koelman, J., Blank, M., Svensson, E. M., Staring, J., Fraser, M., Pinotti, T., McColl, H., Gaunitz, C., Ruiz-Bedoya, T., Granehall, L., Villegas-Ramirez, B., Fischer, A., Price, T. D., Allentoft, M. E., Iversen, A. K. N., Axelsson, T., Ahlström, T., ... Sikora, M. (2024). Repeated plague infections across six generations of Neolithic Farmers. *Nature*, *632*(8023), 114–121.

Swali, P., Booth, T., Tan, C. C. S., McCabe, J., Anastasiadou, K., Barrington, C., Borrini, M., Bricking, A., Buckberry, J., Büster, L., Carlin, R., Gilardet, A., Glocke, I., Irish, J., Kelly, M., King, M., Petchey, F., Peto, J., Silva, M., ... Skoglund, P. (2024). Ancient *Borrelia* genomes document the evolutionary history of louse-borne relapsing fever. In *bioRxiv* (p. 2024.07.18.603748).
<https://doi.org/10.1101/2024.07.18.603748>

ECDC. (2015, November 17). *Louse-borne relapsing fever in the EU*.

<https://www.ecdc.europa.eu/sites/default/files/media/en/publications/Publications/louse-Borne-Relapsing-Fever-in-Eu-Rapid-Risk-Assessment-17-Nov-15.pdf>.

<https://www.ecdc.europa.eu/sites/default/files/media/en/publications/Publications/louse-borne-relapsing-fever-in-eu-rapid-risk-assessment-17-nov-15.pdf>

We have responded to the individual reviewer's comments on a point-by-point basis below. Comments from the reviewers are formatted in *italics*, and quoted text from the updated manuscript is shown in **blue font**.

Individual reviewer responses

Referee #1 (Remarks to the Author):

In the revised version of the manuscript and accompanying tables, the authors have thoughtfully responded to the reviewers' suggestions and comments. The authors have now corrected the classification of pathogens, and they have added helpful material such as the discussion of LBRF in Supp. Inf. 3.

This is now a very clean paper. It is a major contribution both substantively and methodologically. I wholeheartedly recommend publication.

We thank the reviewer and are delighted that they recommend the manuscript for publication.

Referee #4 (Remarks to the Author):

Thanks to the authors for their revisions.

My point about "WAIC-based selection" was apparently insufficiently clear. I agree that DIC is appropriate for comparing these models, but I don't object to WAIC, either. My point was that – in the context of Bayesian inference – the authors should avoid doing selection at all if possible, and to provide a clear justification if they feel that selection is necessary. Selection is popular in this context because it provides sharper confidence bounds than would be supported by the original assumptions, but this is not actually a supportable reason to use it.

We appreciate the concern of the reviewer about the appropriateness of model selection in the context of Bayesian inference and recognize that while their usage is not always recommended, they can nevertheless be helpful in situations such as ours to avoid the inclusion of unnecessary variables (Gelman et al., 2013; van de Schoot et al., 2021). In INLA and 'inlabru', several metrics for model assessment are computed, including DIC, WAIC, CPO, PIT and Marginal likelihood (Held et al., 2010; van de Schoot et al., 2021), and both DIC and WAIC are popular criteria for model choice (<https://becarioprecario.bitbucket.io/inla-gitbook/ch-INLA.html#sec:modelassess>). Given that smaller DIC and WAIC values indicate a better model fit, we report the results from those models in the main text (as also done in (Held et al., 2010)) and, in cases of best fit in models with fewer parameters, model selection leads to reduced model complexity, which is preferable. We note that we report the results for all models in supplementary table S6, and results for individual parameters across different models are highly similar (see summary figure 1 below). We hope this explanation of our reasons for using model selection is clear and satisfactory.

Figure 1. Estimated effect size and direction of effect for predictors across all models and response variables.

Referee #5 (Remarks to the Author):

The authors have addressed the majority of questions raised by me adequately and have provided extensive additional work. However, some questions remain. Please see below:

“We selected a set of 136 bacterial and protozoan genera (11,553 species total) containing at least two established species of human pathogens² to screen for ancient microbial DNA in the genus-level read mapping and authentication stages.”

I had ask for some clarification regarding this in my first review. This should be included in the text as it is unclear. Why were genera excluded based on the number of pathogenic species they carry?

We focused on genera with at least two established pathogenic species in Bartlett et al to strike a balance between including genera responsible for substantial human pathogenic burden and computational feasibility. In the revised manuscript, we have added this to the methods section as follows:

“Following this initial metagenomic classification, a subset of genera was further processed in the genus-level read mapping and authentication stages. For bacterial pathogens, we selected genera with two or more established species of human pathogens from a recent survey of human bacterial pathogens (n=125 genera). Genera with a single pathogenic species were not included in order to balance between including genera responsible for substantial human pathogenic burden and computational feasibility. We further included genera including human protozoan pathogens (n=11 genera), as well as all viral genera (n=1,356).”

Regarding comment 1b/3b:

I appreciate the addition of further validation for pathogenic species, although my reservations about functional and adaptive inferences in the manuscript remain.

For C. diphtheriae as far as I am aware the tox gene is not specific to C. diphtheriae and is present in homologous copies in other species of the genus, which should probably be noted.

We have now added a note on the possible presence of the tox gene in other Corynebacterium species in the supplementary section on *C. diphtheriae* as follows:

“The assembly of strain NCTC3529 carries the gene for the diphtheria toxin responsible for diphtheria when present in *C. diphtheriae* and to a lesser extent in related *Corynebacterium* species (*tox*, NZ_LR134538.1:1,942,803-1,944,485), however due to the low read counts its presence in the two ancient hits could not be determined.”

In Table S5, the KrakenUniq statistics are the same for each header, which makes sense, but this should be specified in the legend somehow.

We thank the reviewer for pointing this out and have added a table legend specifying that the KrakenUniq statistics concern the species-level classification.

Regarding comment 5a/5b:

While I appreciate the addition of the paragraph in the conclusion, it is never directly tied to this discussion nor does it come up within the discussion paragraph themselves, making it feel rather detached. In fact there has been little to no change to this section.

In the end sentences such as this one:

“Strikingly, this period of high LBRF detection coincided with a period without detectable plague activity

(Fig. 4), further supporting the notion that the absence of plague represents a true reduction in incidence

during this period rather than sample size limitations or poor DNA preservation”

Are speculative in nature, based on correlations discerned in the current data. While such a hypothesis can of course be proposed it should be made abundantly clear in the actual discussion why

the observation could wrong or apply to a different set of pathogens, which the methods in ancient DNA and in this study cannot recover etc. This is especially true since you also further expanded the discussion on vectors, and particularly on non-flea vectors for Yersinia pestis, which have been and still are under considerable debate.

As part of editorial changes to streamline the manuscript, we have followed the suggestion of the reviewer and have removed the more speculative parts concerning the possible vectors for Yersinia pestis from this section and incorporated them into the conclusions section. We also modified the sentence highlighted by the reviewer to emphasize that this result provides further evidence that the absence of plague in the same time period is not due to sample or preservation limitations. It now states

“The period of high LBRF detection coincided with a time without detectable plague activity (Fig. 4), reinforcing that the absence of plague is not due to sample size limitations or poor DNA preservation.”

Regarding comment 11:

Regarding cases like NEO627/Ypestis and NEO29/Brecurrentis. Are cases like these removed from the list of validated hits? I might have missed it but I did not see it made clear whether there were criteria for verification based on your additional analysis? While they might prove to be real with additional data, they are currently lacking coverage for a validation.

We consider these cases verified according to our criteria, as they show the expected patterns of ANI and coverage across the genome.

NEO29 shows expected coverage across two different contigs of the *B. recurrentis* assembly, with most reads mapping at the longest contig (n=30, NC_011244.1) and some reads mapping to a shorter contig (n=2, NC_011255.1) (Fig. S4.1). Patterns of ANI also follow the distribution expected for a true positive, with mapping to *B. recurrentis* and the two closest related species *B. duttoni* and *B. crocidurae* showing the highest ANI with similar values (Fig. S4.2).

For NEO627, ANI results also show the expected distributions across the different mappings (Fig. S4.14), and the reads mapping to the main chromosome show the expected even distribution (Fig. S4.16). The absence of reads mapping to the plasmids is likely random dropouts due to low coverage and short plasmid genomes, as other hits with a similar number of reads also show only limited numbers mapping to plasmids (e.g. NEO168, n=63 total reads with n=3 mapping to plasmids, Extended Data Fig. 7c). We also note that NEO627 is part of the same passage grave burial as NEO630, a well verified *Y. pestis* hit with a higher read count (n=461 reads).

Note that i could not access this link without logging in and thus could not review its content (https://www.dropbox.com/scl/fi/8xbzgn5r5gnvjiqibj5dd/readsim.hum_microbe.summary.tsv.gz?rlkey=qa_7sizt4dv03lz63yh59z9ja0&dl=0)

We have provided a new link for the file, which can be downloaded here https://sid.erd.dk/share_redirect/bkE1T90n3w

References

- Bartlett, A., Padfield, D., Lear, L., Bendall, R., & Vos, M. (2022). A comprehensive list of bacterial pathogens infecting humans. *Microbiology*, *168*(12). <https://doi.org/10.1099/mic.0.001269>
- Gelman, A., Hwang, J., & Vehtari, A. (2013). Understanding predictive information criteria for Bayesian models. *Statistics and Computing*, *24*(6), 997–1016.
- Held, L., Schrödle, B., & Rue, H. (2010). Posterior and Cross-validators Predictive Checks: A Comparison of MCMC and INLA. *Statistical Modelling and Regression Structures*, 91–110.
- van de Schoot, R., Depaoli, S., King, R., Kramer, B., Märtens, K., Tadesse, M. G., Vannucci, M., Gelman, A., Veen, D., Willemsen, J., & Yau, C. (2021). Bayesian statistics and modelling. *Nature Reviews Methods Primers*, *1*(1), 1–26.

We provide our responses to the remaining comments from reviewer 4 and 5 below. Comments from the reviewers are formatted in *italics*.

Referee #4 (Remarks to the Author):

When I asked for a “clear justification”, I meant in the MS, not in a rebuttal. Not clear whether the authors thought I was the only person who could possibly want that. In any case, the authors' justification is not responsive to my concerns.

I tried to be very clear that I was not challenging the general idea of information criteria (which would be ridiculous), but instead referring specifically to the context of inference (written as _inference_ in two of my reviews). The authors' primary citations are Gelman...13, which is explicitly about prediction (not inference), and vdS...21, which seems to be about inference, but does not seem to mention information criteria at all!

The authors don't make even an a priori case for why reducing model complexity (in this context, not some other context) is preferable. And of course their statement that lower IC values indicate better “fit” is technically wrong. Best practice would be to report parameter values and CIs from their full Bayesian model. Since they have produced a blizzard of information (table S6 and the rebuttal figure) in support of an argument that this would yield similar conclusions, they should not object.

We want to thank the reviewer for their comment, which has been very helpful and improved the manuscript, and apologize for the misunderstandings in our previous responses.

We have followed the reviewer's suggestion and now show the results of the full model and a comparison between the full model and the best-DIC models in an updated Extended Data Figure 11. As can be seen in this new figure, the full-parameter models and the best-DIC models generally don't differ much for those parameters that are included in both models and parameters that are excluded in the best-DIC model show no effect in the full model (posterior 95% quantile range spanning zero). This similarity of results between the models is the reason we now show only the results of the full models in the summary figure Extended Data Fig. 11a (previously main figure 6 but now moved to ED due to editorial request on main figure number limits), as suggested by the reviewer.

Our initial motivation for using information scores was that, even when doing strict inference, we were interested in the underlying explanations for pathogen presence that were as simple as possible and required the least number of variables (based on parsimony). We understand that this approach makes comparisons across models difficult. Ideally, in a fully Bayesian framework, models with different sets of parameters could be averaged based on their respective probabilities, or ranked based on their Bayes factors, but unfortunately, neither of these is possible under the inlabru framework, which is why we used the DIC for simplicity, to prevent the reader from having to look at several large tables, each for a model with different combinations of parameters. However, our results do suggest that the full model in this case fully captures the fact that some parameters do not seem to affect pathogen presence (their posterior 95% quantile range spans 0).

We also agree with the reviewer that the choice of word (“fit”) when referring to the information scores (which also includes penalization of parameters) is wrong, and that has now been corrected throughout the manuscript.

Referee #5 (Remarks to the Author):

The authors have addressed all my comments satisfactorily. I think the manuscript would benefit from including the reply to comment 11 in the supplementary information, as this is pertinent information. However, I leave this up to the authors.

We agree with the reviewer and have now included this information in the supplementary information.

And lastly I want to note, in case the new link will eventually be included in the publication, that the link provided in the last comment worked but the file had no extension and I could only use it because I had seen the extension on the old dropbox link.

We thank the reviewer for spotting this. The data will be shared through an online repository for the final publication.